# Resilient anatomy and local plasticity of naive and stress haematopoiesis

Qingqing Wu[1,12,13 ✉], Jizhou Zhang[1,12,13], Sumit Kumar[1], Siyu Shen[1], Morgan Kincaid[1], Courtney B. Johnson[1], Yanan Sophia Zhang[1], Raphaël Turcotte[2,3], Clemens Alt[2], Kyoko Ito[3], Shelli Homan[1], Bryan E. Sherman[4], Tzu-Yu Shao[4,5], Anastasiya Slaughter[1,5], Benjamin Weinhaus[1,5], Baobao Song[5,6], Marie Dominique Filippi[1,7], H. Leighton Grimes[1,6,7], Charles P. Lin[8,9], Keisuke Ito[3,10], Sing Sing Way[4,7], J. Matthew Kofron[7,11] & Daniel Lucas[1,7 ✉]

The bone marrow adjusts blood cell production to meet physiological demands in response to insults. The spatial organization of normal and stress responses are unknown owing to the lack of methods to visualize most steps of blood production. Here we develop strategies to image multipotent haematopoiesis, erythropoiesis and lymphopoiesis in mice. We combine these with imaging of myelopoiesis[1] to define the anatomy of normal and stress haematopoiesis. In the steady state, across the skeleton, single stem cells and multipotent progenitors distribute through the marrow enriched near megakaryocytes. Lineage-committed progenitors are recruited to blood vessels, where they contribute to lineage-specific microanatomical structures composed of progenitors and immature cells, which function as the production sites for each major blood lineage. This overall anatomy is resilient to insults, as it was maintained after haemorrhage, systemic bacterial infection and granulocyte colony-stimulating factor (G-CSF) treatment, and during ageing. Production sites enable haematopoietic plasticity as they differentially and selectively modulate their numbers and output in response to insults. We found that stress responses are variable across the skeleton: the tibia and the sternum respond in opposite ways to G-CSF, and the skull does not increase erythropoiesis after haemorrhage. Our studies enable in situ analyses of haematopoiesis, define the anatomy of normal and stress responses, identify discrete microanatomical production sites that confer plasticity to haematopoiesis, and uncover unprecedented heterogeneity of stress responses across the skeleton.

The spatial organization of cells in a tissue—its anatomy—dictates their behaviour and profoundly influences their function[2]. Blood cell production takes place in the bone marrow through progressive differentiation of haematopoietic stem cells and progenitors. The bone marrow has extraordinary plasticity and quickly adjusts blood production to meet physiological demands in response to insults[3,4]. Despite recent progress[5–12] the anatomical organization of normal and stress haematopoiesis remains largely unknown. This is because current approaches do not allow simultaneous imaging of most types of haematopoietic progenitors and their daughter cells, in turn precluding in situ analyses of haematopoiesis. Overcoming this hurdle will be indispensable for defining parent and daughter cell relationships and changes in cell behaviour during differentiation, and to identify the cells and structures enabling normal and stress haematopoiesis.

The different types of haematopoietic stem and progenitor cells (HSPCs) have been defined using complex combinations of antibodies against defined cell surface markers[13–17]. Most of the antibody combinations used to isolate HSPC subsets by fluorescence-activated cell sorting (FACS) are not suitable for confocal imaging analyses. We reasoned that an unbiased analytical pipeline might reveal new combinations of surface markers to visualize haematopoiesis in situ (Fig. 1a). We profiled 247 cell surface markers in phenotypically defined stem cells, multipotent progenitors and lineage-committed myeloid and erythroid progenitors using three established cytometric strategies (Fig. 1a, Extended Data

[1]Division of Experimental Hematology and Cancer Biology, Cincinnati Children's Hospital Medical Center, Cincinnati, OH, USA. [2]Center for Systems Biology and Wellman Center for Photomedicine, Massachusetts General Hospital and Harvard Medical School, Boston, MA, USA. [3]Ruth L. and David S. Gottesman Institute for Stem Cell, Regenerative Medicine Research, Department of Cell Biology and Stem Cell Institute, Albert Einstein College of Medicine, Bronx, NY, USA. [4]Division of Infectious Diseases, Center for Inflammation and Tolerance, Cincinnati Children's Hospital Medical Center, University of Cincinnati College of Medicine, Cincinnati, OH, USA. [5]Immunology Graduate Program, University of Cincinnati College of Medicine, Cincinnati, OH, USA. [6]Division of Immunobiology and Center for Systems Immunology, Cincinnati Children's Hospital Medical Center, Cincinnati, OH, USA. [7]Department of Pediatrics, University of Cincinnati College of Medicine, Cincinnati, OH, USA. [8]Advanced Microscopy Program, Center for Systems Biology and Wellman Center for Photomedicine, Massachusetts General Hospital, Harvard Medical School, Boston, MA, USA. [9]Harvard Stem Cell Institute, Cambridge, MA, USA. [10]Department of Medicine, Albert Einstein Cancer Center, Albert Einstein College of Medicine, Bronx, NY, USA. [11]Division of Developmental Biology, Cincinnati Children's Hospital Medical Center, Cincinnati, Ohio, USA. [12]Present address: Department of Hematology, The First Affiliated Hospital of USTC, Division of Life Sciences and Medicine, University of Science and Technology of China, Hefei, China. [13]These authors contributed equally: Qingqing Wu, Jizhou Zhang. ✉e-mail: wuqingqing@ustc.edu.cn; daniel.lucas@cchmc.org

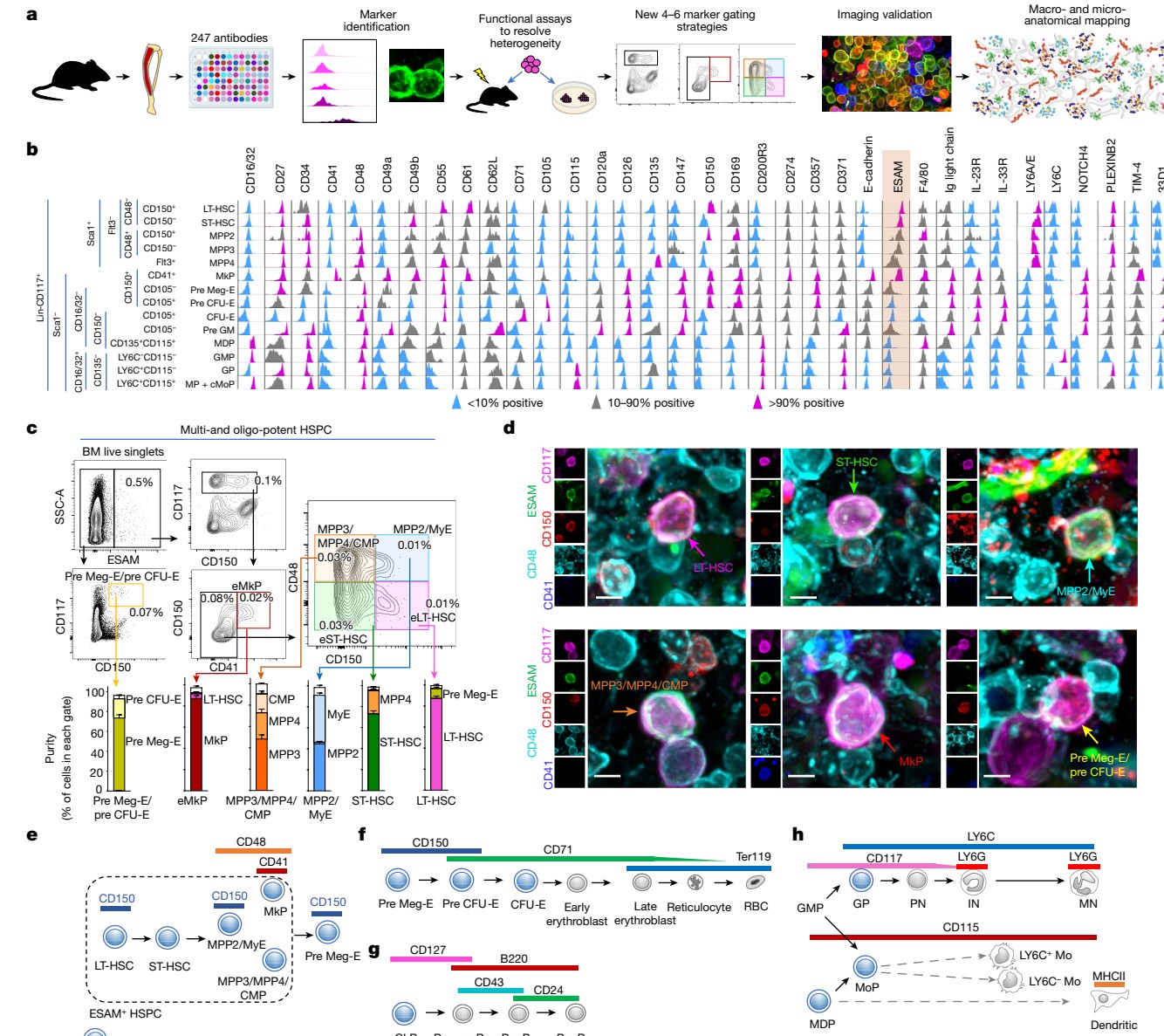

**Fig. 1 | Strategies to image stepwise haematopoiesis. a**, Experimental pipeline. Bone marrow haematopoietic progenitors were immunophenotyped by FACS and differentially expressed markers were identified. When marker expression was heterogeneous, the positive and negative fractions were purified by FACS and their functions were examined in colony-forming or transplantation assays. Guided by this information, we developed staining strategies to enable detection of all cells of interest by FACS followed by validation in whole-mount imaging experiments. Validated strategies were then used to define the anatomy of haematopoiesis. **b**, Histograms showing differential expression of 35 cell surface markers across 14 HSPC types. ESAM (highlighted) expression is restricted to the most primitive cells. Note that pre-GM is a heterogeneous population containing both MDPs and CMPs. LT-HSC, long-term HSC; ST-HSC, short-term HSC; MPP, multipotent pregenitor; MkP, megakaryocyte progenitor; Pre Meg-E, pre-megakaryocyte erythroid progenitor; Pre CFU-E, pre-erythrocyte colony-forming unit; CFU-E, erythrocyte colony-forming unit;

Pre-GM, pre-granulocyte–monocyte progenitor; MDP, monocyte dendritic cell progenitor; GMP, granulocyte–monocyte progenitor; GP, granulocyte progenitor; cMoP, common monocyte progenitor; MP, monocyte progenitors. **c**, FACS isolation strategy for indicated HSPC populations, the histograms show the percentage of classically defined progenitors in each gate ($n = 5$ mice in 3 independent experiments). Data are mean ± s.e.m. BM, bone marrow; CMP, common myeloid progenitor; eLT-HSC, ESAM⁺ LT-HSC; eMkP, ESAM⁺ MkP; eST-HSC, ESAM⁺ ST-HSC; MyE, myeloerythroid progenitor. **d**, Representative images showing identification of indicated HSPCs in whole-mounted sterna. Scale bars, 5 μm. **e**–**h**, Scheme summarizing expression of cell surface markers use to interrogate multipotent cells (**e**), erythropoiesis (**f**), B lymphopoiesis (**g**) and myelopoiesis (**h**). IN, immature neutrophil; MHCII, major histocompatibility complex class 2; MN, mature neutrophil; Mo, monocyte; MoP, monocyte progenitor; PN, preneutrophil; Pre B, pre B cell; Pre-pro B, pre-pro B cell; Pro B, pro B cell; RBC, red blood cell.

Fig. 1a,b, Supplementary Table 1 and refs. 16–18). To be useful for HSPC imaging a marker must: (1) be expressed at sufficient levels for detection (which we experimentally determined to be an absolute fluorescence of $10^3$ over background); and (2) be able to discriminate between at least two types of HSPCs. We thus selected markers that were uniformly expressed in at least one type of HPSC while being absent from one or more HSPC types. Thirty-five markers met these criteria (Fig. 1a,b).

## Imaging multipotent haematopoiesis

In agreement with previous studies, the immunophenotyping showed that ESAM is uniformly expressed in all haematopoietic stem cells[19–21] (HSCs) and subsets of multi- or oligopotent HSPCs but absent in lineage-committed progenitors. Transplantation experiments showed that functional multipotent progenitor 2 (MPP2) and MPP3 cells are

restricted to the ESAM-positive fraction, whereas ESAM⁺ MPP4 cells are up to fivefold more potent than ESAM⁻ MPP4 cells (Fig. 1b and Extended Data Fig. 1c,d).

Transplantation and colony-forming assays also indicated that the 7% of ESAM⁺ pre Meg-E cells (Extended Data Fig. 1c) have monocyte and neutrophil differentiation potential, whereas ESAM⁻ pre Meg-E cells have lost this capacity (Extended Data Fig. 1d–f). We thus defined ESAM⁺ pre Meg-E cells as MyE progenitors and kept the pre Meg-E nomenclature for the ESAM⁻ fraction.

ESAM was also expressed in a small subset of pre-GM cells (Extended Data Fig. 1c). The pre-GM population is a heterogeneous population that contains both CD115⁺ monocyte dendritic cell progenitors and CD115⁻ common myeloid progenitors (CMPs) (Extended Data Fig. 1g). FACS assays indicated that ESAM expression was restricted to a subset of Lin⁻CD117⁺CD115⁻Ly6C⁻CD71⁻CD16/32⁻CD150⁻ CMPs (Extended Data Fig. 1h). This fraction contained all the myeloid engraftment potential, indicating that it represents bona fide CMPs (Extended Data Fig. 1d). These experiments demonstrate that ESAM identifies the functional multi- and oligopotent progenitors in the bone marrow. They also led us to an isolation strategy based on ESAM. This enables simultaneous detection of LT-HSC, ST-HSC, MkP, MPP2 and MyE (containing all functional MPP2s and MyEs) populations, and a mixed population containing functional MPP3s, CMPs and ESAM⁺ MPP4s (Fig. 1c and Supplementary Fig. 1). The eLT-HSC and eST-HSC gates are highly enriched in LT-HSCs and ST-HSCs, respectively (Fig. 1c), and have identical frequencies in limiting dilution competitive transplants as HSCs purified on the basis of SLAM expression (Extended Data Fig. 1i). Each of these six types of HSPC can be detected using five-colour immunofluorescence (Fig. 1d) at similar frequencies when comparing imaging or FACS data, indicating that the strategy detected all cells in the sample (Extended Data Fig. 1j). ESAM also selectively labels Ly6C⁻ sinusoids, Ly6C⁺ arterioles and megakaryocytes (Extended Data Fig. 1k), thus enabling simultaneous interrogation of these important components of the microenvironment.

## Imaging erythropoiesis and lymphopoiesis

The strategy above enabled imaging of pre Meg-E. The next steps of erythrocyte differentiation are pre CFU-E and CFU-E[16]. Functional assays showed that erythroid potential was largely restricted to the CD71⁺ pre CFU-E fraction and that both progenitors can be distinguished from each other and Ly6C⁺CD71⁺ myeloid progenitors on the basis of CD150 and Ly6C expression (Fig. 1b and Extended Data Fig. 2a–c). These results led us to two staining strategies to simultaneously detect all functional pre Meg-E, pre CFU-E, CFU-E and ESAM⁺ HSPC populations or a mixed population of CD117⁺CD71⁺ pre CFU-E and CFU-E cells, and classically defined[22] early and late erythroblasts and reticulocytes by FACS and imaging (Extended Data Fig. 2d–h).

Common lymphoid progenitors (CLPs) can be imaged as CD127⁺Lin⁻ cells[8]. All other steps of B cell maturation can be distinguished on the basis of CD24, CD43, IgM and IgD expression[23]. We combined these strategies to simultaneously image CLP, pre-pro B, pro B and pre B cells (Extended Data Fig. 2i–k).

Armed with these imaging strategies (Fig. 1e–h and Supplementary Table 2) we examined the anatomy of haematopoiesis.

## Distribution of multipotent HSPC

LT-HSC numbers and function are exquisitely regulated by adjacent niche cells[24]. Whether LT-HSCs and downstream progenitors colocalize, and are therefore regulated by the same niche cells, remains an open question[6,9,25]. Imaging of 2-month-old mouse sternum segments showed that all stem cells, and multipotent and oligopotent progenitors are found as single cells with median distances to the closest progenitor of more than 100 µm (more than 10 cell diameters; Fig. 2a,b). To test whether the spatial relationships observed were specific, we compared them with those predicted from random distributions (Extended Data Fig. 3). For most HSPCs we found no differences between the distances from each HSPC measured to all other HSPCs when compared to the random distribution. The exceptions were MPP2/MyE, which were closer to each other than the random distribution, and pre Meg-E, which were further from MPP2/Mye than the random distribution (Fig. 2a,b and Extended Data Fig. 4a).

Most HSPCs showed preferential localization within the microenvironment. At the population level, all the HSPCs except MPP2s were enriched near megakaryocytes when compared to random cells. All HSPCs except ST-HSCs were further than random cells from arterioles. Although sinusoids are a niche for LT-HSCs[26], no HSPC subset preferentially localized to sinusoids or the endosteum (including transcortical blood vessels) when compared with random cells. This is probably owing to the abundance of sinusoids, as most cells localized within 10 µm of these vessels (Fig. 2a,b and Extended Data Fig. 4b–d). These results indicate that LT-HSCs and other multipotent progenitors are not adjacent to each other. Since daughter cells are necessarily adjacent after cell division, the results also indicated that the offspring of HSPCs were either released into the circulation, differentiated into more mature cells that are not detected with the HSPC stain, or moved away from each other. Intravital imaging studies support different degrees of HSC motility in the marrow[9–12]. Because most HSCs are found as single cells in the marrow[27,28] (Fig. 2a) the mobility of HSCs when adjacent to other HSCs or MPPs has not been examined. To explore this in detail we performed follow-up analyses of microscopy-guided transplantation of single Dil-labelled, Tie2⁺ HSCs in the mouse calvarium[29]. In one instance, we observed that (48 h after the initial transplant) the sole transplanted HSC had divided, generating two Dil-labelled, Tie2⁺ cells that were in close proximity. Three hours later, one of the daughter cells was no longer visible in the whole calvarium suggesting that it had moved away or died (Extended Data Fig. 4e). We hypothesized that HSCs move away from each other when in close proximity. Follow-up analyses of single versus multiple cell transplants (5 recipients received a LT-HSC and 6 other recipients received 5, 5, 5, 17, 19 or 22 Tie2⁺ labelled LT-HSCs; the transplanted cells were visualized 15 min after the transfer to confirm correct delivery, and the same region was imaged 24 h later; Fig. 2c) showed that the single transplanted cell was detected in four out of the five recipients of single cells. By contrast, a single donor cell was visualized in one out of the six recipients transplanted with multiple HSCs, whereas no donor cells were detected in the remaining recipients of multiple HSCs (Fig. 2d and Supplementary Table 3). Crucially, all of the recipients showed long-term HSC engraftment that correlated with the number of HSCs transferred (Supplementary Table 3 and ref. 29) indicating that the absent HSCs did not die or terminally differentiate. These experiments suggest that HSCs move away from each other when in close proximity in vivo.

## Production sites for erythropoiesis

In our analyses, all pre Meg-E and pre CFU-E cells were found as single cells through the tissue. pre CFU-E cells separated from pre Meg-E cells and localized in the sinusoids (60% in direct contact) but did not map near CFU-E cells. A previous study showed that CFU-E cells localized to sinusoids[6]. In agreement, we found that CFU-E cells were found in large strings of 3 to 23 cells (mean = 8 ± 4) CFU-E decorating the surface of a single sinusoid and away from arterioles and the endosteum (Fig. 3a–c and Extended Data Fig. 5a,b). Erythroblasts were selectively enriched near CFU-E cells but not pre Meg-E or pre CFU-E cells when compared with random cells (Extended Data Fig. 5c). Indeed, all terminal erythroid cells were selectively enriched within 50 µm (the median distance for random cells) of a CFU-E cell (Fig. 3d–f). Terminal erythropoiesis takes place via sequential downregulation of CD117 and CD71 and upregulation of Ter119[16,22,30]. Higher powered images revealed that when

CFU-E cells detach from the sinusoids, they downregulate CD117 progressively, giving rise to several small clusters of early erythroblasts that bud from the vessel. These progressively upregulate Ter119 to generate large, nearly homogenous clusters of 19 to 96 (mean = 40 ± 4) late erythroblasts that, in turn differentiate into reticulocytes and erythrocytes that remain in close vicinity to the CFU-E strings (Fig. 3d, Extended Data Fig. 5d and Supplementary Video 1). To better understand erythrocyte production, we used *Ubc-creERT2:Confetti* mice. In this model, transient Cre activation leads to irreversible GFP, YFP, RFP or CFP expression in 7.3% of total bone marrow cells. This enables examination of clonal relationships in short-lived cells[1,31]. Confetti fate mapping showed that the CFU-E strings are oligoclonal, whereas the erythroblast clusters are monoclonal (Extended Data Fig. 5e–g). Together, these results indicate that the CFU-E strings identify erythroid production sites—which are formally defined as shown in Extended Data Fig. 5h and Supplementary Table 4—in the sinusoids, where CFU-E cells are recruited to generate defined numbers of red blood cells.

## Production sites for lymphopoiesis

In our analyses, all CLPs were found as single cells localized far (>150 μm) from multipotent HSPCs. Arterioles are a niche for CLPs[8]. In agreement, we found that CLPs were selectively enriched near arterioles and depleted near sinusoids (Fig. 3g,h and Extended Data Fig. 5i–k). Most pre-pro B, pro B and pre B cells were selectively enriched near CLPs, forming loose clusters (2 ± 1 pre-pro B, 3 ± 2 pro B and 16 ± 8 pre B within 150 μm of each CLP). The more mature cells were located further from the CLP, suggesting movement away from the cluster (Fig. 3i–l). *Ubc-creERT2:Confetti* fate mapping showed that these clusters were oligoclonal. We found differentiating cells labelled in the same Confetti colour as the CLP, but these did not map closer to the CLP than expected from random cells (Extended Data Fig. 5l). Together, these experiments suggest that daughter cells move away from the CLP after division but remain associated in loose clusters. This agrees with live-imaging studies that show that pre B cells are highly motile[32]. These results indicated that clusters of CLP and differentiating B cells are oligoclonal B cell production sites (see Extended Data Fig. 5m for step-by-step identification) near arterioles.

## Overall organization of haematopoiesis

We previously identified oligoclonal neutrophil and monocyte and dendritic production sites that selectively localize to distinct sinusoids[1] (Supplementary Fig. 2, Supplementary Table 4). Simultaneous imaging of neutrophil, dendritic and erythroid production sites in the sinusoids showed that these never overlap (Extended Data Fig. 5n,o). Since B lymphopoiesis takes place near arterioles, this indicated that each major blood lineage is produced at specific, non-overlapping production sites. The number of production sites between mice was remarkably consistent, with erythroid and neutrophil sites being the most abundant (Extended Data Fig. 5p). The fact that multipotent HSPC are always found as single cells, whereas lineage-committed progenitors form clusters with daughter cells (Figs. 2a and 3a,i) prompted us to investigate whether this was mediated by cell-autonomous mechanisms. In live-imaging analyses of cultured cells, we found that after cell division, the offspring of HSCs and MPPs rapidly moved away from each other. By contrast, most committed progenitors remained tightly attached after cell division (Fig. 3m,n, Supplementary Fig. 3 and Supplementary Videos 2–6).

These analyses show that the anatomy of haematopoiesis is characterized by different progenitor location and clustering behaviour that changes as cells mature (Fig. 3o). Multipotent HSPCs separate from each other and localize near megakaryocytes and away from arterioles. Lineage-committed progenitors then localize near discrete vessels where they are recruited to lineage-specific production sites with unique spatial and clonal architectures. Immature and mature

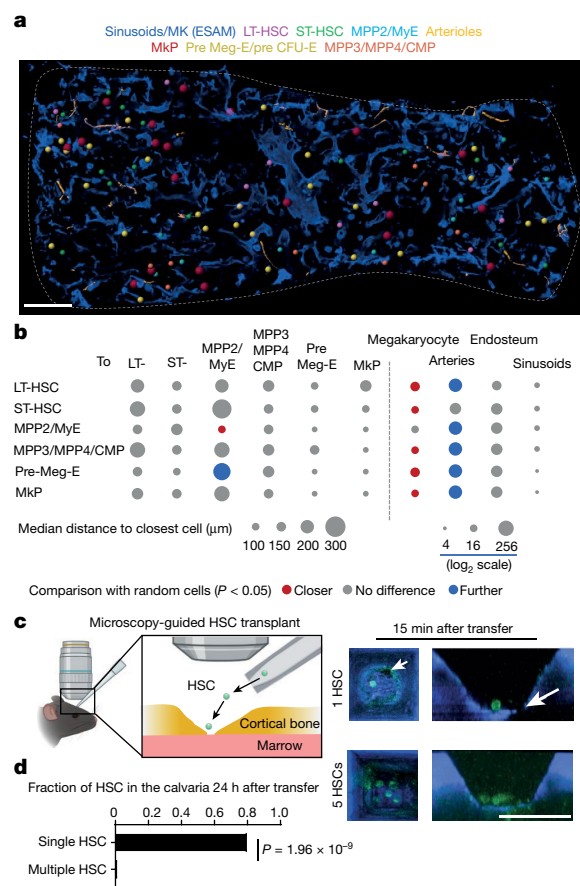

**Fig. 2 | Anatomy of steady-state haematopoiesis in young mice. a,b,** Map showing the location (**a**) and heat map summarizing the median distance from each HSPC to all other indicated cells and structures (**b**) in a 35-μm optical slice of the mouse sternum (n = 35 LT-HSC, 52 ST-HSC, 22 MPP2/MyE, 38 MPP3/MPP4/CMP, 61 MkP, 93 pre Meg-E/pre CFU-E in 5 sternum segments from 4 mice). Statistical differences were calculated using two-tailed unpaired Student's *t*-tests if the data were normally distributed and two-tailed Mann–Whitney test if they were not normally distributed. MK, megakaryocyte. Scale bar, 200 μm. **c**, Tie2[+] HSCs purified from actin-GFP mice were transplanted directly into the calvarial bone marrow of living mice using the approach described in ref. 29 as either single cells (5 recipients received 1 cell) or multiple cells (6 recipients received 5, 5, 5, 17, 19 or 22 cells). Arrows indicate the location of the trafficking single HSC. Scale bar, 50 μm. **d**, The fraction of cells found using intravital microscopy in the whole calvarial bone marrow 24 h following transplantation (***P = 1.96 × 10^{-9}, one-way chi-square test, to compare two proportions). A single cell was visible in 80% of recipients of single cells (4 out of 5) and the cells were all found in close proximity (within 100 μm) of the transplantation site. Only one out of 73 cells was found, as a single cell, in the recipients of multiple cells.

cells leave these production sites to enter the circulation or localize to other bone marrow regions (Fig. 3o,p). This spatial organization is shared across the skeleton, as maps of four other bones (tibia, humerus, lumbar vertebrae and the lambdoid sutures of the skull) revealed almost identical anatomies (Extended Data Fig. 6).

## Resilient anatomy after stress

In response to acute insults, the bone marrow initiates emergency differentiation programmes that lead to marked expansions and/or reductions in the output of one or more blood lineages. This is followed by a return to homeostasis once the insult is removed[33–35]. The lack of tools to visualize differentiation has limited examination of these stress responses in situ. Among the questions that remain are: whether emergency blood production occurs via stress-specific anatomical

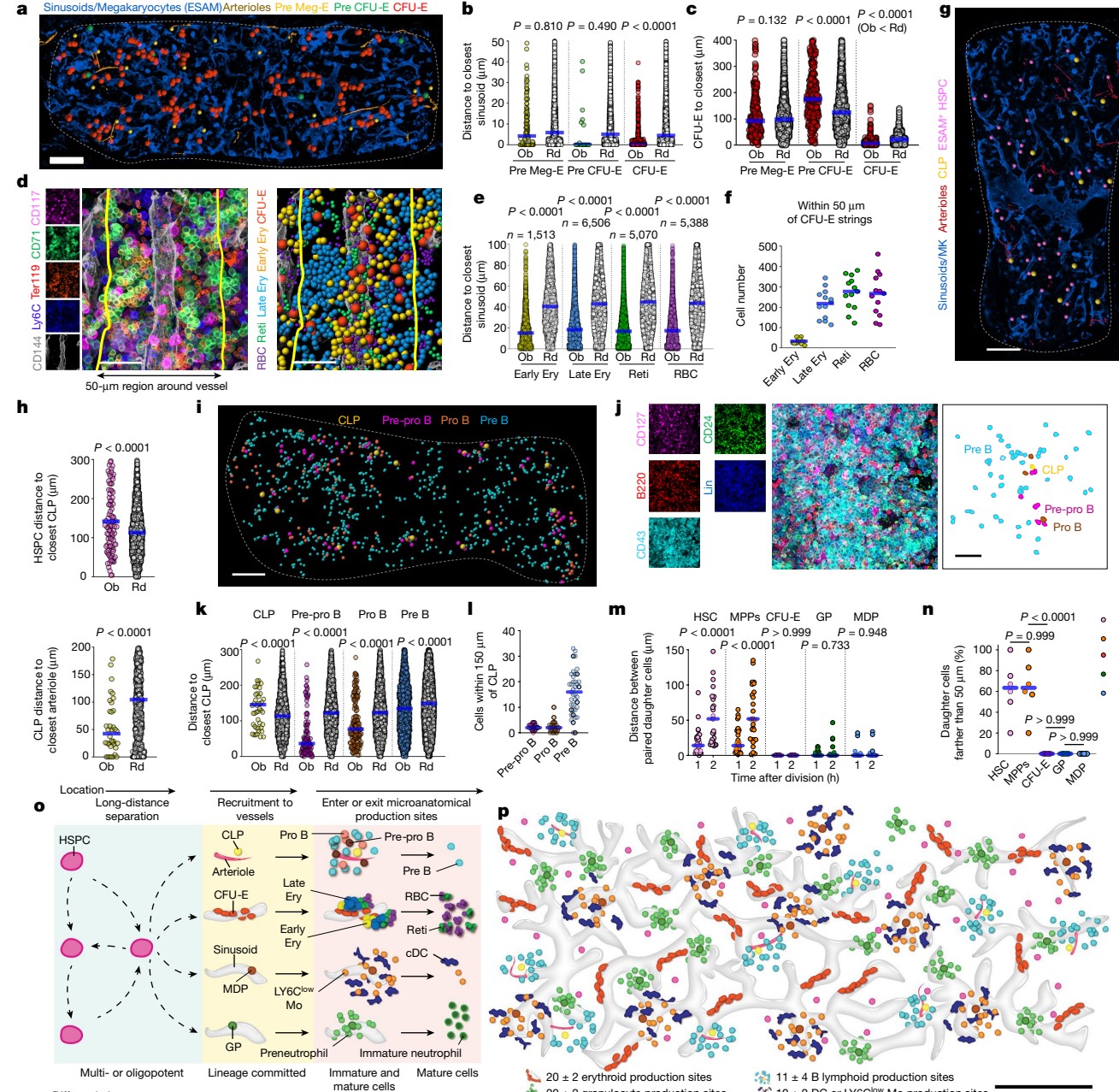

**Fig. 3 | Anatomy of erythropoiesis and lymphopoiesis in the sternum.**
**a–c**, Distribution (**a**) and distance analyses to closest sinusoids of erythroid progenitors (**b**; $n = 111$ pre Meg-E, 18 pre CFU-E and 627 CFU-E) or CFU-E cells to the closest indicated cell (**c**; $n = 318$ CFU-E) in 3 sternum segments from 3 mice. **a**, dots are three times the average size of each cell type. Ob, observed distance; Rd, randomly distributed distance. Scale bar, 200 μm. **d**, Image and scheme of an erythroid production site. Ery, erythrocyte; Reti, reticulocyte. Dots are the average size of each cell type. Scale bar, 50 μm. **e**, Distance analyses from each indicated cell to the closest CFU-E ($n = 5$ sternum segments from 3 mice). **f**, Cell numbers within 50 μm of CFU-E strings ($n = 13$ CFU-E strings randomly selected in 3 sternum segments from 3 mice). **g,h**, Map (**g**) and distance analyses of ESAM⁺ HSPCs to closest CLPs, and CLPs to the closest arteriole (**h**; $n = 104$ ESAM⁺ HSPC, 36 CLP in 3 sternum segments from 3 mice). **g**, Pink dots are three times the average size of the cell type and yellow dots are five times the average size of the cell type. Scale bar, 200 μm. **i,j**, Map and representative production site, illustrating B cell differentiation (the Lin panel contains CD2, CD3, CD5, CD8, CD11b, Ter119, Ly6G, IgM and IgD). **i**, Yellow dots are five times the average size of the cell type. All other dots are three times the average size of each cell type. Scale bars: 200 μm (**i**), 40 μm (**j**). **k**, Distance analyses from each indicated cell to the closest CLP ($n = 50$ CLP, 104 pre-pro B, 162 pro B, 1,932 pre B cells in 3 sternum segments from 3 mice). **l**, Quantification of indicated cells within 150 μm of each CLP ($n = 41$ CLP in 3 sternum segments from 3 mice). **m,n**, Distance between daughter cells at the indicated time points (**m**) and percentage of daughter cells that have separated more than 50 μm (**n**), 2 h after division ($n = 30$ HSC, MPP, CFU-E or MDP and $n = 18$ granulocyte progenitors in 5 independent assays for each indicated progenitors). **o,p**, Schemes showing the anatomy of haematopoiesis in sternum of a two-month-old mouse. Statistical differences were calculated using two-tailed unpaired Student's *t*-tests if the data were normally distributed or two-tailed Mann–Whitney test if they were not normally distributed; *P* values are shown. DC, dendritic cell; cDC, conventional dendritic cell. Scale bar, 200 μm.

structures[5] or exploits the existing structures present during homeostasis; whether these emergency responses are global (all structures in the bone respond to the challenge) or local (only cells in certain bone regions become perturbed); and whether the return to homeostasis also involves restoration of the pre-existing anatomy. To explore these questions, we used three models of acute stress (phlebotomy, *Listeria*

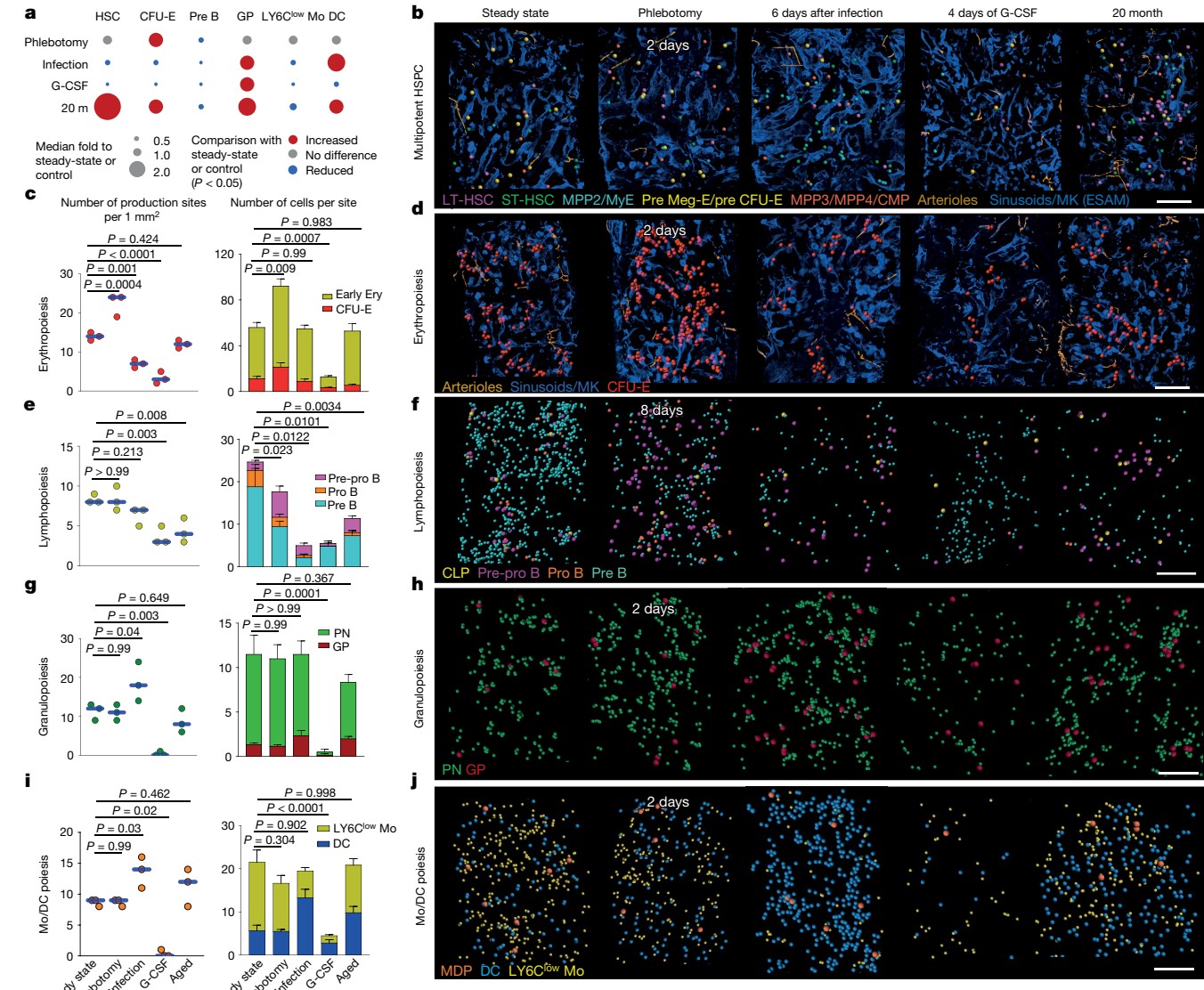

**Fig. 4 | Anatomy of stress responses in the sternum bone marrow. a**, Heat map summarizing changes (normalized to steady state or saline-treated) to the indicated populations in long bones (exemplars of peak erythropoiesis, lymphopoiesis, granulopoiesis and myelopoiesis responses as shown in Supplementary Figs. 4–7) 2 days after phlebotomy (except for lymphopoiesis, which corresponds to day 8 as this is the time point with the lowest lymphopoiesis response, as shown in Supplementary Fig. 4), 6 days after *L. monocytogenes* infection, after 4 days of G-CSF treatment and at 20 months of age (*n* = 7 (2 days after phlebotomy), 6 (6 days after phlebotomy), 4 (G-CSF) and 6 (20-month-old) mice). **b**, Maps showing HSPC location at the indicated time points with indicated challenge. Dots are three times the average size of the relevant cell. **c**–**j**, The number of production sites per mm² (left column; *n* = 3 sternum segments from 3 mice) and the cellularity of each production site (right column; mean ± s.e.m.; *n* = 6 randomly selected production sites in three sternum segments from 3 mice) at the same time point and challenge as in **b**. Maps show the distribution of the indicated cells in a large region of a sternum segment. Statistical differences were calculated using two-way ANOVA *t*-tests if the data were normally distributed and Kruskal–Wallis test if they were not normally distributed. Scale bars, 200 µm.

*monocytogenes* infection and G-CSF treatment) and ageing as a model of chronic impairment of haematopoiesis. These insults cause large expansions and reductions in specific lineages (Fig. 4a and Supplementary Figs. 4–7), while also remodelling the arterioles and sinusoids that support haematopoiesis to varying degrees (Extended Data Fig. 7a–d). Since the anatomy of haematopoiesis in the steady state was similar across the skeleton (Extended Data Fig. 6), we continued to use sternum for these mapping experiments.

Despite major changes in blood production (Fig. 4a), the key anatomical features of haematopoiesis were maintained in all stress models. We found spatial segregation of multi- and oligopotent HSPC (Fig. 4b and Extended Data Fig. 7e) and rare clusters of two to four MPP2 or MyE cells after infection and rare clusters containing both CD41⁻ and CD41⁺ LT-HSCs and clusters of ST-HSCs in old age (Fig. 4b, Extended

Data Fig. 7f–k and Supplementary Fig. 8a–c). Lineage-committed progenitors overwhelmingly mapped to arterioles or sinusoids (although stress caused transient detachment of some progenitors; Supplementary Fig. 9), and mature blood cells were generated in lineage-specific production sites (Fig. 4c–j).

## Production sites enable plasticity

The sternum mapping also showed that the number and output of the production sites adapted to stress in an insult- and lineage-specific manner. After phlebotomy both the number and output of erythroid production sites increased, whereas the numbers of B cell production sites were maintained, but with reduced output. Phlebotomy did not perturb neutrophil and monocyte and dendritic cell (mono/DC)

production sites (Fig. 4c–j). *L. monocytogenes* infection stimulated dendritic cell production by increasing the number of mono/DC sites and changing their fates (to preferential dendritic cell production). It also caused reductions in erythropoiesis and lymphopoiesis by reducing the numbers (for erythroid) and output (both erythroid and lymphoid) of production sites (Fig. 4c–j). G-CSF treatment led to substantial reductions in the numbers and output of all types of production sites examined (Fig. 4b–j). In aged mice, the number of production sites for all lineages was maintained when compared with young mice but erythroid, lymphoid and neutrophil production sites displayed reduced output (Fig. 4b–h). Monocyte and dendritic cell production sites displayed reduced monocyte output, but increased dendritic cell output (Fig. 4i,j). These results indicate that changes at the macro (numbers of production sites) and micro (cell content and output) anatomical level of the production sites orchestrate haematopoietic plasticity to stress (Extended Data Fig. 9e and Supplementary Video 7). Kinetics analyses demonstrated that the changes in the architecture and output of the production sites are fast, synchronous and largely reversible once the acute insult is resolved (Extended Data Figs. 8 and 9). They also indicated that ageing perturbed the production sites (Extended Data Fig. 9a–e).

The increased output of the production sites in response to stress can be mediated by either increased self-renewal of the cells in the site or increased recruitment of progenitors to the site. To distinguish between these two possibilities, we used Confetti mice. Because expression of the fluorescent proteins in this model is irreversible[31], increased self-renewal will necessarily lead to the accumulation of cells labelled by the same fluorescent protein. These analyses did not reveal any increase in the number of Confetti cells, indicating that erythroid and mono/DC production sites expand in response to stress by recruiting additional upstream progenitors (Extended Data Fig. 9f).

These results demonstrated that the basic anatomy of haematopoiesis is durable and resilient to acute insults; that production sites orchestrate haematopoietic plasticity as they adapt their numbers and output to adjust blood production to demand—thus indicating that stress haematopoiesis uses the same structures as steady-state haematopoiesis for generating blood; that all production sites for a given lineage are synchronized as they simultaneously expand or contract in response to insults; that production sites for different lineages are independently regulated; and that the anatomy of haematopoiesis is fully restored once the acute insult is resolved.

## Variable responses across the skeleton

The results showing reductions in neutrophil production sites in the sternum after G-CSF (Fig. 4g,h) were unexpected because G-CSF leads to increases in granulopoiesis in long bones (Fig. 4a and Supplementary Fig. 6). This led us to hypothesize that stress responses vary across the skeleton.

To test this hypothesis, we quantified neutrophil production granulopoiesis in the sternum, tibia and humerus after G-CSF using imaging (Fig. 5a,b) and flow cytometric analyses (Fig. 5c). We found that G-CSF almost doubled the number of granulocyte progenitors and mature neutrophils in long bones when compared with saline controls. In sharp contrast, sternums from the same mice displayed profound reductions in the numbers of granulocyte progenitors and mature neutrophils and a loss of neutrophil production sites (Fig. 5a–c and Extended Data Fig. 10a). The long bones and sternum displayed similar suppression of erythropoiesis, monopoiesis and lymphopoiesis (Extended Data Fig. 10b), suggesting that these bones are equally exposed to G-CSF.

In all the bones, G-CSF treatment led to reductions in the overall number of preneutrophils (Fig. 5a,b). Preneutrophils also mapped farther away from the central granulocyte progenitor when compared to saline-treated controls (Extended Data Fig. 10c–e). This suggested that G-CSF induced faster preneutrophil differentiation and movement away from the central granulocyte progenitor. To examine this

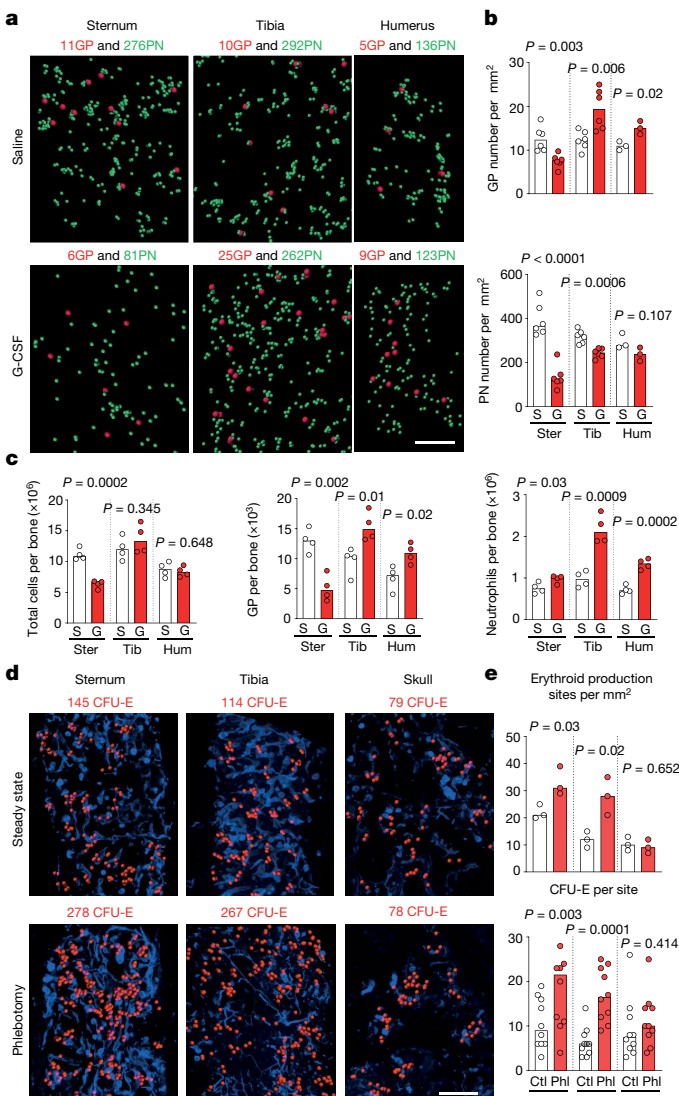

**Fig. 5 | The haematopoietic response to stress varies across the skeleton.**
**a**, Maps showing the distribution of granulocyte progenitors and preneutrophils in whole-mounted sternum, tibia and humerus treated with saline (S) or 250 mg kg⁻¹ day⁻¹ G-CSF (G) for 4 days. Scale bar, 200 μm. **b**, Following G-CSF treatment, the sternum contains fewer granulocyte progenitors and the neutrophil sites disaggregate ($n$ = 6 sternum (ster) or 6 tibia (tib) from 6 mice, or 3 humerus (hum) from 3 mice per treatment). **c**, Number of indicated cells in sternum, tibia, and humerus quantified by FACS ($n$ = 4 mice in 4 independent experiments). **d**, Maps showing the distribution of CFU-Es (terminal erythroid progenitors) in whole-mounted sternum, tibia or lambdoid suture of the skull bone in the steady state or 2 days after phlebotomy. Scale bar, 200 μm. **e**, Erythroid production site frequency (top, $n$ = 3 bones each from 3 mice) and number of CFU-Es per site ($n$ = 10 randomly selected production sites from 3 bones each from 3 mice) in control (ctl) or 2 days after phlebotomy (Phl). Statistical differences were calculated using two-tailed unpaired Student's $t$-tests if the data were normally distributed or two-tailed Mann–Whitney test if they were not normally distributed.

in detail we quantified the distances between the central granulocyte progenitor and clonally related preneutrophils in Confetti mice. In tibia and sternum, G-CSF equally reduced overall granulocyte progenitor output (Extended Data Fig. 10f,g). Additionally, the preneutrophils with the same Confetti label as the central granulocyte progenitor were located much further away from this granulocyte progenitor than in the saline controls. However, these distances were no different between tibia and sternum production sites (Extended Data Fig. 10f).

These results indicate that G-CSF induces preneutrophil localization away from the granulocyte progenitors but that this movement is not faster in the sternum. Together, our results demonstrate differential responses to G-CSF between long bones and sternum.

To determine whether this phenomenon extended to other insults, we induced phlebotomy and examined the response across the skeleton using FACS and imaging. Phlebotomy caused a potent expansion in erythroid production site numbers and output in the sternum, tibia, vertebrae and humerus (Fig. 5d,e and Extended Data Fig. 10h,i). However, we did not detect changes in erythroid production site numbers or output in the skull, even though phlebotomy-induced reductions in lymphopoiesis were similar in all bones, suggesting that the phlebotomy was sensed by haematopoietic cells in the skull (Fig. 5d,e and Extended Data Fig. 10j). These experiments demonstrate that the response of bone marrow production sites to systemic insults is bone- and insult-specific and revealed an unprecedented heterogeneity of stress responses across the skeleton.

## Discussion

Here we have developed strategies to visualize stepwise haematopoiesis across the mouse skeleton. We uncovered a sophisticated and elegant anatomy of haematopoiesis characterized by long-distance spatial separation—probably mediated by cell movement after cell division—of multipotent and oligopotent progenitors that were enriched near megakaryocytes, recruitment of lineage-committed progenitors to distinct blood vessels, and defined microanatomical production sites responsible for producing mature cells for each major blood lineage. This basic anatomy was durable, resilient to acute insults and maintained through the adult lifespan. The bone marrow rapidly adjusted blood cell output to meet physiological demand in response to insults. We showed that this haematopoietic plasticity is mediated by rapid remodelling of the production sites that changed their numbers and output in a lineage- and insult-specific manner. We propose that these production sites and local microenvironments persist through life. A limitation of our study is that we cannot rule out that these lines are motile or transient, as we cannot track the same marrow over time. A second limitation is that we imaged relatively thin slices (35–40 μm), preventing detection of all cells present in the marrow.

Mouse haematopoiesis has almost always been examined in the long bones of the legs. This is because other bones are small and yield limited cell numbers for FACS analyses. Our imaging strategies enabled interrogation of stress responses across the skeleton. Notably, we found that the response to haematopoietic insults varies across the skeleton. We speculate that certain bones have specialized to respond preferentially to specific insults, and this will be the focus of future studies.

The work presented here provides the field with the tools and knowledge necessary to study stepwise blood production in situ, defines the anatomy of normal and stress responses, identifies unique production sites that confer plasticity to haematopoiesis, and uncovers a heterogeneity of stress responses across the skeleton.

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

# Methods

## Data reporting

No statistical methods were used to predetermine sample size. This is because image analyses are extremely time consuming, and it is not possible to examine large numbers of samples. We have previously shown that three bones per condition allow identification of sufficient number of cells to detet changes in location and distribution in the bone marrow[1]. We have thus strived to analyse three bones per condition and included additional bones when possible. All mice were included in the analyses. Mice were randomly allocated to different groups based on the cage, genotype and litter size. For all experiments, we aimed to have the same number of mice in the control and experimental groups. Investigators were not blinded to allocation during experiments and outcome assessment. This is because it was not possible to blind the investigator to the type of bone examined as there are readily identified by shape. Similarly, the insults used generated such evident changes in cellular content (haemorrhage, G-CSF, infection) or shape of the bone (ageing, bones are larger) that it was not possible to blind the investigator to the type of insult examined.

## Mice

All mouse experiments—except live mouse imaging experiments—were approved by the Institutional Animal Care Committee of Cincinnati Children's Hospital Medical Center. Live mouse imaging experiments were performed in compliance with institutional guidelines and approved by the Subcommittee on Research Animal Care (SRAC) at Massachusetts General Hospital. The following mouse strains were used: C57BL/6J-*Ptprc*[b] (CD45.2), B6.SJL-*Ptprc*[a] *Pepc*[b]/BoyJ (CD45.1), B6.Cg-*Ndor1*[Tg(UBC-cre/ERT2)1Ejb]/1 J (*Ubc-creERT2*), C57BL/6-Tg(CAG-EGFP)131Osb/LeySopJ (actin-GFP) and B6.129P2-*Gt(ROSA)26Sor*[tm1(CAG-Brainbow2.1)Cle] (R26R-Confetti). R26R-Confetti mice were crossed with *Ubc-creERT2* mice to generate *Ubc-creERT2:Confetti* mice. All mice were maintained on a C57BL/6 J background. Eight to twelve (2-month-old) and 80 to 100 weeks (20-month-old) male and female mice were used. All mice were bred and aged in our vivarium or purchased from the Jackson Laboratory. Mice were maintained at the vivarium at Cincinnati Children's Hospital Medical Center under a 14-h light:10-h dark schedule, 30–70% humidity, 22.2 ± 1.1 °C, and specific-pathogen-free conditions.

## Tamoxifen treatment

*Ubc-creERT2:Confetti* mice were treated with two pulses of tamoxifen in the diet (400 mg of tamoxifen citrate per kg of rodent diet, Envigo). Each pulse was two weeks long and pulses were two weeks apart. Since committed haematopoietic progenitors do not persist in vivo for longer than two weeks we chased the mice for eight weeks to ensure that all Confetti-labelled immature and mature haematopoietic cells originated from upstream progenitors.

## *L. monocytogenes* infection

The wild-type virulent *L. monocytogenes* strain 10403s was back-diluted from overnight culture for 2 h for early log phase growth ($OD_{600}$ 0.1) in BD Difco brain-heart infusion medium (Thermo Fisher Scientific, 237500) at 37 °C, then washed and diluted in 200 µl sterile saline and injected via the lateral tail vein to mice ($1 \times 10^4$ colony-forming units (CFU) per mouse). Mice were euthanized for bone marrow analyses on day 6 and 20 after infection.

## Phlebotomy mice model

To induce erythropoietic stress by blood loss, isoflurane-anaesthetized mice were phlebotomized (15–20 µl blood per gram of body weight from the retro-orbital venous sinus of mice) with a calibrated heparinized capillary tube. Mice were euthanized for bone marrow analyses on day 2, 8, and 20 after phlebotomy.

## G-CSF treatment

Mice received subcutaneous injections of G-CSF (R&D) twice a day at a dose of 150 µg kg$^{-1}$ for four days. Mice were euthanized for bone marrow analyses 2–3 h after the final morning G-CSF dose at day 5, and 30 days after G-CSF treatment. Mice received subcutaneous injections of 0.1% low endotoxin bovine serum albumin (Sigma) were analysed as control.

## Cell preparation for flow cytometry and cell sorting

Mice were anaesthetized with isoflurane followed by cervical dislocation. For long bones, bone marrow cells were flushed out of the femurs with a 21-gauge needle in 1 ml of ice-cold PEB buffer (2 mM EDTA and 0.5% bovine serum albumin in PBS). For sternum, vertebrae, and skull the bones were chopped into small pieces with scissors in 1 ml of ice-cold PEB buffer. Peripheral blood was collected from the retro-orbital venous sinus of mice, followed by red blood cell lyses with 1 ml of lysis buffer (150 mM $NH_4Cl$, 10 mM $NaCO_3$ and 0.1 mM EDTA). Cells were centrifuged for 5 min at 1,100 rpm under 4 °C, resuspended in ice-cold PEB, and used in subsequent assays. For FACS analyses, cells were stained with a cocktail of biotinylated lineage antibodies for 30 min, washed twice, and stained with streptavidin-conjugated magnetic beads (BD Bioscience, 557812). Magnetic cell depletion was performed according to the manufacturer's protocol. CountBright Absolute Counting Beads (Thermo Fisher Scientific, C36950) were used to count bone marrow and blood cell numbers in a BD LSRFortessa Flow Cytometer (BD Bioscience).

## FACS analyses and LEGENDScreen

Cells were analysed in an LSRFortessa Flow Cytometer (BD Biosciences), LSR II Flow Cytometer (BD Biosciences) or FACS-purified in a FACSAria II Cell Sorter (BD Biosciences) or an SH800S Cell Sorter (Sony Biotechnology). Dead cells and doublets were excluded on the basis of FSC and SSC distribution and DAPI exclusion (Sigma-Aldrich, D9542). Antibodies used were: B220 (clone RA3-6B2), CD2 (clone RM2-5) CD3e (clone 145-2C11), CD4 (clone RM4-5), CD5 (clone 53-7.3), CD8 (clone 53-6.7), CD11b (clone M1/70), CD11c (clone N418), CD16/32 (clone 93), CD24 (clone 30-F1), CD31 (clone A20), CD41 (clone MWReg30), CD42d (clone 1C2), CD43 (clone S11), CD45 (clone 30-F11), CD45.1 (clone A20), CD45.2 (clone 104), CD48 (clone HM48-1), CD71 (clone RI7217), CD105 (clone MJ7/18), CD115 (clone AFS98), CD127 (clone A7R34), CD135 (clone A2F10), CD144 (clone BV13), CD150 (clone TC15-12F12.2), ESAM (clone 1G8), Gr1 (clone RB6-8C5), IgD (clone 11-26 c.2a), IgM (clone RMM-1), Ly6C (clone HK1.4), Ly6G (clone 1A8), Sca-1 (clone D7), Ter119 (clone TER-119), MHCII (clone M5/114.15.2), from BioLegend; CD34 (clone RAM34) and CD117 (clone 2B8), from BioLegend or Thermo Fisher Scientific; CD71 (clone C2) from BD Bioscience. For immunophenotyping experiments, LEGENDScreen Mouse PE Kit (BioLegend, 700005) was used according to the manufacturer's instructions. In brief, fresh bone marrow cells were stained with a cocktail of biotinylated lineage antibodies for 30 min followed by a stain with streptavidin. Cells were washed twice and resuspended at a concentration of $1 \times 10^7$ cells per ml PEB buffer containing antibodies for HSPC identification. Equal amount of cells were transferred into each well of the LEGENDScreen 96-well plates. Plates were incubated for 45 min on ice in the dark. Cells were then washed twice and resuspended in PEB buffer and kept on ice until acquisition on a BD LSRFortessa Flow Cytometer (BD Bioscience). FACS data were analysed with FlowJo software (Tree Star). Dilutions used for each antibody were 1:200, except for CD11b, which was used at 1:500. Gating strategies for all analyses are shown in Supplementary Fig. 10 and Supplementary Fig. 11. Antibodies that did not yield specific signals in confocal imaging are listed in Supplementary Table 5.

## CFU assay

FACS-purified cells were suspended in IMDM + 2% FBS, added into the methylcellulose culture medium (Stem Cell Technologies, MethoCult

M3334, M3434, M3436 and M3534), mixed thoroughly, plated in duplicate 35 mm culture dishes (Greiner Bio-One, 627160), and incubated at 37 °C with 5% $CO_2$ in air and ≥ 95% humidity, for 7–10 days. Colonies were identified and counted based on cluster size and cell morphology using a Nikon Eclipse Ti inverted microscope (Nikon Instruments) equipped with 4×, 10× and 40× objectives.

## Extreme limiting dilution assays

Adult CD45.1[+] recipient mice were lethally irradiated (700 rad plus 475 rad, 3 h apart). Then 15, 7, 3 or 1 FACS-purified CD45.2[+] LT-HSCs or ST-HSCs were mixed with $2 \times 10^5$ CD45.1[+] competitor mouse bone marrow cells and transplanted by retro-orbital venous sinus into lethally irradiated CD45.1[+] recipients within 6 h after the second irradiation. Peripheral blood chimerism was determined by FACS analyses at week 16 post-transplant. HSC frequencies were calculated by using extreme limiting dilution analysis[36].

## Transplant of ESAM[+] and ESAM[−] progenitor subsets in sublethally irradiated recipients

Adult CD45.1[+] recipient mice were sublethally conditioned with a single dose of 700 rad. The indicated number of FACS-purified ESAM[+] or ESAM[−] HSPCs was transplanted via retro-orbital venous sinus injection within 6 h after irradiation. Peripheral blood chimerism was determined by FACS analyses on day 10, 20, 30 and 40 post-transplant.

For transplants of pre Meg-E subsets, we transferred 2,000 ESAM[+] or ESAM[−] pre Meg-E purified from Ubc-GFP mice into CD45.1[+] recipient mice sublethally conditioned with a single dose of 700 rads. Peripheral blood chimerism (including platelets and red blood cells) was determined by FACS analyses on day 6, 12 and 18 post-transplant.

## Whole-mount immunostaining

In experiments requiring visualization of blood vessels in the absence of ESAM, mice were intravenously injected with 10 µg of Alexa Fluor 647 anti-mouse CD144 antibody (BV13, BioLegend) and euthanized 10 min after injection. In experiments requiring visualization of CLP, mice were intravenously injected with 2 µg of Alexa Fluor 647 anti-mouse CD127 antibody (A7R34, BioLegend) and euthanized 5 min after injection. Whole-mount sternum immunostaining has been described[37]. In brief, the sterna were dissected and cleaned of soft and connective tissue, followed by sectioning along the sagittal or coronal plane to expose the bone marrow under a dissecting microscope (Nikon SMZ1500 Stereomicroscope). Each half of the sternum was fixed in 4% PFA (Electron Microscopy Sciences, 15710) in DPBS (Thermo Fisher Scientific, 14190144) for 3 h on ice. Each fragment was further washed with DPBS after fixation and blocked with 10% goat serum (Sigma-Aldrich, G9023) for 1 h, followed by staining with 100 µl staining buffer (2% goat serum in DPBS and the indicated antibodies) on ice. For whole-mount analyses of tibia and humerus the bones were cleaned and soft and connective tissue and bisected along the sagittal plane to expose the bone marrow and then processed as the sternum segments above. For whole-mount analyses of the L5 vertebrae we cleaned the soft and connective tissue and removed the spinal cord. With a surgical blade we removed the body of the vertebrae and bisected it to expose the marrow. For the whole-mount analyses of the lambdoid sutures we dissected the top of the skull from the frontal to occipital bones. Then we used a surgical blade to bisect the lambdoid sutures along the transversal plane. The exposed suture was further bisected by cutting along the horizontal plane to expose the bone marrow inside. All bones were then stained as indicated above for the sternum.

## Confocal imaging

Confocal imaging was performed in a Nikon A1R GaAsP Inverted Confocal Microscope, Nikon A1R LUN-V Inverted Confocal Microscope, or Nikon AXR Inverted Confocal Microscope. Specifications for the Nikon A1R GaAsP Inverted Confocal Microscope: high-power 405 nm, 442 nm, 488 nm, 561 nm, 640 nm and 730 nm solid-state diode lasers. Specifications for the A1R LUN-V Inverted Confocal Microscope: high-power 405 nm, 445 nm, 488 nm, 514 nm, 561 nm and 647 nm solid-state diode lasers. Specifications for the AXR Inverted Confocal Microscope: high-power 405 nm, 445 nm, 488 nm, 514 nm, 561 nm, 594 nm, 640 nm and 730 nm solid-state diode lasers. All microscopes were equipped with a fully encoded scanning $xy$ motorized stage, piezo-$z$ nosepiece for high-speed $z$-stack acquisition, resonant and galvanometric scanners, 1 high-quantum efficiency, low-noise Hamamatsu photomultiplier tube, and three high-quantum efficiency gallium arsenide phosphide photomultiplier tubes (GaAsP-PMTs) for overall 400–820 nm detection. An LWD Lambda S 20XC water-immersion objective (Nikon, MRD77200) was used and images were taken using the resonant scanner with 8× line averaging, 1,024 × 1,024 pixels resolution, and 2-µm $z$-step. For high-power images we used a LWD Lambda S 40XC water-immersion objective (Nikon, MRD77410) with a resonant scanner and 8× line averaging, 1,024 × 1,024 pixels resolution, 0.5-µm $z$-step.

## Image and distance analyses

Original images (.ND2 format file) were denoised by a built-in artificial intelligence algorithm (Denoise.AI) and stitched together using the NIS Elements software (Nikon, version 5.20.02 and 5.30.03). The denoised and stitched ND2 files were converted to Imaris (.IMS) files using Imaris software (Bitplane, version 9.5 to 9.9). Because not all antibodies penetrate to the same depth within the tissue, we only examine the first 35 µm of the sternum image, which we have previously shown are uniformly stained through the tissue[1]. Cells of interest were labelled with dots with the Imaris Spots function in manual mode and the $x$, $y$ and $z$ coordinates of dots were automatically computed. Sinusoids, arterioles, and megakaryocytes were segmented based on channels of CD144, CD41, ESAM and Ly6C using the Imaris Surface function. The diameters of each type of cell were measured manually in 3D view in Imaris software and were as follows: CD41[−] LT-HSC, 8.67 ± 1.23 µm; CD41[+] LT-HSC, 8.94 ± 0.91 µm; ST-HSC, 8.68 ± 1.10 µm; MPP2, 7.98 ± 1.05 µm; MPP3, 8.48 ± 1.32 µm; MkP, 14.45 ± 3.88 µm; pre Meg-E, 9.49 ± 1.34 µm; pre CFU-E, 13.92 ± 1.70 µm; CFU-E, 12.67 ± 1.88 µm; early erythroblast, 8.86 ± 1.61 µm; late erythroblasts, 7.92 ± 1.36 µm; reticulocytes, 5.17 ± 0.76 µm; RBC, 4.38 ± 0.60 µm; CLPs, 7.40 ± 0.97 µm; pre-pro B, 8.9 ± 0.61 µm; pro B, 7.71 ± 1.23 µm; pre B, 6.10 ± 0.61 µm; MDP, 12.13 ± 1.19 µm; GP, 11.70 ± 0.99 µm; PN, 10.21 ± 1.08 µm; Ly6C[low] Mo, 9.30 ± 1.17 µm; cDC, 12.33 ± 2.69 µm. The distance from each cell to the closest vascular structures and megakaryocytes was obtained with the Imaris Distance Transform Matlab Xtension and then subtracted the mean radius for each cell type. The distance between cells was calculated using Matlab software (MathWorks, version 2018a) with the coordinates exported from Imaris and then subtracted the mean radius for each cell. All software were installed in HP Z4 windows 10 x64 workstations equipped with Dual Intel Xeon processor W-2145, 192GB ECC-RAM, and an Nvidia Quadro RTX 5000 16GB GDDR6 graphics card.

## Confetti imaging

For our imaging experiments we used 6 fluorescent channels (405 nm, 445 nm, 488 nm, 514 nm, 561 nm and 647 nm). In the Confetti model, Cre recombination leads to expression of GFP (488 nm), YFP (514 nm), RFP (561 nm) and CFP (445 nm), thus occupying 4 out of 6 channels used for imaging. To overcome this limitation and analyse spatial relationships between Confetti-labelled cells we routinely used a dump channel with Alexa 488 or FITC-labelled antibodies (same fluorescence as GFP). We discarded cells showing green fluorescent from analyses and compared YFP, RFP and CFP labelled cells of interest. To analyse the clonal relationships between CFU-E and erythroblasts, we used Ly6C-Alexa 488, and discarded Ly6C[+]GFP[+] cells from analyses. To analyse the clonal relationships between CLP and B precursors, we used Lin[−] Alexa 488 (the Lin panel contains CD2, CD3e, CD5, CD8, CD11b, Ter119, Ly6G, IgM and IgD), and discarded Lin[+]GFP[+] cells from analyses.

## Random simulations

Sternal fragments were stained with anti-CD45 and anti-Ter119 antibodies to detect all haematopoietic cells, with anti-CD144, anti-ESAM, anti-CD41 and anti-Ly6C to detect sinusoids, arterioles, and megakaryocytes. 3D binary segmentation tools in NIS Elements software were used to automatically annotate CD45+ or Ter119+ cells. In brief, high-resolution images (0.31 µm per pixel $xy$, 0.6 µm per pixel $z$) acquired with a 40× water-immersion objective (NA 1.15) were deconvolved, and CD45 and Ter119 fluorescent membrane channels were added into a single channel with the floating-point math, converted into 12-bit data, and pre-processed to normalize intensities in-depth and min/max intensities. The '3D darkspot detection' algorithm enables the detection of cells of different sizes. This segmentation algorithm considers the distribution of intensities in $x$, $y$ and $z$ 3D region watershed dark centroid to bright membrane. This will account for non-spherical cells and include all dark space inside the cell membrane stain. The generated 'inside cell' binary data was exported to the Imaris software and used to place dots representing each haematopoietic cell (48,964 to 81,248 cells) in each 35-µm optical slice $z$-stack of each sternum fragment. We then used Research Randomizer[38] to randomly select dots representing each type of haematopoietic cell at the same frequencies found in vivo through the bone marrow cavity and measured the distances between these random cells or with vessels as above. Each random simulation was repeated 100–200 times.

To generate random distributions of cells in experiments using Confetti mice, we first obtained the coordinates and Confetti colour for each type of cell in each section analysed. Then we used Research Randomizer to randomize the Confetti label while maintaining the spatial coordinates of each cell. We then measured the distances between these cells with randomized colours. Each random simulation was repeated 100–200 times.

## Production site identification

Production sites for each lineage were identified by comparing the observed distributions of distances with that of random cells as described in each figure.

## Microscopy-guided HSC transplantation in the bone marrow of live animals

Microscopy-guided HSC transplants into the skull of living mice have been reported in detail before[29]. In brief, Tie2+CD150+CD48low/−CD135− Lin−Sca1+Kit+ LT-HSCs were purified from actin-GFP mice or stained with DiI. The skull was then exposed, and the vasculature visualized by rhodamine-B, 70 kMW dextran injection. Second harmonic generation was used to localize bone marrow cavities. Then laser ablation was used to etch a microwell in the bone, with a small opening (about one cell diameter) at the bottom of the microwell that connects to the bone marrow cavity. The opening of the bone marrow cavity was confirmed by lack of second harmonic generation signal and bone marrow leakage. HSCs were loaded in a straight glass micropipette (28–32 µm diameter, Origio) attached to a pump (SAS11/2-E, Research Instruments). Single (1) or multiple (5) HSCs were slowly released into the optical tweezer one at a time and the trapped cells were guided to the bottom of the microwell under image guidance. For the transplant of 17, 19, and 22 cells, multiple cells were first released into the microwell from the micropipette, and the laser tweezer was used to move the cells down to the bottom of the microwell. After the delivery, imaging was performed every 5 min for up to 15 min to ensure that the cell remained at the delivery site. Subsequent imaging was performed as described[29].

## Live-imaging analyses of haematopoietic behaviour after cell division

HSC (Lin−CD117+Sca1+CD48−), MPPs (Lin−CD117+Sca1+CD48+), MDPs, granulocyte progenitors and CFU-Es were purified by FACS and plated in 18-well microplates with liquid medium. Live-cell images were taken using a CIC widefield Nikon Ti2 inverted SpectraX system. Cells were cultured in a Tokai Hit incubation system for 12 h to make sure cells were fully decanted. Live-cell images were taken every 15 min for 36 h. HSC and MPP were cultured in F12 medium supplemented with 10 mM HEPES, 1× penicillin–streptomycin–glutamine (P/S/G), 1× insulin–transferrin–selenium–ethanolamine (ITSX), 1 mg ml⁻¹ polyvinyl alcohol (PVA), 100 ng ml⁻¹ thrombopoietin (TPO), and 10 ng ml⁻¹ stem cell factor (SCF). MDP and granulocyte progenitors were cultured in Iscove's Modified Dulbecco's Medium with 25 mM HEPES and L-glutamine containing 10% (vol/vol) FBS, 1 mM sodium pyruvate, penicillin (100 U ml⁻¹) and streptomycin (100 µg ml⁻¹) with a combination of cytokines (50 ng ml⁻¹ SCF, 20 ng ml⁻¹ LIF, 10 ng ml⁻¹ IL-3, 20 ng ml⁻¹ IL-6). CFU-E were cultured in Iscove's Modified Dulbecco's Medium with 25 mM HEPES and L-glutamine containing 10% (vol/vol) FBS, 1 mM sodium pyruvate, penicillin (100 U ml⁻¹) and streptomycin (100 µg ml⁻¹) with a combination of cytokines (3.0 U ml⁻¹ recombinant human EPO, 10 ng ml⁻¹ recombinant mouse IL-3, 10 ng ml⁻¹ recombinant mouse IL-6, 25 ng ml⁻¹ recombinant mouse SCF and 50 ng ml⁻¹ recombinant mouse TPO). All cytokines were purchased from Stem Cell Technologies.

## Quantifications of vessel length, diameter and branching

Bone marrow vessels were detected based on ESAM and Ly6C expression (sinusoids ESAM+Ly6C−, arterioles ESAM+Ly6C+). We defined a branch as the point where two or more lumens connect. A vessel is a vascular structure—with a continuous lumen—between two branching points. Vessel length and diameter were measured manually using the measurement tool in Imaris. Diameter reported was the largest value for the whole vessel.

## Statistics

All statistical analyses were performed using Prism 9 (GraphPad Software). For graphs quantifying cells in different mice, we indicate the mean, and each dot corresponds to one mouse. For graphs showing distances between cells or structures, or quantifying cells in production sites, we indicate the median or mean respectively, and each dot corresponds to one cell or production site as indicated. Statistical analyses between two samples were performed by using Student's $t$-test if the data were normally distributed and Mann–Whitney test if the data were not normally distributed. For statistical analysis between multiple samples analyses were performed using two-way ANOVA followed by Sidak's multiple comparisons test if the data were normally distributed or Kruskal–Wallis test if they were not normally distributed. No statistical methods were used to predetermine sample size.

## Reporting summary

Further information on research design is available in the Nature Portfolio Reporting Summary linked to this article.

## Data availability

For Fig. 1, image files are available at https://doi.org/10.17632/27wvzpyf5h.3. Flow cytometry datasets are available at https://doi.org/10.17632/9m2m4bcz4p.1, https://doi.org/10.17632/cpbz9f6pbc.1, https://doi.org/10.17632/nynrk39fww.1, https://doi.org/10.17632/w4yxd3crty.1, https://doi.org/10.17632/w9jf53792g.1, https://doi.org/10.17632/bf87grp7m6.1 and https://doi.org/10.17632/3r2w6x8v6f.1. For Fig. 2, image files are available at https://doi.org/10.17632/27wvzpyf5h.3, https://doi.org/10.17632/z3mdsdyw8d.1, https://doi.org/10.17632/gnyzrhx33v.1 and https://doi.org/10.17632/zvynjf48j5.1. For Fig. 3, image files are available at https://doi.org/10.17632/27wvzpyf5h.3, https://doi.org/10.17632/fdgkc5w74w.2 and https://doi.org/10.17632/4ncmwf3mw6.2. For Fig. 4,

image files are available at https://doi.org/10.17632/m9mhc9k6dc.2, https://doi.org/10.17632/zm8xwmc66r.2, https://doi.org/10.17632/7y9ymhzbhh.2, https://doi.org/10.17632/33zw4pz2cr.2, https://doi.org/10.17632/tf37wycrmf.2 and https://doi.org/10.17632/54r2vgxnsx.2; flow cytometry datasets are available at https://doi.org/10.17632/3vvz5nt8g4.1, https://doi.org/10.17632/pf6sxfm4vd.1, https://doi.org/10.17632/ttdks4rtxm.1, https://doi.org/10.17632/mhxvc5ndzm.1, https://doi.org/10.17632/g96gctkbzs.1 and https://doi.org/10.17632/g3dtf8474d.2. For Fig. 5, image files are available at https://doi.org/10.17632/94pnmbhysg.2, https://doi.org/10.17632/r55xkk7x4f.1, https://doi.org/10.17632/7p3xmtcfnz.1, https://doi.org/10.17632/z34pb5bhrw.1, https://doi.org/10.17632/nd7d275yfb.1, https://doi.org/10.17632/vxvnmyxj3h.1 and https://doi.org/10.17632/bkp5ftg3mf.1; flow cytometric datasets are available at https://doi.org/10.17632/pf6sxfm4vd.1 and https://doi.org/10.17632/3vvz5nt8g4.1. For Extended Data Fig. 1, image files are available at https://doi.org/10.17632/p5bjvxcw5d.1; flow cytometry datasets are available at https://doi.org/10.17632/j7br6w8d8w.1, https://doi.org/10.17632/cs6ptxf5r2.1 and https://doi.org/10.17632/g9rn83p7zd.1. For Extended Data Fig. 2, image files are available at https://doi.org/10.17632/54r2vgxnsx.3 and https://doi.org/10.17632/p5bjvxcw5d.1; flow cytometry datasets are available at https://doi.org/10.17632/g9rn83p7zd.1. For Extended Data Fig. 3, image files are available at https://doi.org/10.17632/fhswzm84vb.1. For Extended Data Fig. 4, image files are available at https://doi.org/10.17632/y393hyhdvp.2. For Extended Data Fig. 5, image files are available at https://doi.org/10.17632/br3h29mx95.2. For Extended Data Fig. 6, image files are available at https://doi.org/10.17632/7p3xmtcfnz.1, https://doi.org/10.17632/pp7v4hbyc9.1, https://doi.org/10.17632/s5kk53kwv2.1, https://doi.org/10.17632/2hyzp4zdp2.1, https://doi.org/10.17632/drpjcx35sx.1, https://doi.org/10.17632/knpm7xdsc2.1, https://doi.org/10.17632/vgjfjnh8gv.1, https://doi.org/10.17632/pzxxdp2fgf.1, https://doi.org/10.17632/jps7g9x7nx.1, https://doi.org/10.17632/br3h29mx95.2, https://doi.org/10.17632/m35v8w7vwk.1 and https://doi.org/10.17632/n2fy465m3d.1. For Extended Data Fig. 7, image files are available at https://doi.org/10.17632/6bz9ffr3gh.1 and https://doi.org/10.17632/nrmy4w5sx3.1; flow cytometry datasets are available at https://doi.org/10.17632/g9rn83p7zd.1. For Extended Data Fig. 8, image files are available at https://doi.org/10.17632/zr85nd3bdg.1, https://doi.org/10.17632/j2ggw5j822.1, https://doi.org/10.17632/svt57k426w.1, https://doi.org/10.17632/7w4dcrzdmv.1 and https://doi.org/10.17632/rftzxh2ch9.1. For Extended Data Fig. 9, image files are available at https://doi.org/10.17632/tjm4xkrtft.1, https://doi.org/10.17632/5t7sdpn6x5.1 and https://doi.org/10.17632/gtj646wfh8.1. For Extended Data Fig. 10, image files are available at https://doi.org/10.17632/5t7sdpn6x5.1, https://doi.org/10.17632/fh57twv7yh.1, https://doi.org/10.17632/6mygy39dr5.1, https://doi.org/10.17632/tjm4xkrtft.1, https://doi.org/10.17632/khwfy4xz3r.1, https://doi.org/10.17632/hgvmztk9ff.1, https://doi.org/10.17632/34xbrvzb8v.1, https://doi.org/10.17632/484t7hf6m6.1, https://doi.org/10.17632/pp7v4hbyc9.1, https://doi.org/10.17632/ygxphvxr8y.1 and https://doi.org/10.17632/s5kk53kwv2.1; flow cytometry datasets are available at https://doi.org/10.17632/pf6sxfm4vd.1 and https://doi.org/10.17632/3vvz5nt8g4.1. Source data are provided with this paper.

## Code availability

Matlab script for distance analysis is available at https://doi.org/10.17632/ndngps45s2.1.

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

**Acknowledgements** The authors thank the Confocal Imaging Core, the Research Flow Cytometry Core at the Division of Rheumatology, the Veterinary Services, and the Media lab at Cincinnati Children's Medical Center for experimental and technical assistance. This work was supported by the National Heart Lung and Blood Institute. We thank A. Hidalgo for critical reading of this manuscript. D.L. is supported by R01 HL136529, R01 HL153229, R01 HL158616, R01HL160614 and U54 DK126108 and is a scholar of the Leukemia and Lymphoma Society. J.Z. was supported by The Edward P. Evans Foundation. M.D.F. is supported by R01 HL151654. H.L.G. is supported by R01 HL122661. S.S.W. is supported by the NIH through award DP1AI131080, the HHMI Faculty Scholar's Program (grant no. 55108587), Burroughs Wellcome Fund, and the March of Dimes Foundation Ohio Collaborative. C.P.L. and Keisuke Ito were supported by R01DK115577 (Multi PI). Keisuke Ito is also supported by R01DK098263, R01HL148852, R01HL069438 and U2CDK129502, and is a scholar of the Leukemia and Lymphoma Society. Data were generated using an SH800 cell sorter funded by NIH grant S10OD023410. The 96-well plate scheme in Fig. 1a, the 6-cm plate scheme in Fig. 1a and the mouse head, microscope and pipette tip in Fig. 2c were adapted from BioRender.com.

**Author contributions** D.L. conceptualized and managed the study. D.L., J.Z., Q.W., J.M.K., H.L.G., C.P.L, Keisuke Ito and M.D.F. designed experiments. Q.W. and J.Z. performed most imaging experiments and analyses. R.T., C.A. and Kyoko Ito assisted imaging analyses. C.B.J., A.S., B.W., M.K., B.S., S.H., S.K., S.S. and Y.S.Z. maintained mice and collected data. B.E.S., T.-Y.S. and S.S.W. performed infection experiments. D.L., Q.W. and J.Z. assembled the figures and wrote the manuscript, with editorial input from all authors.

**Competing interests** The authors declare no competing interests.

**Additional information**
**Correspondence and requests for materials** should be addressed to Qingqing Wu or Daniel Lucas.

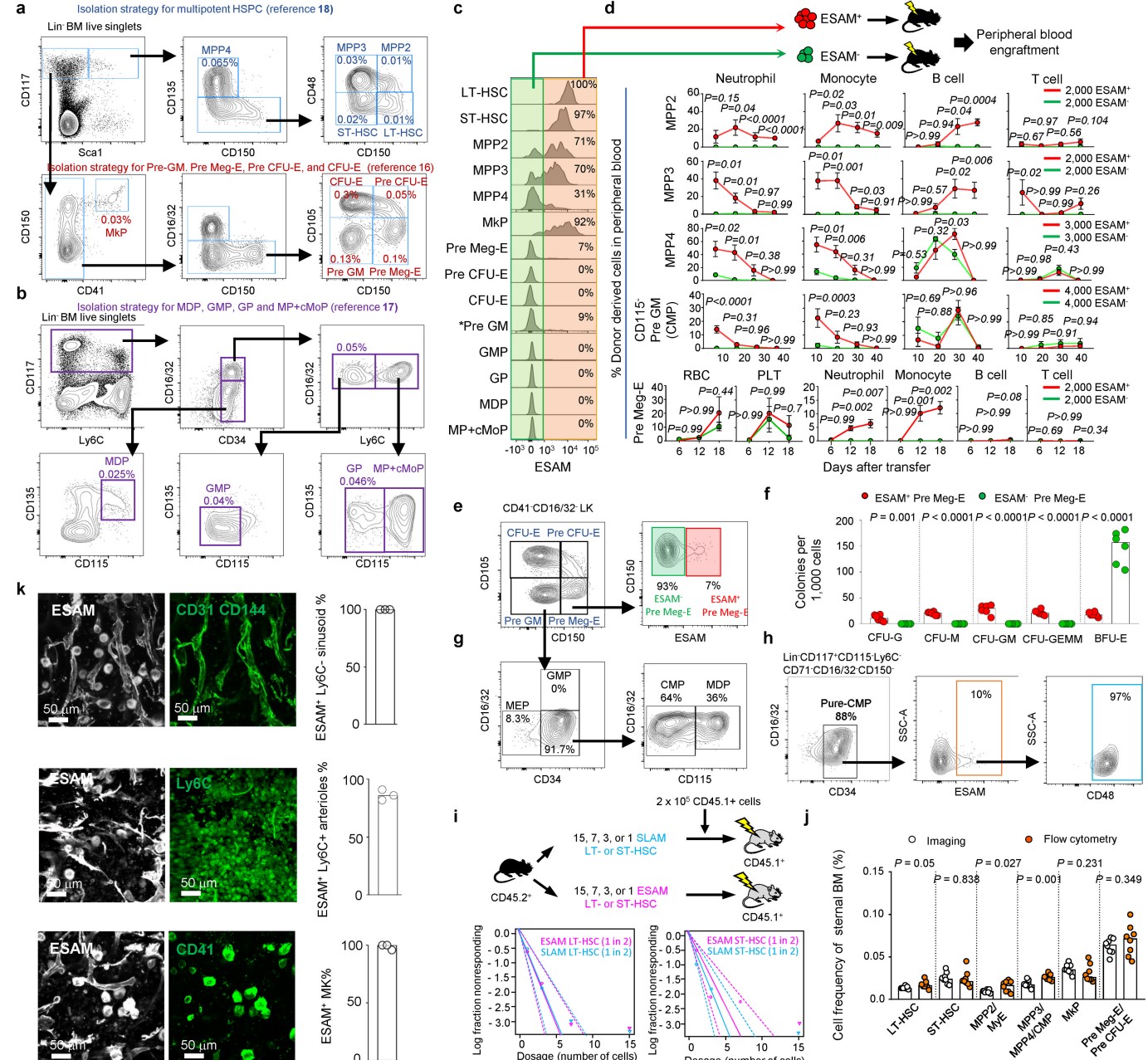

**Extended Data Fig. 1 | Development and validation of strategies to map hematopoiesis. a**, **b**, FACS plots showing the gating strategies used –as previously described[16–18]- to interrogate the 14 progenitors immunophenotyped in Fig. 1. **c**, ESAM expression in indicated HSPC. **d**, Percentage of donor-derived cells in the peripheral blood after transplantation of the indicated progenitors (n = 4 mice/per group). All data represent mean ± s.e.m. Statistical analysis were performed using Two-way ANOVA, followed by Multiple unpaired two tailed t tests, P values are shown. **e**, **f**, FACS plots (e) and colony-forming assays (f) showing that Lin⁻CD117⁺Sca1⁻CD41⁻CD16/32⁻CD105⁻CD150⁺ Pre Meg-E from (a) are heterogeneous and contain ESAM⁺ cells with myeloid and erythroid colony-forming activity (ESAM⁺ MyE) and bona fide Pre Meg-E (ESAM⁻) (n = 6 mice from 3 independent experiments). Statistical differences were calculated using two-tailed unpaired Student's t-test if the distributions were normal and two-tailed Mann-Whitney if not normal. P values are shown **g**, FACS plots showing that the Lin⁻CD117⁺Sca1⁻CD41⁻CD16/32⁻CD105⁻CD150⁻ Pre GM from (a) are heterogeneous and contain both common myeloid progenitors (CMP)

and monocyte dendritic cell progenitors (MDP). **h**, FACS plots showing the gating strategy to identify pure CMP and expression of ESAM and CD48 in these CMP. **i**, Experimental scheme and plots of extreme limiting dilution analyses (16 weeks after transplantation) showing the estimated LT- and ST-HSC frequency (solid bars) and confidence intervals (dotted lines) in the bone marrow of recipients transplanted with the indicated numbers of SLAM or ESAM LT- or ST-HSC (n = 15 recipient mice in independent experiments per group and dilution except for the single cell transplanted group were n = 20). Cell frequencies were calculated using Extreme Limiting Dilution Analysis[36]. **j**, Cell frequencies detected by FACS (white) or confocal imaging (orange) in the sternum when using the strategy shown in Fig. 1c (n = 9 mice for each group). Statistical differences were calculated using two-tailed unpaired Student's t-test if the distributions were normal and two-tailed Mann-Whitney if not normal. P values are shown **k**, Representative images and quantification demonstrating that ESAM stains all CD31CD144⁺ bone marrow sinusoids, Ly6C⁺ arterioles, and CD41⁺ megakaryocytes. n = 3 sternum segments from 3 mice.

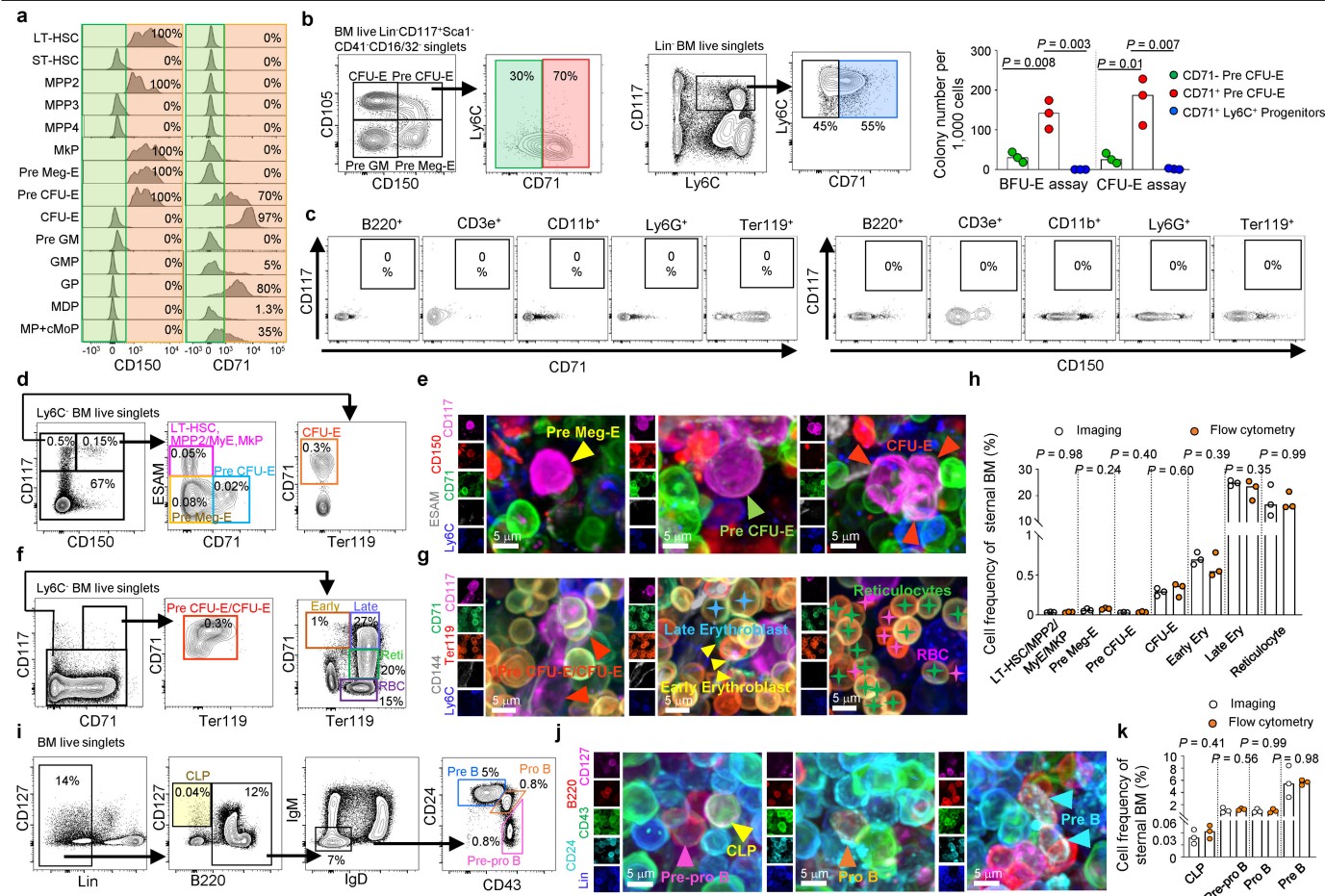

**Extended Data Fig. 2 | Development and validation of strategies to map hematopoiesis. a**, Bona-fide CD150$^+$ Pre Meg-E and Pre CFU-E do not express ESAM whereas all other CD150$^+$ HSPC are ESAM$^+$ (compare with Extended Data Fig. 1c). CD71 -a classical marker of erythroblasts- is highly expressed by all CFU-E, 70% of Pre CFU-E, and 30–80% of Ly6C$^+$ monocyte and neutrophil progenitors but absent in all other HSPC. **b**, To functionally resolve these heterogeneous populations we assessed erythroid differentiation potential using colony-forming assays. CD71$^+$ Pre CFU-E contain almost all the erythroid activity whereas Ly6C$^+$CD71$^+$ progenitors were incapable of generating erythroid cells (n = 3 mice in 3 independent experiments). **c**, Expression analyses showed that mature hematopoietic cells do not coexpress CD71 and CD117 or CD150 and CD117. **d-g**, These led us to two FACS strategies to simultaneously detect all functional Pre Meg-E, Pre CFU-E, CFU-E, and ESAM$^+$ HSPC (d, e) or a mixed

population of CD117$^+$CD71$^+$ Pre CFU-E and CFU-E, and classically defined early and late erythroblasts, reticulocytes, and erythrocytes by FACS and imaging (f, g). **h**, Frequencies for the indicated 8 populations in the sternum bone marrow by FACS (white) or imaging (orange) (n = 3 mice for each group). **i-j**, FACS gating strategies (i) for isolation of -and representative images (j) - of B lymphopoiesis (the Lin panel contains CD2, CD3e, CD5, CD8, CD11b, Ter119, Ly6G, IgM, and IgD). **k**, Frequencies for the indicated B cell populations in the sternum bone marrow by FACS (white) or imaging (orange) (n = 3 mice for each group). All data represent individual values with median plot. Unless otherwise indicated statistical differences were calculated using two-tailed unpaired Student's t-test if the distributions were normal and two-tailed Mann-Whitney if not normal. P values are shown.

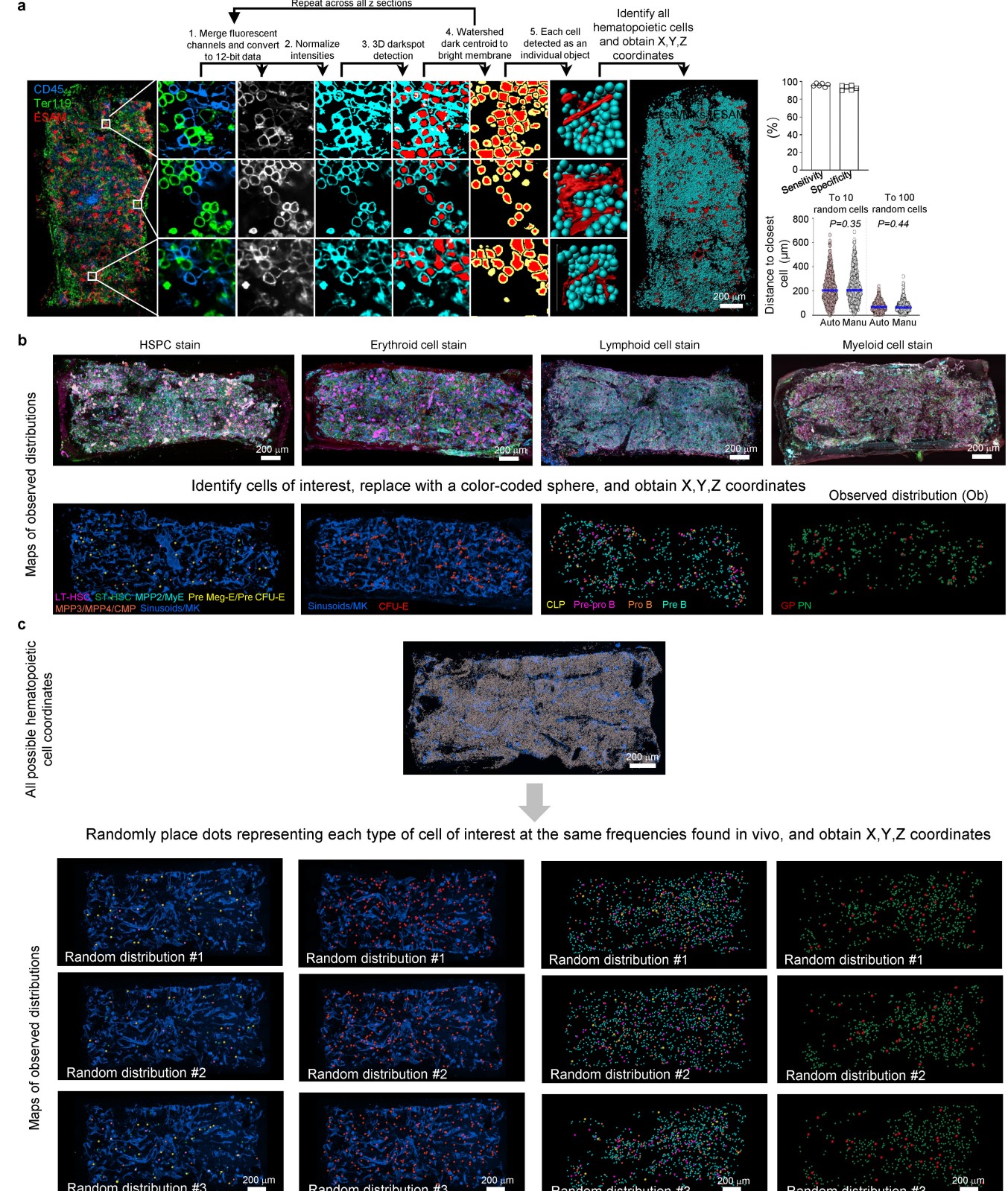

**Extended Data Fig. 3** | See next page for caption.

**Extended Data Fig. 3 | Strategies for automatic segmentation and random simulations. a**, Experimental workflow for automatic cell segmentation. Fluorescence in the individual channels is merged, converted to 12-bit data, and intensities normalized. The "3D darkspot detection" algorithm detects cells of different sizes. We then watershed each dark centroid to the bright membrane. This is repeated across all z sections until each cell is annotated as an individual object. The generated "inside cell" binary data was exported from Nikon Elements software to Bitplane Imaris software and used to place dots representing each hematopoietic cell. The upper histogram shows sensitivity (= number of correctly segmented cells divided by the number of cells identified manually) and specificity (= number of correctly segmented cells divided by the number of identified cells using the automatic algorithm) of cell segmentation (n = 6 areas in 2 sternum segments from 2 mice). The lower histograms compare the distribution of distances between randomly selected cells segmented manually or through the automatic algorithm. Statistical differences were calculated using two-tailed unpaired Student's t-tests; P values are shown. **b**, Experimental workflow to perform statistical analyses by comparing observed distributions with that of random cells. Stained bone marrows are imaged, all cells and structures of interest identified, and their X, Y, Z coordinates recorded. The cells of interest are replaced with color coded spheres corresponding to each cell type to generate maps with the observed distributions found in vivo. The cell coordinates are then used to quantify distances between any cell and structure of interest. In parallel, different bones are stained with anti-CD45 and anti-Ter119 antibodies (to detect all hematopoietic cells in the tissue) and ESAM and Ly6C (to detect megakaryocytes, sinusoids, and arterioles). **c**, We then obtain the coordinates of all (CD45[+] and/or Ter119[+]) hematopoietic cells as well as sinusoids, arterioles, and megakaryocytes (identified as shown in a). The coordinates of the hematopoietic cells are used to randomly place dots –representing each type of hematopoietic cells for which random simulations are desired- at the same frequencies found vivo to generate random distributions. The coordinates of the selected random dots are then used to measure the distances between these random cells or to sinusoids, megakaryocytes, or arterioles. Each random simulation is repeated a hundred times.

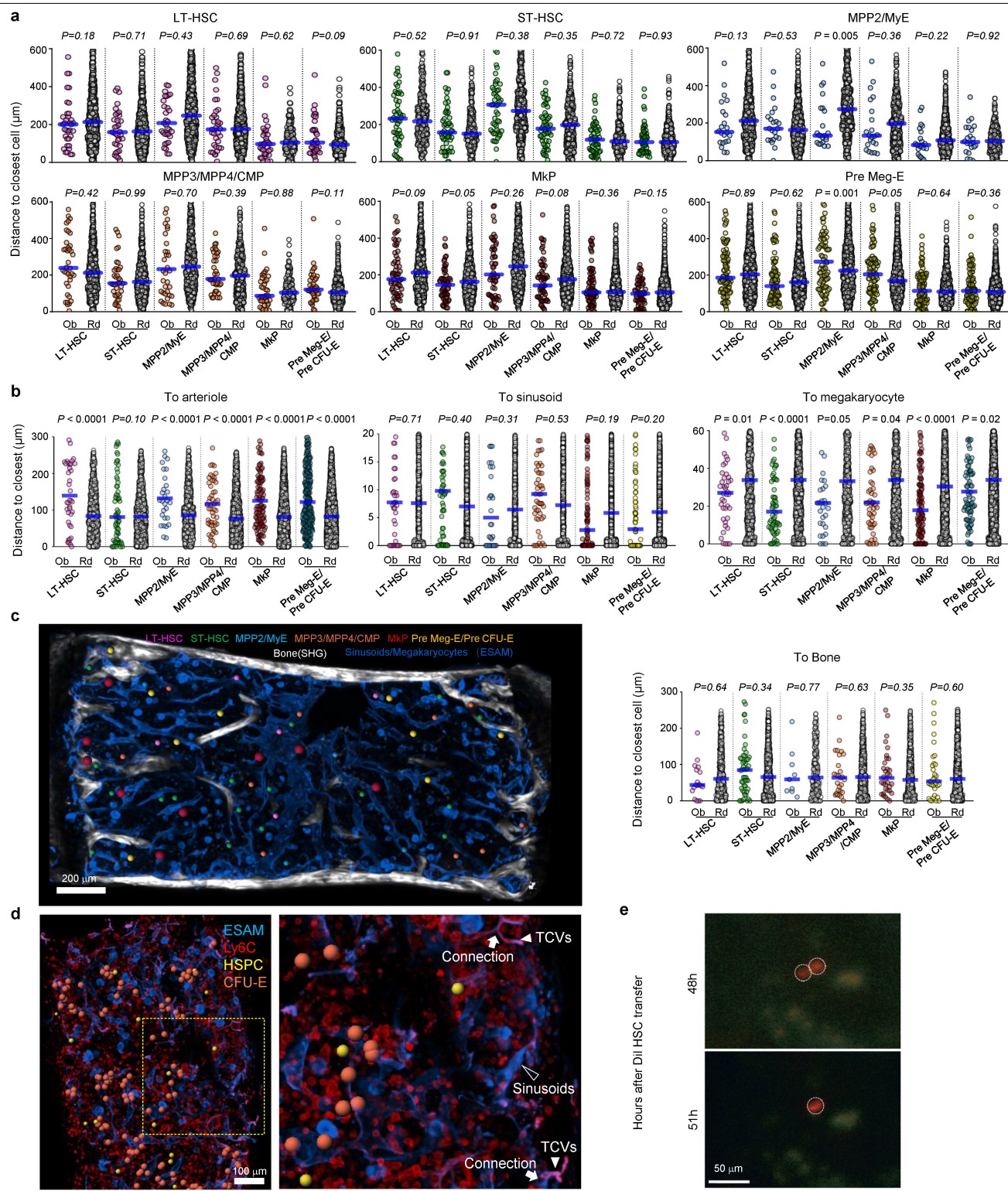

**Extended Data Fig. 4 | HSPC localization in the mouse sternum. a**, Histograms showing the distribution of distances from each HSPC to the closest indicated cell (n = 66 ST-HSC, 30 MPP2/MyE, 41 MPP3/MPP4/CMP, 61 MkP, 82 Pre Meg-E/Pre CFU-E in 5 sternum segments from 4 mice). **b**, Distance analyses from each indicated cell to the closest sinusoid, arteriole, or megakaryocyte (n = 42 LT-HSC, 63 ST-HSC, 32 MPP2/MyE, 55 MPP3/MPP4/CMP, 117 MkP, 85 Pre Meg-E in 5 sternum segments from 3 mice). **c**, Map showing the distribution of the indicated HSPC in relation to the bone (detected using second harmonic generation). The histogram indicates that the HSPC do not map near bone surfaces (n = 19 LT-HSC, 50 ST-HSC, 9 MPP2/MyE, 27 MPP3/MPP4/CMP, 34 MkP, 35 Pre Meg-E in 4 sternum

segments from 3 mice). **d**, In agreement, neither ESAM⁺ HSPC nor CFU-E localized near the transcortical blood vessels (TCV) that penetrate the bone. **e**, Intravital calvaria bone marrow confocal images 48 hours (top) and 51 hours (bottom) following microscopy-guided transplantation of a Tie2⁺ HSC co-labelled with DiI. As a single cell was transplanted, the two cells visible at 48 hours are necessarily originating from cell division (Bottom). One of the two cells visible at 48 hours was no longer visible in the whole calvaria at 51 hours. For all panels: statistical differences were calculated using two-tailed unpaired Student's t-test if the distributions were normal and two-tailed Mann-Whitney if not normal. P values are shown.

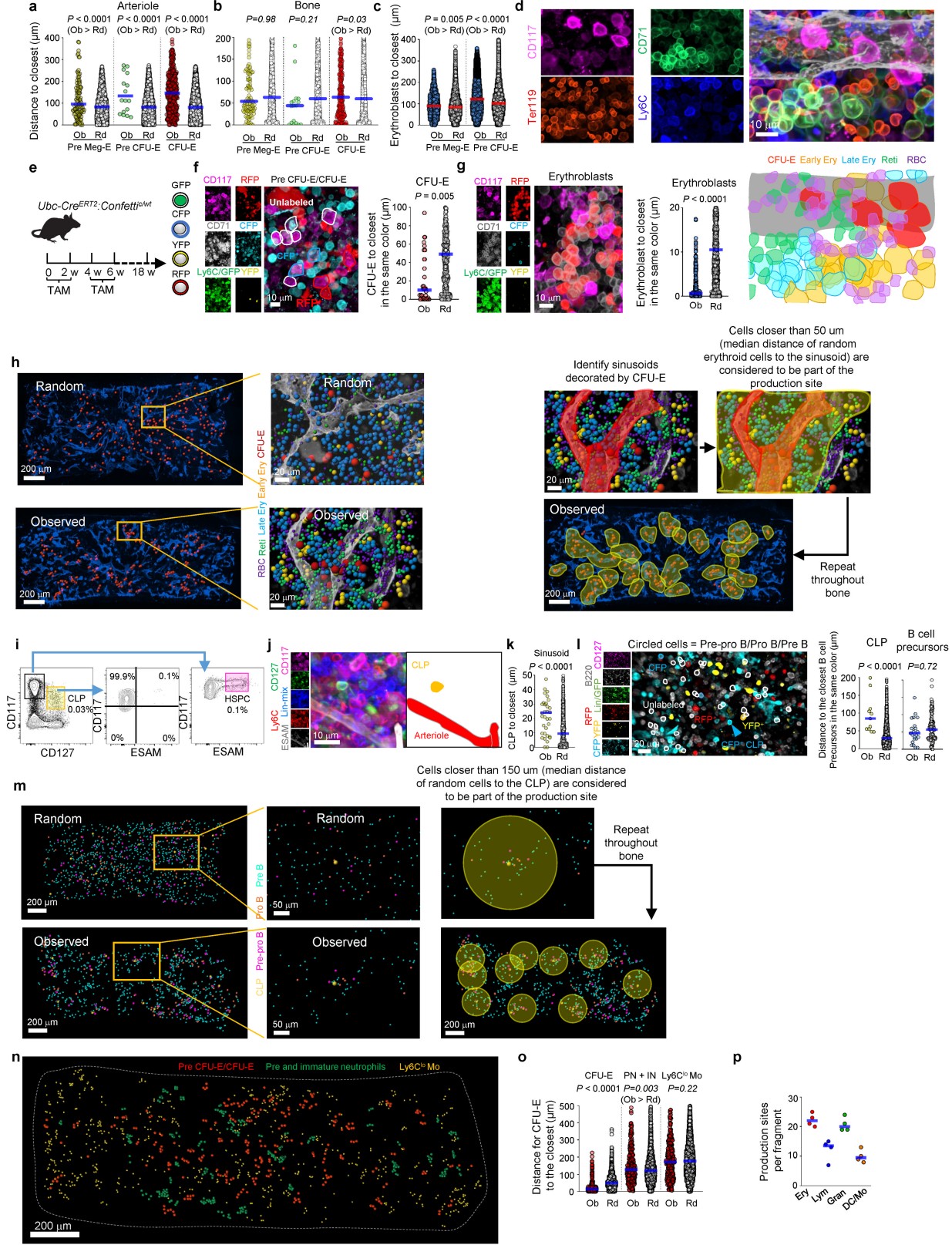

**Extended Data Fig. 5 |** See next page for caption.

**Extended Data Fig. 5 | Mapping erythropoiesis and B Lymphopoiesis in whole mounted sternum segments. a**,**b**, Distance analyses from each erythroid progenitor to the closest arteriole (a, n = 111 Pre Meg-E, 18 Pre CFU-E, and 523 CFU-E in 3 sternum segments from 3 mice) or bone (b, n = 109 Pre Meg-E, 18 Pre CFU-E, and 315 CFU-E in 3 sternum segments from 3 mice **c**, Distance analyses from erythroblasts (CD117⁻CD71⁺Ly6C⁻, containing early and late erythroblasts, n = 4461 in 3 sternum segments from 3 mice) to the closest indicated erythroid progenitor. **d**, High-power image showing the distribution of late erythroid cells around CFU-E. **e**, Scheme showing the pulse (with tamoxifen, TAM) and chase strategy to label hematopoietic cells in confetti mice. **f**, Representative images and distance analyses of confetti-labeled CFU-E to the closest CFU-E labeled in the same color (n = 38 confetti-labeled CFU-E in 5 sternum segments from 3 tamoxifen-treated confetti mice). **g**, Representative image showing confetti-labeled erythroblasts (CD117⁻CD71⁺Ly6C⁻, containing early and late erythroblasts) in tamoxifen-treated confetti mice. Due to a lack of available fluorescence channels for analyses, Ly6C and GFP were combined in a single dump channel. The histogram shows distance analyses of each confetti-labeled erythroblast to the closest erythroblast labeled in the same color (CD117⁻CD71⁺Ly6C⁻, containing early and late erythroblasts, n = 267 labeled erythroblasts in 4 sternum segments from tamoxifen-treated confetti mice). **h**, Step by step identification of erythroid production sites. In contrast to random cells, CFU-E cluster together and attach to sinusoids. Because adjacent CFU-E and erythroblasts are clonally related (see panels f, g) we consider that the center of the erythroid production site is a sinusoid decorated with 3 or more adjacent CFU-E. The mean distance of random cells to the closest CFU-E was 46.18 μm for early erythroblasts, 48.33 μm for late erythroblasts, 48.98 μm for reticulocytes, and 49.98 μm for erythrocytes (Fig. 3e). We thus considered all erythroid cells within 50 μm of the CFU-E decorated sinusoid as belonging to that production site. These analyses are then repeated through the bone to identify all production sites in the section. In the rare cases when the edges of two production sites overlap the erythroid cells were assigned to the closest of the two production sites. **i**, FACS gating strategy for simultaneous detection of CD117⁺CD127⁺Lin-mix⁻ CLP and CD117⁺ESAM⁺ HSPC (the Lin-mix panel contains B220, CD2, CD3e, CD5, CD8, CD11b, Ter119, Ly6G, CD41, Ly6C). **j**, **k** Representative image (j) and distance analyses (k) showing that CLP map near Ly6C⁺ arterioles and away from sinusoids (n = 36 CLP in 3 sternum segments from 3 mice). **l**, Representative image and distance analyses of confetti-labeled CLP or B cell precursors (B220⁺Lin⁻ cells, the Lin panel contains CD2, CD3e, CD5, CD8, CD11b, Ter119, Ly6G, IgM, and IgD, containing all Pre-pro B, Pro B, and Pre B cells) to the closest B cell precursors labeled in the same color (n = 11 confetti-labeled CLP, 23 confetti-labelled B cell precursors in 3 sternum segments from 3 tamoxifen-treated confetti mice). **m**, Identification of lymphoid production sites. In contrast to random cells CLP cluster together with Pre-pro B and Pro B cells. The mean distance for random cells to the closest CLP was 135.5 μm for Pre-pro B, 136.1 μm for Pro B, and 161.9 μm for Pre B (Fig. 3k). Based on this we defined each production site as the 150 μm region centered around each CLP. These analyses are then repeated through the bone to identify all production sites in the section. In the rare cases when the edges of two production sites overlap the B cells were assigned to the closest of the two production sites. **n**, Erythroid, neutrophil, and mono/DC production sites do not colocalize. The map shows the distribution of Pre CFU-E / CFU-E (labeling erythroid production sites), Pre- and immature neutrophils (labeling neutrophil production sites), and Ly6Cˡᵒ Monocytes (Ly6Cˡᵒ Mo, labeling dendritic cell/ Ly6Cˡᵒ monocyte production sites) in the sternum. Map dots are the average size of the relevant cells. **o**, Distance analyses from each CFU-E to the closest indicated cells (PN + IN: Pre and immature neutrophils; n = 297 CFU-E in 3 sternum segments from 3 mice). **p**, Quantification of each type of production site per sternum segment (n = 4 sternum segments from 4 mice). For all panels: statistical differences were calculated using two-tailed unpaired Student's t-test if the distributions were normal and two-tailed Mann-Whitney if not normal. P values are shown.

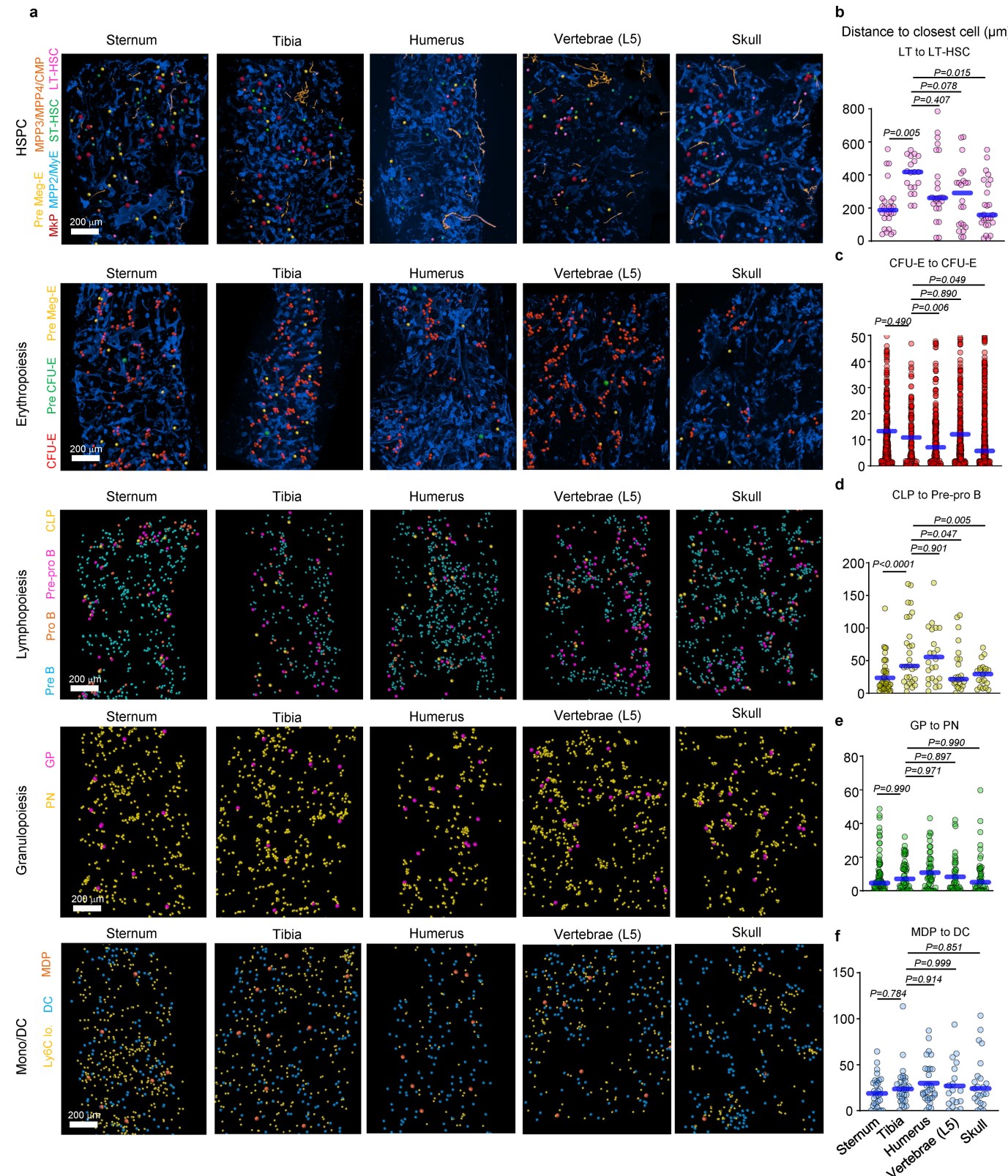

**Extended Data Fig. 6 | The anatomy of hematopoiesis in the steady-state is maintained across the skeleton. a**, Maps of the indicated HSPC in similar sized regions of the sternum, tibia, humerus, L5 vertebrae, and lamboid sutures of the skull. **b-f**, graphs showing the distance from each LT-HSC (b, n = 19, 21, 25, 23 and 26), CFU-E (c, n = 221, 501, 495,139 and 548), CLP (d, n = 28, 13, 50,15 and 22), GP (e, n = 79, 51, 51, 46 and 46) or MDP (f, n = 29, 29, 30, 19 and 23) to the closest indicated cell in the five bones examined. Note that the frequencies of LT-HSC are very different between bones leading to larger separations between cells (b). Statistical analyses were conducted using two-way ANOVA t-tests if the distributions were normal or Kruskal-Wallis test if not normal. P values are shown.

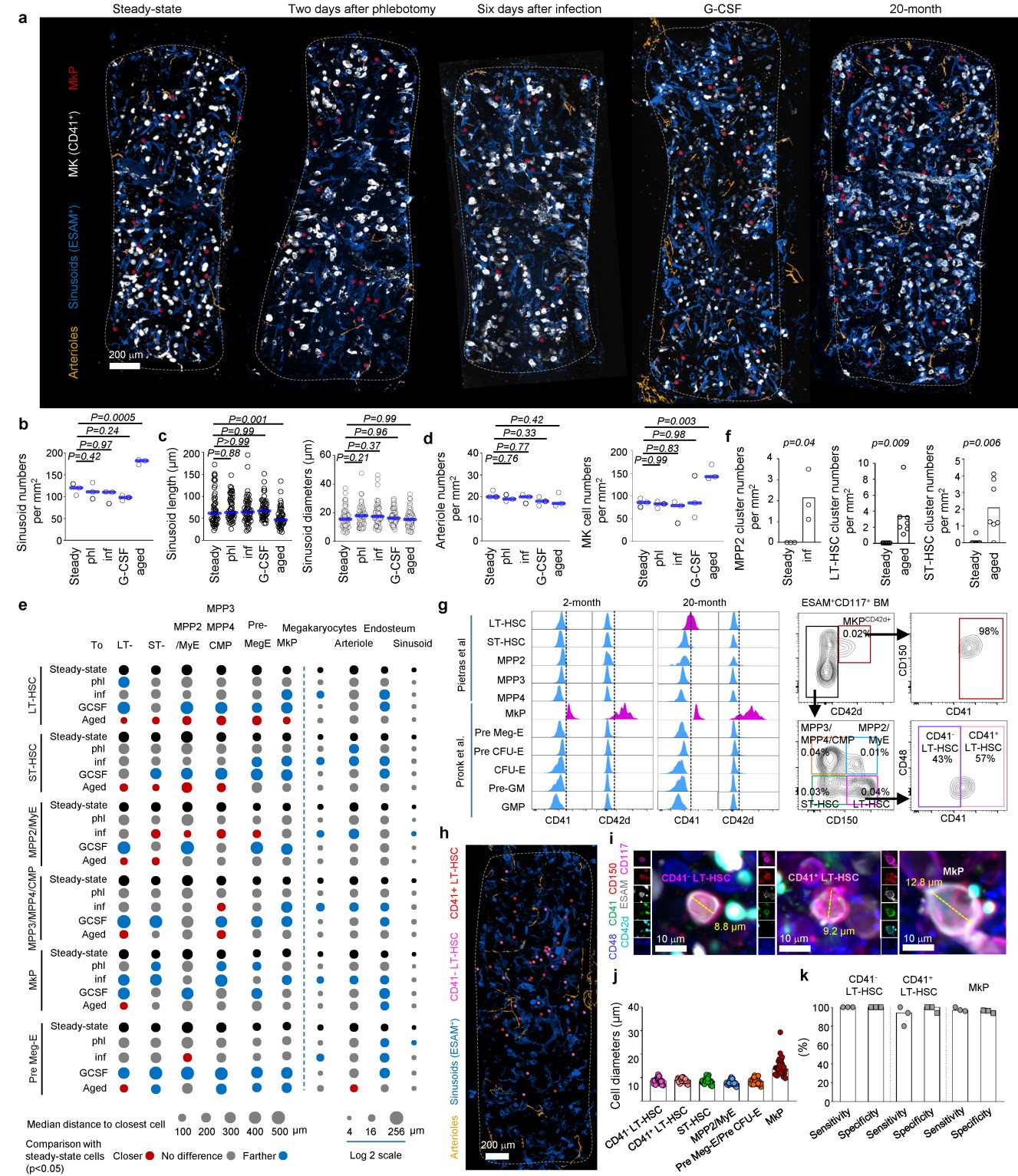

**Extended Data Fig. 7** | See next page for caption.

**Extended Data Fig. 7 | Effect of the different insults in the organization of the microenvironment and distribution of multipotent HSPC. a**, Maps showing the vascular organization of the bone at the indicated time points after insult. **b-d**, vessel numbers (b, n = 3 sternum segments from 3 mice for each group), sinusoid length and diameter (c, n = 50 randomly selected sinusoids from 3 sternum segments from 3 mice for each group); and arteriole and megakaryocyte numbers (d, n = 3 sternum segments from 3 mice for each group) in mice in the steady-stat (steady), 2 days after phlebotomy (phl), 6 days after *L. monocytogenes* (inf), four days of G-CSF (G-CSF), and 20 months of age (aged). **e**, Dot plot summarizing the median distance from each HSPC to all other indicated cells and structures at the indicated time points after insult. (n = 41 LT-HSC, 52 ST-HSC, 25 MPP2/MyE, 41 MPP3/MPP4/CMP, 61 MkP, and 82 Pre Meg-E in 4 sternum segments from four mice in steady-state; n = 21 LT-HSC, 35 ST-HSC, 16 MPP2/MyE, 30 MPP3/MPP4/CMP, 38 MkP, and 73 Pre Meg-E in 3 sternum segments from three mice two days after phlebotomy; n = 15 LT-HSC, 19 ST-HSC, 56 MPP2/MyE, 39 MPP3/MPP4/CMP, 17 MkP, and 57 Pre Meg-E in 3 sternum segments from three mice six days after infection; n = 10 LT-HSC, 16 ST-HSC, 11 MPP2/MyE, 23 MPP3/MPP4/CMP, 52 MkP, and 16 Pre Meg-E in 3 sternum segments from three mice four days of G-CSF treatment = 300 LT-HSC, 236 ST-HSC, 39 MPP2/MyE, 72 MPP3/MPP4/CMP, 133 MkP, and 57 Pre Meg-E in 3 sternum segments from three 20-month old mice. LT-HSC in aged mice contain CD41⁺ and CD41⁻ LT-HSC identified as shown in the next panels. Statistical differences were calculated using two-way ANOVA t-tests if the distributions were normal or Kruskal-Wallis test if not normal; P values are shown. **f**, Quantification of MPP2/Mye clusters six days after infection (n = 3 sternum segments from 3 mice in steady-state for control, n = 3 sternum segments from 3 mice six days after infection) and LT-HSC and ST-HSC clusters in aging (n = 7 sternum segments from 4 mice in steady-state for control, n = 7 sternum segments from 4 20-month old mice). Statistical differences were calculated using two-tailed unpaired Student's t-test if the distributions were normal and two-tailed Mann-Whitney if not normal. P values are shown. **g,h**, Developing strategies to image CD41⁺ and CD41⁻ LT-HSC in aged mice. A limitation of our imaging of multipotent hematopoiesis is that we used CD41 to distinguish MkP from other ESAM⁺ HSPC (Fig. 1e). However, aging causes accumulation of myeloid-biased CD41 LT-HSC. This prevents using solely CD41 expression to discriminate between CD41⁺ MkP and CD41⁺ LT-HSC in aged mice. CD42d is exclusively expressed in MkP but not HSC (doi:10.1016/j.stem.2019.06.007) and can be used together with CD41 to distinguish MkP from CD41⁺ LT-HSC in aged mice by FACS (g) and imaging (h). **i,j**, To simultaneously image CD41 positive and negative LT-HSC with other multipotent HSPC we used cell size. The panels show representative images (i) and quantification of cell diameters (j) demonstrating that ESAM⁺CD117⁺CD48⁻CD150⁺CD42⁻CD41⁺ myeloid-biased LT-HSC, ESAM⁺CD117⁺CD48⁻CD150⁺CD42⁻CD41⁻ lymphoid-biased LT-HSC and ESAM⁺CD117⁺CD150⁺CD42⁺CD41⁺ MkP can be distinguished based on cell size (n = 30 randomly selected cells from each type in 3 sternum segments from three 20-month-old mice). **k**, Sensitivity (= Number of ESAM⁺CD117⁺CD48⁻CD150⁺CD41⁻ LT-HSC, ESAM⁺CD117⁺CD48⁻CD150⁺CD41⁺ LT-HSC, ESAM⁺CD117⁺CD150⁺CD41⁺ MkP correctly identified based on CD150 and CD41 expression and cell size, divided by the number of cells identified when counterstained with CD42 (CD41⁻CD42⁻ LT-HSC, CD41⁺CD42⁻ LT HSC, and CD41⁺CD42⁺ MkP) and specificity (= Number of ESAM⁺CD117⁺CD48⁻CD150⁺CD41⁻ LT-HSC, ESAM⁺CD117⁺CD48⁻CD150⁺CD41⁺ LT-HSC, ESAM⁺CD117⁺CD150⁺CD41⁺ MkP correctly identified based on CD150 and CD41 expression and cell size, divided by the number of cells identified based on CD150 and CD41 expression and cell size) for distinguishing myeloid and lymphoid biased LT-HSC and MkP based on CD41 and CD150 expression and cell size (compared with CD42d based identification, 47 CD41⁺ LT-HSC, 55 CD41⁻ LT-HSC, and 70 MkP in n = 3 sternum segments from two 20-month-old mice were analyzed).

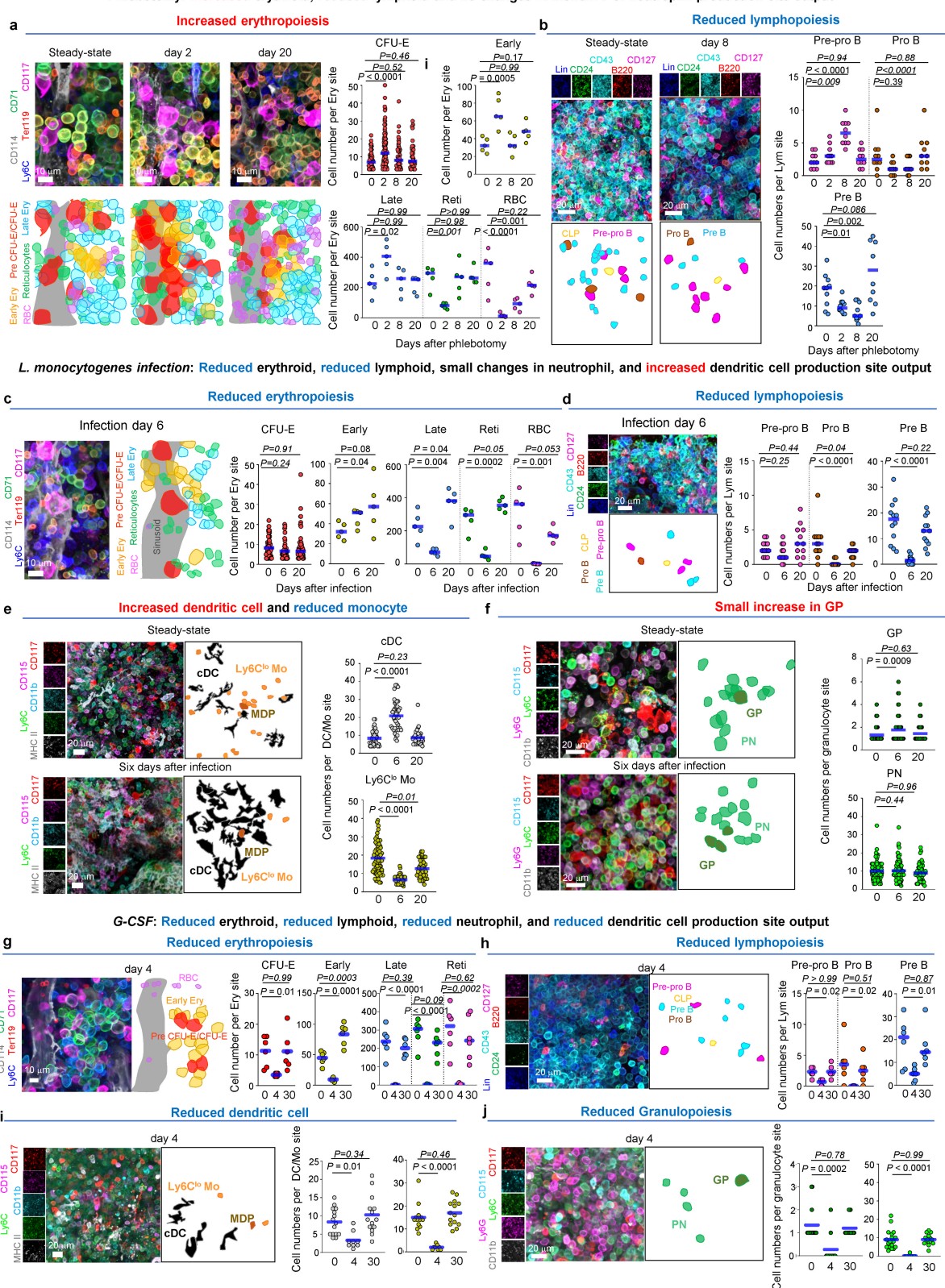

**Extended Data Fig. 8** | See next page for caption.

**Extended Data Fig. 8 | Changes in the microarchitecture of the production sites in response to acute hematopoietic stress. a**, After phlebotomy the erythroid production sites respond with increases in CFU-E, and early and late erythroblasts, and reductions in reticulocytes and red blood cells. The images show the changes in cell content in representative erythroid production sites at the indicated time points after phlebotomy. The dot plots show the number of indicated cells per production site. CFU-E (n = 65, 99, 58, and 40 production sites for days 0, 2, 8, and 20, 3 sternum segments from 3 mice per time point) and indicated erythroid cells (n = 5 randomly selected production sites) in erythroid production sites after phlebotomy. **b**, Representative images and number of cells per B cell site at the indicated time points after phlebotomy (n = 10 randomly selected production sites in 3 sternum segments from 3 mice for each time point). **c**, After infection, erythroid production sites show reduced output that is progressively restored. The image shows a representative erythroid production site after infection. The dot plots show the number of CFU-E (n = 65, 45, and 60 production sites for days 0, 6, 20, three sternum segments from 3 mice per time point) and indicated erythroid cells (n = 5 randomly selected production sites for each time point, in 3 sternum segments from 3 mice) from control or *L. monocytogenes*-infected mice. **d**, Representative image and number of cells per B cell production site from control or *L. monocytogenes*-infected mice after infection (n = 12 randomly selected production sites in 3 sternum segments from 3 mice for each time point). **e**, Representative images and number of cells showing increased dendritic cell and reduced monocyte production in DC/Ly6C$^{lo}$ sites after *L. monocytogenes* infection mice (n = 49, 35, and 29 randomly selected production sites for days 0, 6, 20, in 3–5 sternum segments from 3 mice per time point). **f**, as (e) but for granulocyte production sites (n = 96, 88, and 50 randomly selected production sites in 3–5 sternum segments from 3-4 mice per time point). **g-j**, representative images and number of cells per production site demonstrating that G-CSF treatment leads to reductions in the output of erythroid (g, n = 6, 6, 6), lymphoid (h, n = 6, 6, 6), mono/DC (i, n = 15, 9, 14), and neutrophil (j, n = 15, 11, 15) production sites followed by a return to homeostasis at day 30. For all panels: statistical differences were calculated using two way ANOVA t-tests if the distributions were normal or Kruskal-Wallis test if not normal; P values are shown.

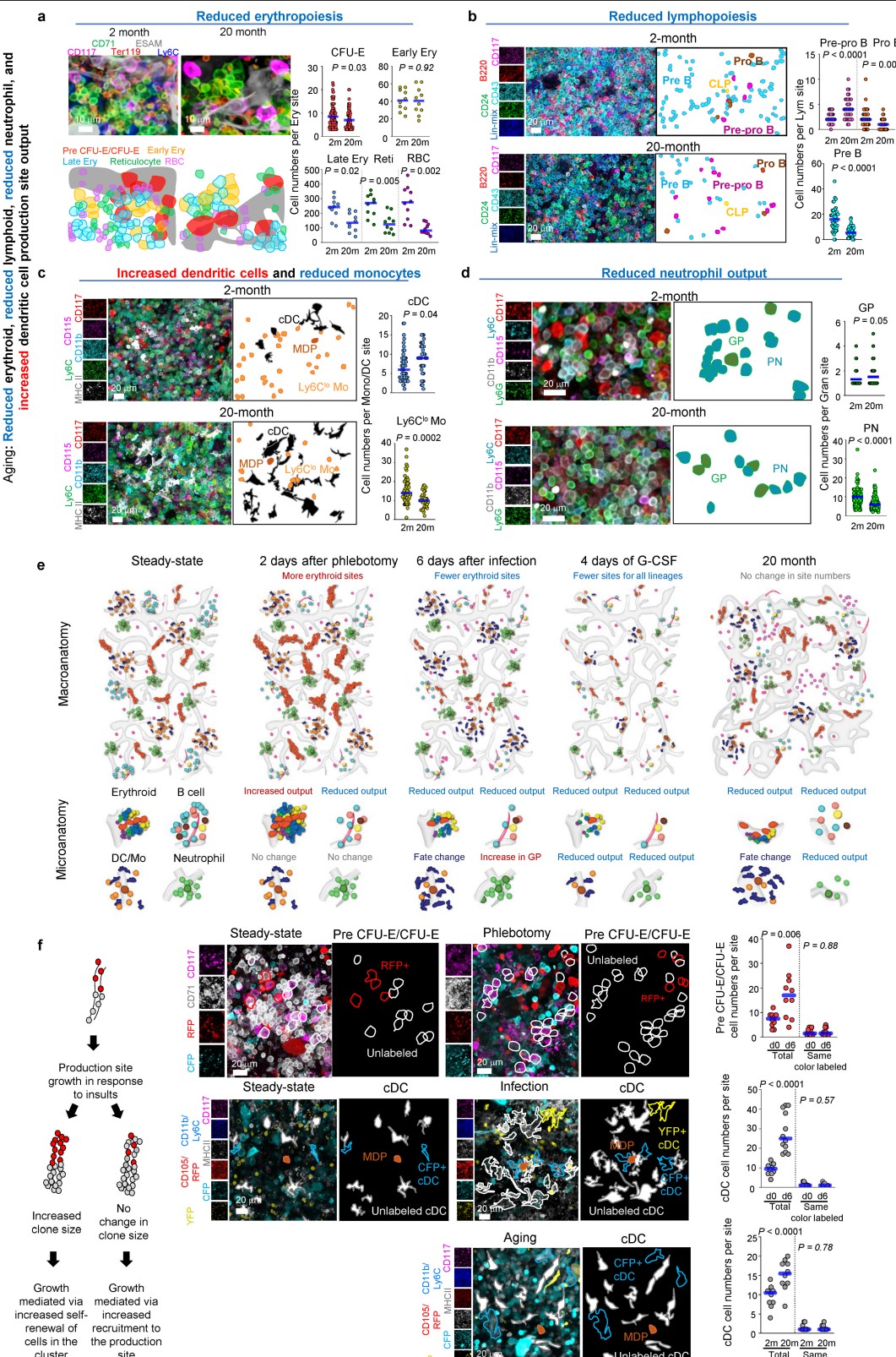

**Extended Data Fig. 9** | See next page for caption.

**Extended Data Fig. 9 | Plasticity of production sites. a**, Representative images showing the changes in erythroid production sites in the aged sternum. The dot plots show the number of CFU-E (n = 96 and 88 production sites in 3 sternum segments from 3 mice) and indicated erythroid cells (n = 10 randomly selected production sites) in erythroid production sites in 2- and 20-month-old mice. **b**, Representative images showing the changes in lymphoid production sites in the aged sternum. The dot plots show the number of cells per B cell production site in 2- (n = 41) and 20-month-old mice (n = 38 randomly selected B cell sites in 3 sternum segments from 3 mice). **c**, Representative images showing the changes in mono/DC production sites in the aged sternum. The dot plots show the number of cells per mono/DC site in 2- (n = 49) or 20-month-old (n = 34 DC/Ly6C$^{lo}$ monocytes production sites in 3–5 segments from 3 mice). **d**, Representative image images showing the changes in neutrophil production sites in the aged sternum. The dot plots show the number of GP and preneutrophils (PN) cells per granulocyte production line in 2- (n = 96) and 20-month-old (n = 113 neutrophil production sites in 4-5 segments from 4 mice). **e**, Schemes illustrating the macroanatomical (production site numbers and gross changes in size) and microanatomical (detailed architecture of the production site) changes in response to insults. **f**, To examine how production sites grow in response to stress we used confetti mice. If the site grows via increased self-renewal, then the number of cells labeled by the same fluorescent protein should increase. Alternatively, if the site grows via increased cell recruitment to the site, then the number of confetti-labeled cells in the cluster will remain constant. The top images and dot plots show the changes in total and confetti-labeled cells in each erythroid production site in the steady-state or 2 days after phlebotomy-driven expansion. The bottom images and dot plots show the changes in total and confetti-labeled dendritic cells in each mono/DC production site 6 days after *L. monocytogenes* infection or 20 months of age. n = 10 for erythroid production sites, n = 10 for mono/DC sites in response to infection, and n = 10 for mono/DC sites in 20-month-old mice. For all panels: statistical differences were calculated using two-tailed unpaired Student's t-test if the distributions were normal and two-tailed Mann-Whitney if not normal. P values are shown.

# Reporting Summary

## Statistics

For all statistical analyses, confirm that the following items are present in the figure legend, table legend, main text, or Methods section.

| n/a | Confirmed | |
|---|---|---|
| ☐ | ☒ | The exact sample size (*n*) for each experimental group/condition, given as a discrete number and unit of measurement |
| ☐ | ☒ | A statement on whether measurements were taken from distinct samples or whether the same sample was measured repeatedly |
| ☐ | ☒ | The statistical test(s) used AND whether they are one- or two-sided<br>*Only common tests should be described solely by name; describe more complex techniques in the Methods section.* |
| ☒ | ☐ | A description of all covariates tested |
| ☒ | ☐ | A description of any assumptions or corrections, such as tests of normality and adjustment for multiple comparisons |
| ☐ | ☒ | A full description of the statistical parameters including central tendency (e.g. means) or other basic estimates (e.g. regression coefficient) AND variation (e.g. standard deviation) or associated estimates of uncertainty (e.g. confidence intervals) |
| ☐ | ☒ | For null hypothesis testing, the test statistic (e.g. *F*, *t*, *r*) with confidence intervals, effect sizes, degrees of freedom and *P* value noted<br>*Give P values as exact values whenever suitable.* |
| ☒ | ☐ | For Bayesian analysis, information on the choice of priors and Markov chain Monte Carlo settings |
| ☒ | ☐ | For hierarchical and complex designs, identification of the appropriate level for tests and full reporting of outcomes |
| ☒ | ☐ | Estimates of effect sizes (e.g. Cohen's *d*, Pearson's *r*), indicating how they were calculated |

*Our web collection on statistics for biologists contains articles on many of the points above.*

## Software and code

Policy information about availability of computer code

| Data collection | NIS-Elements software (Nikon, version 5.20.02 and 5.30.03).    BD FACSDiva software (BD, version 9.0). |
|---|---|
| Data analysis | NIS-Elements software (Nikon, version 5.20.02 and 5.30.03), Imaris software (Bitplane, version 9.5 to 9.9), and Matlab software (MathWorks, version 2018a)   GraphPad Prism software (Graphpad, version 9). FlowJo (Tree Star, version 10). |

For manuscripts utilizing custom algorithms or software that are central to the research but not yet described in published literature, software must be made available to editors and reviewers. We strongly encourage code deposition in a community repository (e.g. GitHub). See the Nature Portfolio guidelines for submitting code & software for further information.

## Data

Policy information about availability of data

All manuscripts must include a data availability statement. This statement should provide the following information, where applicable:
- Accession codes, unique identifiers, or web links for publicly available datasets
- A description of any restrictions on data availability
- For clinical datasets or third party data, please ensure that the statement adheres to our policy

Source Data for quantifications described in the text or shown in graphs plotted in Figures 1-5 and Extended Data Figures 1-10 are available with the manuscript.
Datasets for all flow analysis and images shown in Figures 1-5 and Extended Data Figures 1-10 are publicly available: "raw image data for "Resilient anatomy and local microplasticity of naïve and stress hematopoiesis "", Mendeley Data, V2, doi: 10.17632/27wvzpyf5h.2

# Human research participants

Policy information about <u>studies involving human research participants and Sex and Gender in Research.</u>

| | |
|---|---|
| Reporting on sex and gender | N/A |
| Population characteristics | N/A |
| Recruitment | N/A |
| Ethics oversight | N/A |

Note that full information on the approval of the study protocol must also be provided in the manuscript.

# Field-specific reporting

Please select the one below that is the best fit for your research. If you are not sure, read the appropriate sections before making your selection.

☒ Life sciences ☐ Behavioural & social sciences ☐ Ecological, evolutionary & environmental sciences

For a reference copy of the document with all sections, see <u>nature.com/documents/nr-reporting-summary-flat.pdf</u>

# Life sciences study design

All studies must disclose on these points even when the disclosure is negative.

| | |
|---|---|
| Sample size | Because analyses of large bone marrow samples is a very time consuming process (1-6 hours per sample) it is not possible to examine large numbers of samples. We have previously shown that 3 bones per condition allow identification of sufficient numbers of cells to detect changes in location and distribution in the bone marrow (doi:10.1038/s41586-021-03201-2 (2021)). Based on this we have strived to analyze 3 bones per condition. At the request of the referees, we have increased the number of samples for some experiments (e.g. Figure 5). |
| Data exclusions | all mice were included in the analyses. no data were excluded from the analyses |
| Replication | Results are representative of at least three independent experiments. All attempts at replication were successful. |
| Randomization | Mice were randomly allocated to the different groups based on the cage, genotype, and litter size. |
| Blinding | In this manuscript we have imaged bones from WT (or reporter mice) subjected to different insults and stained with different antibodies. It was not possible to blind the investigator to the type of bone examined as these are readily identified by shape. Similarly, the insults used (hemorrhage, G-CSF, L. monocytogenes infection, and aging) generate such evident changes in cellular content in the bone marrow (hemorrhage, G-CSF, L. monocytogenes infection) or shape of the bone (aged mice, bones become larger) that it was not possible to blind the investigator to the type of insult examined. |

# Reporting for specific materials, systems and methods

We require information from authors about some types of materials, experimental systems and methods used in many studies. Here, indicate whether each material, system or method listed is relevant to your study. If you are not sure if a list item applies to your research, read the appropriate section before selecting a response.

## Materials & experimental systems

| n/a | Involved in the study |
|---|---|
| ☐ | ☒ Antibodies |
| ☒ | ☐ Eukaryotic cell lines |
| ☒ | ☐ Palaeontology and archaeology |
| ☐ | ☒ Animals and other organisms |
| ☒ | ☐ Clinical data |
| ☒ | ☐ Dual use research of concern |

## Methods

| n/a | Involved in the study |
|---|---|
| ☒ | ☐ ChIP-seq |
| ☐ | ☒ Flow cytometry |
| ☒ | ☐ MRI-based neuroimaging |

# Antibodies

| | |
|---|---|
| Antibodies used | B220-APCCy7, BioLegend, catalog number 103224; B220-AF488, BioLegend, catalog number 103225; CD3-biotin labeled, BioLegend, catalog number 100304; CD3-PE, BioLegend, catalog number 100307; CD3-AF488, BioLegend, catalog number 100321; CD3-AF647, BioLegend, catalog number 100322; CD8-biotin labeled, BioLegend, catalog number 100704; CD8-PECy7, BioLegend, catalog number 100722; CD11b-biotin labeled, BioLegend, catalog number 101204; CD11b-PECy7, BioLegend, catalog number 101216; CD11b-AF488, BioLegend, catalog number 101217; CD11b-AF647, BioLegend, catalog number 101218; CD11c-FITC, BioLegend, catalog number 117305; CD11c-PE, BioLegend, catalog number 117307; CD16/32-PE, BioLegend, catalog number 101308; CD16/32-APCCy7, BioLegend, catalog number 101328; CD24-PE, BioLegend, catalog number 138503; CD24-BV421, BioLegend, catalog number 101825; |

CD31-PE, BioLegend, catalog number 102507; CD31-AF647, BioLegend, catalog number 102516; CD34-FITC, Thermo Fisher Scientific, catalog number 11-0341-85; CD34-efluor 660, Thermo Fisher Scientific, catalog number 50-0341-82; CD41-FITC, BioLegend, catalog number 133904; CD41-BV605, BioLegend, catalog number 133921; CD41-biotin labeled, BioLegend, catalog number 133930; CD42d-F488, Invitrogen, catalog number 53-0421-82; CD42d-APC, BioLegend, catalog number 148506; CD43-PE, BioLegend, catalog number 143209; CD45-PE, BioLegend, catalog number 103106; CD45.1-PE, BioLegend, catalog number 110708; CD45.1-APC, BioLegend, catalog number 110714; CD45.1-AF488, BioLegend, catalog number 110718; CD45.2-FITC, BioLegend, catalog number 109806; CD45.2-APC, BioLegend, catalog number 109814; CD45.2-AF700, BioLegend, catalog number 109822; CD48-AF488, BioLegend, catalog number 103414; CD48-AF647, BioLegend, catalog number 103416; CD71-AF647, BD Biosciences, catalog number 563504; CD71-PE, BioLegend, catalog number 113807; Biotin anti-mouse CD71, BioLegend, catalog number 113803; CD105-PECy7, BioLegend, catalog number 120410; CD115-PE, BioLegend, catalog number 135506; CD115-AF488, BioLegend, catalog number 135512; CD115-BV421, BioLegend, catalog number 135513; CD117-AF488, BioLegend, catalog number 105816; CD117-APCCy7, BioLegend, catalog number 105826; CD117-BV421, BioLegend, catalog number 105827; CD117-PECF594, BioLegend, catalog number 105834; CD117-PECy7, Thermo Fisher Scientific, catalog number 25-1171-81; CD117-BV480, BD, catalog number 566074; CD127-PE, BioLegend, catalog number 135009; CD127-Biotin, Biolegend, catalog number 135005; CD127-BV785, BioLegend, catalog number 135037; CD127-AF647, BioLegend, catalog number 135020; CD135-APC, BioLegend, catalog number 135310; CD144-AF647, BioLegend, catalog number 138108; CD150-PE, BioLegend, catalog number 115904; CD150-AF488, BioLegend, catalog number 115916; CD150-BV421, BioLegend, catalog number 115925; ESAM-PE, BioLegend, catalog number 136204; ESAM-FITC, Biolegend, catalog number 136205; IgM-AF488, BioLegend, catalog number406522;IgD-AF700, BioLegend, catalog number 405729; IgD-AF488, BioLegend 405718; IgM-APC, BioLegend, catalog number 406525; MHC II-biotin labeled, BioLegend, catalog number 107603; MHC II-AF488, BioLegend, catalog number 107616; MHC II-AF647, BioLegend, catalog number 107618; Ly6C-biotin labeled, BioLegend, catalog number 128004; Ly6C-AF647, BioLegend, catalog number 128010; Ly6C-AF488, BioLegend, catalog number 128022; Ly6C-PerCP, BioLegend, catalog number 128028; Ly6G-biotin labeled, BioLegend, catalog number 127604; Ly6G-PerCPCy5.5, BioLegend, catalog number 127616; Ly6G-PECy7, BioLegend, catalog number 127617; Ly6G-AF488, BioLegend, catalog number 127626; Sca1-FITC, BioLegend, catalog number 108106; Sca1-PECy7, BioLegend, catalog number 108114; Ter119-biotin labeled, BioLegend, catalog number 116204; Ter119-AF488, BioLegend, catalog number 116215; Ter119-AF647, BioLegend, catalog number 116218; Ter119-AF700, BioLegend, catalog number 116220; Gr1-biotin labeled, Biolegend, catalog number 108403;

| Validation | Antibodies were validated by comparing their staining pattern on mouse BM cells by FACS. For imaging experiments each antibody was validated by testing that the frequency of cells stained matched the one obtained in FACS analyses. |

# Animals and other research organisms

Policy information about studies involving animals; ARRIVE guidelines recommended for reporting animal research, and Sex and Gender in Research

| Laboratory animals | C57BL/6J-Ptprcb (CD45.2+) mice; B6.SJL-PtprcaPepcb/BoyJ (CD45.1+) mice, JAX: 002014; B6.Cg-Ndor1Tg(UBC-cre/ERT2)1Ejb/1J, JAX: 007001;B6.129P2-Gt(ROSA)26Sortm1(CAG-Brainbow2.1)Cle (R26R-Confetti). JAX:017492. Both male and female at 8-14 weeks old for young age mice and 80-100 weeks old for old age mice were used in our experiments. |
| Wild animals | No wild animals were involved in our study. |
| Reporting on sex | Both male and female mice were used in our experiments. |
| Field-collected sample | No field collected samples were used in the study |
| Ethics oversight | All mouse experiments –except live mouse imaging experiments- were approved by the Institutional Animal Care Committee of Cincinnati Children's Hospital Medical Center. Live mouse imaging experiments were performed in compliance with institutional guidelines and approved by the Subcommittee on Research Animal Care (SRAC) at Massachusetts General Hospital. The following mouse strains were used: C57BL/6J-Ptprcb (CD45.2), B6.SJL-PtprcaPepcb/BoyJ (CD45.1), B6.Cg-Ndor1Tg (UBC-cre/ERT2)1Ejb/1J (Ubc-creERT2), C57BL/6-Tg(CAG-EGFP)131Osb/LeySopJ (Actin-GFP) and B6.129P2-Gt(ROSA)26Sortm1(CAG-Brainbow2.1)Cle (R26R-Confetti). R 26R-Confetti mice were crossed with Ubc-creERT2 mice to generate Ubc-creERT2:Confetti mice. All mice were maintained on a C57BL/6J background. Eight to twelve (2-month-old) and eighty to a hundred weeks (20-month-old) male and female mice were used. All mice were bred and aged in our vivarium or purchased from the Jackson Laboratory. Mice were maintained at the vivarium at Cincinnati Children's Hospital Medical Center under a 14-hours light:10-hours darkness schedule, 30–70% humidity, 22.2 ± 1.1 °C, and specific-pathogen-free conditions. |

Note that full information on the approval of the study protocol must also be provided in the manuscript.

# Flow Cytometry

## Plots

Confirm that:

☒ The axis labels state the marker and fluorochrome used (e.g. CD4-FITC).

☒ The axis scales are clearly visible. Include numbers along axes only for bottom left plot of group (a 'group' is an analysis of identical markers).

☒ All plots are contour plots with outliers or pseudocolor plots.

☒ A numerical value for number of cells or percentage (with statistics) is provided.

## Methodology

| Sample preparation | Mice were euthanized by isoflurane inhalation followed by cervical dislocation. Bone marrow cells were harvested by flushing bones with 1 ml of ice-cold PEB buffer (2 mM EDTA and 0.5% bovine serum albumin in PBS). Blood was collected from the retro-orbital venous sinus in tubes containing EDTA. Red blood cells in peripheral blood were lysed by the addition of 1 ml of RBC lysis buffer (150 mM NH4Cl, 10 mM NaCO3 and 0.1 mM EDTA). Cells were immediately decanted by centrifugation, resuspended in ice-cold PEB. Cells were stained under dark for 30 min in PEB buffer containing antibodies, washed thrice |

| | with ice cold PBS. |
|---|---|
| Instrument | Cells were stained in the dark for 30 minutes in ice-cold PEB buffer containing antibodies, washed thrice with ice-cold PEB, and analyzed in an LSRFortessa™ Flow Cytometer (BD Biosciences), LSR II Flow Cytometer (BD Biosciences), or FACS-purified in a FACSAria™ II Cell Sorter (BD Biosciences) or an SH800S Cell Sorter (Sony Biotechnology). |
| Software | FACSDiva software (BD Biosciences) for data collection and FlowJo (Tree Star) for data analysis. |
| Cell population abundance | Freshly sorted cells were examined by the same FACS sorter again and the purity is >95%. |
| Gating strategy | In all experiments, debris were excluded by using Forward scatter/Side scatter (FSC/SSC). Doublets were excluded by double forward (FSC-A and FSC-W), and side scatter (SSC-A and SSC-H). Dead cells were excluded as DAPI+ cells. BM LT-HSC, ST-HSC, MPP2, MPP3, MPP4, MkP, Pre Meg-E, Pre CFU-E, CFU-E, Pre GM, GMP, GP, MoP, MDP, CMP were gated as previously described[1, 2, 3]. Briefly, BM LT-HSC are Lineage-CD117+Sca1+CD135-CD150+CD48-,BM ST-HSC are Lineage-CD117+Sca1+CD135-CD150-CD48-, BM MPP2 are Lineage-CD117+Sca1+CD135-CD150+CD48+, BM MPP3 are Lineage-CD117+Sca1+CD135-CD150-CD48+,BM MPP4 are Lineage-CD117+Sca1+CD135+. Both in the BM and peripheral blood, B cells and T cells were gated as B220+ or CD3+, respectively. Neutrophils were gated as Ly6G+ cells in the peripheral blood. Gating strategies for other cells were detaily described in the manuscript and figures.<br>1, Yanez, A. et al. Granulocyte-Monocyte Progenitors and Monocyte-Dendritic Cell Progenitors Independently Produce Functionally Distinct Monocytes. Immunity 47, 890-902 e894, doi:10.1016/j.immuni.2017.10.021 (2017).<br>2, Pronk, C.J., et al., Elucidation of the phenotypic, functional, and molecular topography of a myeloerythroid progenitor cell hierarchy. Cell Stem Cell, 2007. 1(4): p. 428-42.<br>3, Pietras, E. M. et al. Functionally Distinct Subsets of Lineage-Biased Multipotent Progenitors Control Blood Production in Normal and Regenerative Conditions. Cell Stem Cell 17, 35-46, doi:10.1016/j.stem.2015.05.003 (2015). |

☒ Tick this box to confirm that a figure exemplifying the gating strategy is provided in the Supplementary Information.

