## [Peer Review File · Nature]

Manuscript Title: Resilient anatomy and local plasticity of naïve and stress hematopoiesis

Reviewer Comments & Author Rebuttals

Reviewer Reports on the Initial Version:

Referees' comments:

Referee #1 (Remarks to the Author):

The authors identify a set of 35 markers that can be labelled by antibodies and allows the unequivocal distinction of all relevant types of hematopoietic stem and progenitor cells by multicolor imaging. After proving the validity of their marker set with functional assays using transplantation models they set out to characterize the base-line, stress- and aging-induced spatial organization of hematopoiesis in the murine bone marrow on the level of individual cells. The principal question and the used approach are both highly original and provide an exciting and extremely interesting view into the mechanics of hematopoiesis in situ. As such they are highly relevant for the field of hematopoiesis and beyond. However, I have identified several critical issues that need to be addressed before the study provides a fully convincing case for Nature.

Specific points (in chronological order from introduction to discussion) are:

- 1) Ref. 10: intravital imaging of HSC motility has been achieved considerably earlier than in author's ref 10. Please also refer to [10.1182/blood-2008-12-195644](https://doi.org/10.1182/blood-2008-12-195644)
- 2) Fig. 1A: the indicated subtypes of stem cells are not fully reproduced in the surface marker analysis in Fig. 1b and in Fig. 1c. CMP is missing and MP+cMoP are not noted in Fig. 1a. This is confusing and unclear. Further, I could not find a structured list of all 247 antibodies tested. This should be provided as a supplementary table.
- 3) The fact that ESAM is a HSC marker is known since at least 2009. What is the novel discovery made in Fig. 1c? This must be explained in relation to existing literature.
- 4) Fig. 1d: the experimental design is very hard to impossible to understand from the figure. It implies that each recipient received ESAM+ and ESAM- cells at the same time, which was not the case. So, I suggest to add an "or" between ESAM+ and ESAM- in the scheme. Then I find it hard to conceive that MPPs have different performance in repopulation, if almost 6 times fewer ESAM- cells were transferred in comparison to ESAM+. This number must be identical. For MPP4 the ratio changes in favor of ESAM- cells. Also this makes it difficult to compare the performance of the cells. The legend is also unclear: was a total of e.g. 5 mice analyzed in the MPP2 experiment in 3 different runs, or were 3 different runs performed each with 5 mice per group? Finally, the labelling of the y-axis in 1D should also contain a "%".
- 5) Fig. 1f as final result of all work in 1a-e is very nice. Please make sure the margins of all images align at the top and bottom.
- 6) Fig. 2A and related: all bone marrow imaging was done in the sternum, which is a flat bone. However, transplantation and CFU assays were all performed with cells sorted from long bones. Since the blood vessel system of long bones is different from flat bones (see e.g. [10.1038/s42255-018-0016-5](https://doi.org/10.1038/s42255-018-0016-5)), it is not clear, whether both bone marrow sites behave the same and hence stem cells from both sites behave the same. Therefore, the analysis of HSC and MPP must also be performed in long bones as has been done for different questions before (e.g. [10.1038/nature15250](https://doi.org/10.1038/nature15250)).
- 7) Mentioning the distance of LT-HSC to each other as $>100 \mu\text{m}$: for non-expert readers authors should add a comparison to the size of the cells. From their images in fig. 1f this should be $\sim 8 \mu\text{m}$. So, $100 \mu\text{m}$ equal to 12-13 cell diameters.
- 8) Furthermore, it is essential to provide a 3-D analysis of the spatial arrangement of HSC in the marrow. Currently the results suggest that all cells are in one single plane, which they are

obviously not. It is hence required to perform a volumetric analysis and consider all neighboring cells in a volume around one cell. Imaging just the top 35 μm of sternum is not sufficient to reach this goal, if distances between single HSC can be as big as 100 μm and more. Issues with poor antibody penetration can be solved, e.g. by using flushed marrow plugs (less preferred due to corruption of trans cortical blood vessels) or e.g. cut bones as in 10.1038/nature15250 or [https://www.cell.com/immunity/pdfExtended/S1074-7613\(18\)30038-4](https://www.cell.com/immunity/pdfExtended/S1074-7613(18)30038-4).

9) Fig. 2c and related supplemental figures: the labelling of the lower part of the graph is difficult to read if upside down. Please flip.

10) Fig. 2e: are the shown CFU-E inside, above or below of the blood vessel? This is not clear without a 3-D analysis. Authors should also analyze in long bones specifically, how these cell types and the other investigated stem/progenitor cells localize with respect to the recently described transcortical blood vessels (10.1038/s42255-018-0016-5), which are known to be the main routes for bone entry/exit of blood.

11) Suppl. Fig. 4k/l: the amount of Ly6G label in the marrow appears to be extremely low. Neutrophil contents in murine marrow are several millions per long bone (see e.g. <https://www.ncbi.nlm.nih.gov/pmc/articles/PMC3732092/>) and hence one should see many Ly6G+ cells. Where does this incompatibility come from? Or is it a specificity of sternal bone marrow?

12) The scheme in Fig. 2o suggests that CFU-E and MDP are within blood vessels. Is this intended? From the data and logics, it appears they should be close to but not inside of sinusoids.

13) Fig. 3e: the studies on stress hematopoiesis are very interesting. It is, however, very unexpected, that neutrophil production does not seem to sense these changes. Hence, I am missing clinically relevant models, that mimic gross neutrophil changes in the bone marrow. Mobilization by G-CSF/AMD3100 can mimic cytokine/drug-induced neutrophil/stem cell mobilization and bone marrow reconstitution after lethal irradiation mimics clinical stem cell transplantation. These models are all very close to real-world clinical situations and should thus be investigated with the tools established in the study.

14) I very much appreciate the graphical summary in Fig. 4l. I would suggest to also make some kind of animation of these main features and provide this as additional supplemental movie, since the subject is very complex and not easy to grasp for non-experts.

15) Discussion: I consider the statement, that the bone marrow "was the last major tissue, where the spatial relationships between progenitors and differentiating cells – the anatomy- remained uncertain" a bit far-fetched. I am certain we do not have comparably detailed information on many other tissues. Hence, I suggest to tone down the wording here.

16) Authors state in discussion, that "stem cells migrate through bone marrow". It has been shown by intravital imaging long ago (see e.g. 10.1182/blood-2008-12-195644) that they actually hardly move but instead are mainly sessile. Things change dramatically after stress mobilization which is actually the reason for my critique above regarding the investigations on stress hematopoiesis. Please adapt the wording.

17) I cannot find references in the text to Refs. 30 and 31.

Referee #2 (Remarks to the Author):

Here, Wu et al develop new 4-6 color labeling scheme for hematopoietic progenitors that can be used with confocal microscopy. They use this scheme to perform broad microscopic analysis of hematopoiesis in the bone marrow at steady state and under stress conditions of phlebotomy and infection. The application of innovative microscopic analysis to characterize the number, distribution, and proximity of a wide variety of hematopoietic progenitor cells and differentiated hematopoietic cells in situ is a significant advance and important to the field.

Overall the manuscript would benefit from better clarity and rigor in presentation and validation by comparison with existing data. For example, once the authors apply their scheme for identifying various cell types, they should quantify the total number of cells of each cell type in a large section of marrow and compare that to known frequencies from published work. Throughout the manuscript they refer to "production lines" but no definition of this item is given, and the figure

(e.g. Figure 2E) where this is supposedly demonstrated resembles more of a snake or a blob than a line. Therefore, it is unclear how the number of production lines or their cellularity is calculated for Figure 4C. The authors describe some highly unexpected data in which progeny from confetti-labeled cells cluster but do not map near their immediate precursors. The explanation for this is rather perfunctory and does not address the larger question of whether it is an experimental artifact (e.g. symmetric division or loss of labeling from progenitor cells). Finally, the authors tend to make gross generalizations about their data rather than just reporting the findings. A simple reporting of the data and demonstration of its validity would make for a stronger paper.

Specific Comments:

1) The authors use ESAM as a marker to image HSPCs in mice under different conditions including infection and aging. ESAM is known to be widely expressed in HSCs and plays a role in hematopoiesis (Sudo et al., 2012, Sudo et al. 2016); it is not clear whether ESAM is functionally important in maintaining the anatomy of the bone marrow. What are the factors contributing to the distribution of HSCs and the migration of progenitors? How do the authors address the possibility that ESAM-negative HSCs, which are missed by this technique, are localized closer to progenitors?

2) The "experimental pipeline" is depicted in Figure 1A, but the details of this pipeline are not provided in the methods or the supplement. How are the populations defined? CMP are not in this figure. Are we to glean the staining schemes from the incomplete references shown in the figure e.g. "Pietras et al?" Was IL7Ra (CD127) included as a marker for CLP identification?

3) It is interesting that ESAM+ Pre Meg-E differentiate more towards monocyte and neutrophil compared to ESAM- cells (Figure E1C) This should be in the main figure

4)

5) Throughout the manuscript the authors refer to "production lines" but no definition of this item is given, and the figure (e.g. Figure 2E) where this is supposedly demonstrated resembles more of a snake or a blob than a line. Therefore, it is unclear how the number of production lines or their cellularity is calculated for Figure 4C.

6) Re:

"Unexpectedly, confetti-labeled MkP did not map near megakaryocytes labeled by the same color (Extended Data Fig. 3f, g). These results support a model where megakaryocyte progenitors are recruited towards active sites of megakaryopoiesis and terminally differentiate to generate 2-4 mature megakaryocytes."

This is an unexpected finding and the proposed explanatory model is not at all clear. How do the authors exclude other possibilities such as disappearance of the originating MkP due to symmetric differentiation or loss of labeling of the relevant MkP.

The same criticism applies to the discussion of CLP /Pre-Pro B, Pro B, Pre B cells:

Unexpectedly, clonal fate mapping demonstrated that confetti-labeled CLP do not map near Pre-pro B, Pro B, or Pre B labeled with the same fluorescence protein (Extended Data Fig. 4h). These results indicate that the location of common lymphoid progenitors identifies oligoclonal B cell production lines near arterioles.

In Figure 2I to 2K the distances from HSPC to CLP and CLP to Pre-pro B are shown. However, what about the distance from HSPC to pre-pro B?

7) In general the figures are not at all easy to follow and would benefit from fewer panels shown in a larger format. In one particularly important example, Figure 2E is supposed to show an erythroid "production line" The distribution of the cells hardly seems to resemble a line but rather shows a snaking pattern of clustered cells. How is this "production line" defined? What is the evidence that these cells originate from the same progenitor? How does one distinguish one production line from a neighboring one?

8) The confetti data in Figure 3f is an important point but is so small that the reader must take the authors' word for it. The representative yellow MkP does have other yellow cells around it but those are not noted in the diagram.

9) In general the results are written inappropriately in the present tense and with broad overgeneralization. For example, the authors state that "Infection causes minimal increases in myeloid production lines." The data are hardly sufficient to support such a broad statement. One might say, "In this model, minimal changes in myeloid production were seen after infection" As another example of over-generalization, the authors state that "all production lines for a given lineage are synchronized as they simultaneously expand or contract in response to insults." This is an overstatement as the timing of expansion and contraction has not been measured.

10) In evaluating the effects of aging, the authors state that "number of production lines was maintained or slightly increased when compared to 2-month-old mice (Fig. 4c). " How are "production lines" defined? How was cellularity per cluster defined?

Referee #3 (Remarks to the Author):

Wu et al. used new combinations of antibodies to image hematopoietic stem cells, multipotent progenitors and various kinds of restricted hematopoietic progenitors. Along with the prior study published by this group (Zhang et al. Nature 2021), they are performing more complex imaging of the bone marrow than other groups, using more markers to image more cell populations. In this study, they mapped various aspects of blood cell production, including multipotent cells, megakaryocyte, erythroid and lymphoid progenitors. Some of what they reported would be expected from prior studies but the suggestion that multipotent daughter cells migrate far away from their mother cells while differentiating cells remain in clonal "production lines" is new and interesting. The authors conclude that erythropoiesis takes place in production lines adjacent to sinusoids and B lymphopoiesis takes place in production lines near arterioles. Prior studies have localized these progenitors to these blood vessels but Wu et al. add to the prior studies by showing that the differentiated progeny of these restricted progenitors undergo local clonal expansion, in contrast to multipotent cells. The use of the confetti reporter to distinguish different clones is a notable innovation that helps provide insight into what cells remain adjacent to sibling cells and what cells migrate away. Finally, they report that stresses don't seem to alter the overall anatomy of the bone marrow but increase or decrease the expansion of differentiating clones next to blood vessels.

1. On lines 90-91 the authors conclude that "the eLT-HSC and eST-HSC gates are highly enriched in LT-HSC or ST-HSC (Fig. 1e) and have identical activity in competitive transplants as SLAM HSC". But the relevant figures show that the authors transplanted hundreds or thousands of cells in the reconstitution assays. Using such large numbers of cells, it would be impossible to know if there are modest differences in purity between the markers they use and more widely used marker combinations. I suspect the purity of the cell populations they characterized are pretty good, but formally, one would have to perform single cell (or near single cell) assays to know how pure these cell populations are. Many prior studies have done this, performing single cell reconstitution assays in irradiated mice and sorting single cells into culture to determine what percentage of cells have the expected activities. It would be helpful if the authors could provide more information on the purity of the cell populations they imaged.

2. It was hard to read the first section of the paper because the combinations of markers that were used to identify each cell population didn't seem to be listed in a simple way. Readers seem to have to figure it out by scanning through lots of flow cytometry plots in Figure 1. It would be useful to add a table that lists each cell population that was imaged in vivo, whether it is thought to be pure or a mixed population of progenitors (i.e. what it contains) and the markers used to identify the cells during imaging.

3. The observation that stem cells are scattered throughout the bone marrow as single cells has been published in several prior studies. Based largely on this observation, Wu et al. assert that

"multipotent progenitors move away from their parent and sister cells and migrate hundreds of micrometers as they differentiate"; however, it's not clear how they can conclude that. It seems like there are alternative explanations for the observation that HSCs are present as single cells in the bone marrow. One possibility is that daughter cells are released into circulation (rather than migrating through the bone marrow). Another possibility is that at least one of the daughter cells dies, or changes its markers by maturing out of the HSC/multipotent progenitor pool. Do the authors have data to rule out these possibilities? It may not be possible to test whether daughter cells enter circulation but it's a possibility that should be acknowledged.

4. The authors conclude "LT-HSC, ST-HSC, and other multipotent progenitors do not occupy the same niches". Again, it's not clear how the authors could know that, or perhaps, what exactly they mean. Even if we accept that HSC daughter cells migrate hundreds of microns from the site of division, perhaps they are migrating to from one specialized microenvironment that is designed to sustain multipotent cells to another specialized microenvironment that is designed to sustain multipotent cells. Based on prior studies from this and other groups, there seem to be different microenvironments around blood vessels that are specialized for the maintenance of different kinds of stem and progenitor cells. If so, maybe there is a single kind of environment that is specialized to maintain LT-HSCs, ST-HSC, and multipotent progenitors, distributed in different locations around the sinusoids throughout the bone marrow, and HSCs/multipotent daughter cells migrate among these specialized sites when they divide?

5. A prior study concluded that at least some B lineage progenitors localize to sinusoids (Immunity 20:707, 2004) while Wu et al., in agreement with another prior study, conclude that lymphoid progenitors are around arterioles. These studies do not necessarily contradict each other as they may each be imaging different lymphoid progenitors. However, it would be helpful if Wu et al. could address the question of whether they think all lymphoid progenitors localize to arterioles or whether they see some evidence for a subset of lymphoid progenitors that is around sinusoids.

6. The authors conclude that in aged mice "lymphoid- and myeloid-biased LT-HSC localize much closer to other LT-HSC than predicted... leading in some cases to the emergence of loose clusters of myeloid- and lymphoid-biased LT-HSC". Can the authors be more precise about what they mean here? Are lymphoid biased HSCs clustered with other lymphoid biased HSCs (and myeloid with myeloid) or do HSCs cluster together irrespective of whether they are myeloid or lymphoid biased? It's an important detail because lymphoid biased HSCs probably arise from myeloid biased HSCs only after they have undergone multiple rounds of division.

Referee #4 (Remarks to the Author):

The manuscript "Resilient anatomy and local microplasticity of naïve and stress hematopoiesis" by Wu, Zhang et al. describes a new strategy to image long term hematopoietic stem cells (LT), multipotent progenitors, megakaryocyte progenitors, erythroid progenitors, and lymphoid progenitors in their bone marrow niche. The authors use the novel imaging strategy to describe the anatomy of murine hematopoiesis in steady state and after acute stress and upon aging. The data show distinct microanatomical areas of selective hematopoietic production lines. According to the data, the different production lines are resilient upon acute stress and aging, while the anatomy of the bone marrow is temporarily (in case of acute stress) or permanently (upon aging) remodeled in a cell- and lineage-specific manner.

The manuscript contains a vast amount of data and high-quality images that show with excellent resolution and very elegant graphical support the anatomy of the bone marrow in the murine sternum.

The overall dataset is in its nature quite descriptive. The imaging technology is absolutely novel and qualitatively impressive. However, the major biological findings are largely anticipated by previous publications. Overall, the data constitutes mainly an in-depth high-quality confirmation of previous observations. There are a few points where the authors offer an interpretation of their dataset that could expand our knowledge on the biology of the hematopoietic system. However, these claims either require to be explained and phrased more clearly or deserve to be corroborated

by additional mechanistic data.

Major general points:

1) Was only the sternum used throughout the analyses? Why? Would the long bones (femurs, tibiae) or the calvarium show any difference? Whole-mount histology has been described in several type of bones. The validation of the differentiation lines' localization in different bones would corroborate the findings. In line with this point, also vascular and differentiation lines alteration upon stress and aging should be verified in other bones.

2) Line 77-78: "We thus defined ESAM+ Pre 77 Meg-E as myeloerythroid (MyE) progenitors". In Extended figure 1c, ESAM+ MegE are contributing much less to erythroid cells in comparison to the ESAM- counterpart. The data suggest that ESAM+ Meg-E give rise to myeloid cells while mainly ESAM- Meg-E differentiate to erythroid cells. This aspect should be clarified by demonstrating in Tx that ESAM+ Meg-E are sufficient to produce most erythroid cells, or the analysis and interpretation should be revised accordingly.

3) Page 5-6, lines 117-123: "all stem cells and multipotent progenitors are found as single cells, evenly distributed through the bone marrow, and are no closer to each other than predicted from random simulations. This spatial segregation is not due to differential interaction with the microenvironment as most multipotent progenitors showed localization similar to that of LT-HSC: closer than random to megakaryocytes but farther than random from arterioles". How is the random distribution calculated? Many conclusions throughout the manuscript are drawn related to the random distribution, and it would help the reader having this aspect clearly described in the main text. Does the random distribution take into consideration the frequency of a given cell type? Can cell frequency affect randomness? ST look closer to arterioles and further from sinusoids than LT. Also, ST look closer to the other cell types than to LT. so the conclusion about not being closer than predicted from random distribution is right but not complete in describing that some cell types are closer to others.

4) The statistical approach used for the analysis in fig2 f-g, Extended figure 3c and in similar analyses throughout the manuscript should be revised. The data don't look like having a normal distribution and the choice of a parametric test should be reconsidered, unless an analysis of normality distribution has been performed. In this case this point should be justified in M&M section. The random distribution in each of the graphs of fig2 f-g and Extended figure 3c looks the same for all the different cell types and it changes based on the reference to sinusoids, arterioles or Megakaryocytes. The difference in frequency of each cell type doesn't affect the random distribution calculation for that given cell type? Why the random distribution in fig 2c and in Extended figure 3b doesn't look the same for all cell types like for the other graphs?

5) Page 6, lines 125-127: "These results indicate that multipotent progenitors move away from their parent and sister cells and migrate hundreds of micrometers as they differentiate". It is difficult to support that the data show that multipotent progenitors "move away" from their parent and sister cells. If there are no difference from random co-localization, it is difficult to believe that is not just a stochastic observation and that each cell is where it is because of randomness and in this sense that there is any direction to move away from anywhere. This conclusion needs to be better supported by data or by stronger mathematical modeling. LT and progenitors have different proliferation rate. Cells actively dividing and sister cells would be found closer after division. It is unexpected not to see differences at all between quiescent LT and more active progenitor cells. Do the authors observe any specific localization of proliferating cells? For example, can the authors stain for Ki67+ LT and progenitors and inform on whether there is any difference based on proliferation?

6) Overall, the claim of randomness vs implying that LT-HSC, ST-HSC, and other multipotent progenitors do not occupy the same niches it's confusing. If, as assumed in the manuscript, LT and

progenitors are evenly distributed through the bone marrow and there is no differential interaction with the microenvironment, it is not clear how it has been defined that each cell type has a niche. It looks more similar to a homogeneous suspension of different cells and the co-distribution of cells would be mainly affected by the frequency of a given cell type. Can the authors clarify this point?

7) Eventually the data show only that LT and progenitor cells are localized in different niches, there is no experimental evidence of migration of these cells. The term migration implies a directionality and it is very speculative from the data to suggest which cell moves and where. So, unless experimentally demonstrated, this statement should be rephrased.

8) Line 140-142: The authors claim that the absence of the progenitor in the same area where the differentiation cluster is localized indicates that the progenitor terminally differentiates. The idea is very nice and the in situ identification of this step has never been provided so far. However, this claim should be supported for example by an in vitro differentiation assay or a tracer specific-line, or by live imaging, as the histology per se can't exclude that the differentiated cells have been produced in another niche and then migrated in the new one, as suggested for LT and progenitors.

9) Line 143-145: same point about the term migration as explained before

10) Line 150-153: Experimental proof of the suggested differentiation steps are missing. Moreover, the authors should show that CFU-E CD177- CD71+ are far from sinusoids but not co-localizing with differentiate erythroid cells. It would be interesting to support the data showing the distance from sinusoids for CFU-E, CFU-E cd177- CD71+, differentiated erythroid cells together in the same plot.

11) Line 161: Extended figure 4 g is confusing. Arterial staining is missing or at least not shown despite the distance analysis is present (fig 2j). Until this point in the manuscript the ESAM staining has been exclusively associated with stem cells, megakaryocytes and sinusoid identification, how is this explained? In general, all these supporting cartoons are slightly confusing because when compared to the microscope image next to it they don't look like depicting the same. Either it should be explained better what do these cartoons depict or they should be shown next to the exact same image (maybe simplified) that was used to generate them

12) Line 164-169: Analysis in confetti mice (line 168-169) does not recapitulate the previous observation (line 164-165). Can these data be explained more clearly?

13) Line 211: Data in figure 3e and Extended figure 5a are in contrast with the text: in the text the authors indicate that number of cells in the B cell line does not decrease after hemorrhage, however data in figure 5a indicated a reduction of these cells. Figure 3e only shows that global lymphoid cell number is not affected, there is no proof that B cells are not altered here. Can the author clarify this point?

14) Line 212-213: the data is only qualitative. A quantification is necessary to support the statement.

15) Line 207-217: This analysis is very interesting, but the outcome is somehow expected. As the differentiation lines adapt/respond differently to the stress it would be interesting to check how the different cells in each line are affected (i.e., clonal contribution). It would be nice to perform the experiment in confetti mice or in reporter cell lines to better describe the plasticity of the lines.

16) Line 220: the B cell line per se has not been tested, but only CLPs and lymphoid cells in general. Quantification of B cell number and localization should be given to support the data.

17) Line 225: Missing quantification in Extended figure 7f-h.

18) Page 11 lines 259-265: It would be clearer to show a direct comparison between young and aged samples. The data in panels (Fig. 2a-c) are difficult to compare with (Fig. 4d).

19) Page 11 lines 265-270: The reduction in erythropoiesis and lymphopoiesis upon aging have been extensively reported. It is unclear why the authors describe them as "microanatomical adaptations"? What do the authors mean with "microanatomical adaptations"? It looks like the changes upon aging are due to an adaptation to the changes in the microenvironment but these microenvironmental changes are not clearly described nor the mechanism that should justify the "adaptation". Can the authors provide data that the reduction in erythropoiesis and lymphopoiesis are due to a microanatomical adaptations? Otherwise, it looks unclear what are the authors referring to.

Minor points:

- Line 70-72: ESAM as a marker of HSPCs has already been described/suggested (i.e. Yokota T. Blood 2009 and Tomohiko I. Blood 2015). References should be included.
- A table to recap all the surface markers associated to each cell type for each production line would be extremely helpful.
- A table to recap the surface markers Abs and the secondary Abs and relative conjugated fluorophores used in the histological analysis would be helpful to guarantee data reproducibility.

Author Rebuttals to Initial Comments:

Response to referees for “Resilient anatomy and local microplasticity of naïve and stress hematopoiesis“ by
Wu, Zhang et al.,

Response to Referee 1.....Page 2

Response to Referee 2.....Page 17

Response to Referee 3.....Page 31

Response to Referee 4.....Page 37

Referee #1 (Remarks to the Author):

The authors identify a set of 35 markers that can be labelled by antibodies and allows the unequivocal distinction of all relevant types of hematopoietic stem and progenitor cells by multicolor imaging. After proving the validity of their marker set with functional assays using transplantation models they set out to characterize the base-line, stress- and aging-induced spatial organization of hematopoiesis in the murine bone marrow on the level of individual cells. The principal question and the used approach are both highly original and provide an exciting and extremely interesting view into the mechanics of hematopoiesis in situ. As such they are highly relevant for the field of hematopoiesis and beyond. However, I have identified several critical issues that need to be addressed before the study provides a fully convincing case for Nature.

We are deeply grateful to the reviewer for the positive review and extremely helpful comments. We have addressed all the critiques in detail in the point-by-point response below. We would also like to highlight three new major pieces of data that further increase the rigor and novelty of our findings. These are:

- 1) Identical anatomy of normal hematopoiesis across the skeleton:** in addition to the sternum we have now determined the anatomy of hematopoiesis across the skeleton by mapping four additional bone types: tibia, humerus, vertebrae, and the lamboid sutures of the skull. In all cases we found that the anatomy of hematopoiesis *–in the steady-state–* was essentially identical to that of the sternum (which we described in the initial version of the manuscript). *These new results are now shown in **Extended Data Fig. 6**.*
- 2) Differential movement and clustering of hematopoietic progenitors:** combining in vivo HSC transfer with live imaging of the skull we demonstrate that HSC move away from each other when in close proximity in vivo (**Fig. 2c,d**). Using live imaging of cultured cells we demonstrate that the offspring of HSC and multipotent progenitors always relocate away from each other. In contrast, the offspring of committed progenitors remains in contact (**Fig.2 q,r** and **Extended Data Figure 5**). These results explain the differences in progenitor clustering found using our ex vivo mapping and support that cell autonomous mechanisms control differential progenitor clustering at different maturation stages. *These new results are now shown in **Figures 2c,d; Fig. 2q,r** and **Extended Data Figure 5**.*
- 3) Stress responses vary across the skeleton:** as requested, we mapped the anatomy of the stress response after G-CSF. At the request of reviewer 4 we also mapped the stress responses across the skeleton. These great suggestions led us to an unexpected discovery: the bone marrow response to a hematopoietic insult varies depending on the bone examined. For example, while tibia responds to G-CSF treatment by dramatically increasing neutrophil production the sternum responds to G-CSF by shrinking neutrophil production sites and neutrophil output. Similarly, hemorrhage triggers CFU-E amplification in the tibia and sternum but not in the skull. These experiments demonstrate that the bone marrow response to hematopoietic insults varies across the skeleton, and that this response is bone and insult specific. This is a new paradigm in the field that was possible due to our ability to interrogate smaller bones that are difficult to interrogate by FACS and thus have not been routinely examined in the literature. *These new results are now shown in **Figure 4** and **Extended Data Figure 10**.*

Specific points (in chronological order from introduction to discussion) are:

1) Ref. 10: intravital imaging of HSC motility has been achieved considerably earlier than in author's ref 10. Please also refer to 10.1182/blood-2008-12-195644

The reviewer is right, and we apologize for this oversight. The requested reference (now reference #11) has been added to line 72 of the manuscript. The text now reads "The bone marrow has extraordinary plasticity and quickly adjusts blood production to meet physiological demands in response to insults^{3,4}. Despite recent progress⁵⁻¹² the anatomical organization of normal and stress hematopoiesis remains largely unknown."

2) Fig. 1A: the indicated subtypes of stem cells are not fully reproduced in the surface marker analysis in Fig. 1b and in Fig. 1c. CMP is missing and MP+cMoP are not noted in Fig. 1a. This is confusing and unclear. Further, I could not find a structured list of all 247 antibodies tested. This should be provided as a supplementary table.

We thank the reviewer for identifying this problem. We have edited the corresponding for panel. Due to the staining strategies used (**Extended Data Fig. 1a**) we did not immunophenotype CMP directly. Instead, we profiled the Pre-GM, which is a mixed population containing both MDP and CMP (**Extended Data Fig. 1c**). We now indicate in (**Figure 1b**, reproduced below) that Pre-GM are a mixed population. We subsequently identified bona-fide CMP within the Pre-GM gate as shown in **Extended Data Fig. 1e,f**. MP and cMoP are now noted clearly in **Fig. 1b** below.

We now also provide a list of all 247 antibodies tested in **Supplementary Table 1**.

Figure 1 b, Histograms showing differential expression of 35 cell surface markers across 14 HSPC types.

3) The fact that ESAM is a HSC marker is known since at least 2009. What is the novel discovery made in Fig. 1c? This must be explained in relation to existing literature.

As indicated by the reviewer and references 20 and 21 ESAM is enriched in LT-HSC and other progenitors. Our novel findings are:

- 1) that ESAM also identifies the most potent MPP2, MPP3, MPP4 and CMP in functional transplantation assays (**Fig. 1d**)

- 2) that ESAM in combination with the other markers shown in **Fig. 1e** can be used to simultaneously image six distinct cell populations that are highly enriched in specific types of stem cells or multipotent progenitors. Essentially, ESAM allows replacing the lineage panel, Sca1 and Flt3 stains (as the latter two do not work in imaging) to image the indicated cell populations.

To make this point clearer the text (lines 111-117) now reads: “These experiments demonstrate that ESAM identifies multi- and oligopotent progenitors in the bone marrow. They also led us to an isolation strategy replacing the Lineage and Sca1 and CD135 antibodies –which are classically used to prospectively isolate multipotent progenitors by FACS but do not work in immunofluorescence- with ESAM. This strategy allows simultaneous detection of highly purified LT-HSC, ST-HSC, megakaryocyte progenitors (MkP), MPP2 and MyE (containing all functional MPP2 and MyE), and a mixed population containing the functional MPP3, CMP, and ESAM⁺ MPP4 (**Fig.1e**)”

4) Fig. 1d: the experimental design is very hard to impossible to understand from the figure. It implies that each recipient received ESAM⁺ and ESAM⁻ cells at the same time, which was not the case. So, I suggest to add an "or" between ESAM⁺ and ESAM⁻ in the scheme. Then I find it hard to conceive that MPPs have different performance in repopulation, if almost 6 times fewer ESAM⁻ cells were transferred in comparison to ESAM⁺. This number must be identical. For MPP4 the ratio changes in favor of ESAM⁻ cells. Also this makes it difficult to compare the performance of the cells. The legend is also unclear: was a total of e.g. 5 mice analyzed in the MPP2 experiment in 3 different runs, or were 3 different runs performed each with 5 mice per group? Finally, the labelling of the y-axis in 1D should also contain a "%".

We are deeply thankful for the suggestions on how to improve the figure. As requested, we have modified the scheme to indicate that different groups of mice received ESAM⁺ or ESAM⁻ cells. We have also repeated the transplantation experiments using identical numbers of ESAM⁻ and ESAM⁺ cells as requested (New **Fig. 1c,d reproduced in the next page**). These new results confirm that multilineage engraftment potential is essentially restricted to the ESAM⁺ fraction. Since we repeated these experiments transplanted equal numbers of cells we have edited the legend to indicate that a total of four recipient mice per group were transplanted.

Figure 1 c, ESAM expression in indicated HSPC. **d**, Percentage of donor-derived cells in the peripheral blood after transplantation of the indicated progenitors (n = 4 mice). All data represent mean ± s.e.m. Statistical analysis was performed using Two-way ANOVA, followed by Multiple unpaired t tests, P values are shown.

5) Fig. 1f as final result of all work in 1a-e is very nice. Please make sure the margins of all images align at the top and bottom.

Thank you and edited as requested.

6) Fig. 2A and related: all bone marrow imaging was done in the sternum, which is a flat bone. However, transplantation and CFU assays were all performed with cells sorted from long bones. Since the blood vessel system of long bones is different from flat bones (see e.g. 10.1038/s42255-018-0016-5), it is not clear, whether both bone marrow sites behave the same and hence stem cells from both sites behave the same. Therefore, the analysis of HSC and MPP must also be performed in long bones as has been done for different questions before (e.g. 10.1038/nature15250).

This is a very important point. We have now performed our imaging analyses of hematopoiesis across the skeleton including two long-bones (tibia, humerus), lumbar vertebrae, and the interparietal bone of the skull. In all cases we confirm that the anatomy of hematopoiesis in young mice in the steady-state across the skeleton is essentially identical to the one that we described for the sternum (new **Extended Data Fig. 6** reproduced below).

Extended Data Figure 6. The anatomy of hematopoiesis in the steady-state is maintained across the skeleton. a, To determine whether the anatomy of hematopoiesis was comparable across the skeleton we imaged and compared hematopoiesis in the sternum, tibia, humerus, L5 vertebrae, and the lamboid sutures of the skull. In all bones we observed long distance separation of multipotent stem and progenitors, selective recruitment of committed progenitors to vessels, and terminal maturation in lineage specific production sites. Since bone sizes are dramatically different, the maps show the distribution of the indicated cells in areas of similar size for each bone. **b-f**, We noted that some cell types display large variations in frequencies between bones –e.g. LT-HSC were less abundant in the tibia- leading to changes in distances to closest cells. The graphs show the distances from each LT-HSC (b, n= 19, 21, 25, 23 and 26), CFU-E (c, n=221, 501, 495, 139 and 548), CLP (d, n=28, 13, 50, 15 and 22), GP (e, n=79, 51, 51, 46 and 46) or MDP (f, n=29, 29, 30, 19 and 23) to the closest indicated cell in the five bones examined.

7) Mentioning the distance of LT-HSC to each other as >100 μm : for non-expert readers authors should add a comparison to the size of the cells. From their images in fig. 1f this should be $\sim 8\mu\text{m}$. So, 100 μm equal to 12-13 cell diameters.

As requested, this is now indicated in the text of the manuscript (lines 141-144): “Imaging of 2-month-old mouse sternum segments showed that all stem cells, multipotent, and oligopotent progenitors are found as single cells with median distances to the closest progenitor of more than 100 μm (>10 cell diameters, Fig. 2a,b)“

8) Furthermore, it is essential to provide a 3-D analysis of the spatial arrangement of HSC in the marrow. Currently the results suggest that all cells are in one single plane, which they are obviously not. It is hence required to perform a volumetric analysis and consider all neighboring cells in a volume around one cell. Imaging just the top 35 μm of sternum is not sufficient to reach this goal, if distances between single HSC can be as big as 100 μm and more. Issues with poor antibody penetration can be solved, e.g. by using flushed marrow plugs (less preferred due to corruption of trans cortical blood vessels) or e.g. cut bones as in 10.1038/nature15250 or [https://www.cell.com/immunity/pdfExtended/S1074-7613\(18\)30038-4](https://www.cell.com/immunity/pdfExtended/S1074-7613(18)30038-4).

This is extremely challenging. Our experimental protocol is based on minimal manipulation of the bone marrow combined with fast staining to preserve epitopes of the hematopoietic cells. This is the reason that we can use directly conjugated FACS antibodies to detect all cells of interest. Our group has tried multiple times to adapt clearing and deep staining protocols but key epitopes (ESAM, CD48, and others) disappear due to the long fixation and permeabilizations required. Nevertheless, we tried multiple different approaches to increase the depth of the stains. As shown in **Reviewer Fig. 1a-c** in the next page most of these approaches were unsuccessful. The only method that showed increased depth was embedding the whole mounted sternum in OptiPrep Density Gradient medium. These resulted in better matching of the refractive indexes between the sample and the objective increasing imaging depth to 40 μm (**Reviewer Fig. 1d**). Based on these experiments we do not believe that it is technologically feasible to perform deeper imaging analyses of hematopoiesis without generation of specific fluorescent reporter mice (as done in the nature manuscript cited by the reviewer which required the use of $\alpha\text{-catulin}^{\text{GFP}}$ reporter mice 10.1038/nature15250)

However, we would like to highlight an important new piece of data that further supports our initial conclusions. In response to point #6 above we have imaged the distribution of HSPC in five different bones including the lamboid sutures of the skull. The lamboid sutures are thin enough that we were able to image the whole tissue until reaching the endosteal surface (**Reviewer Fig. 1e**). In this bone the spatial organization of hematopoiesis is identical to the sternum and very similar to that of long-bones (**Extended Data Figure 6a,b** available in the previous page). These results strongly suggest that our quantifications in thin optical sections faithfully represent the actual distributions present in the tissue.

Because it was technologically impossible to visualize stepwise hematopoiesis in the whole bone in deep imaging we acknowledge this as a limitation in the discussion lines 325-328: “A limitation of our study is that we cannot rule out that these lines are motile -or transient- as we cannot track the same marrow over time. A second limitation is that we image relatively thin slices (30-35 μm) preventing detection of all cells present in the marrow”

Reviewer Figure 1.

9) Fig. 2c and related supplemental figures: the labelling of the lower part of the graph is difficult to read if upside down. Please flip.

At the request of reviewer 4 we have modified these graphs (now **Extended Data 3a**) to also show the dots corresponding to the random cells. No graph is now labeled upside down.

10) Fig. 2e: are the shown CFU-E inside, above or below of the blood vessel? This is not clear without a 3-D analysis. Authors should also analyze in long bones specifically, how these cell types and the other investigated stem/progenitor cells localize with respect to the recently described transcortical blood vessels (10.1038/s42255-018-0016-5), which are known to be the main routes for bone entry/exit of blood.

We apologize for the lack of clarity. CFU-E specifically attach to exterior of the vessel wall (i.e., facing the marrow). We now provide 3D videos clearly showing the location of these CFU-E with respect to vessels (**Supplementary Movie 1**).

As requested, we have now imaged the location of HSPC in regards to the endosteum (which contains all transcortical blood vessels **Extended Data Fig. 3c**, see below). As a population, all types of multipotent HSPC and CFU-E are far away (median distance 5-8 cell diameters) from the endosteum – and transcortical blood vessels and these distributions were not different from that of random cells (**Extended Data Fig. 3e** and **Extended Data Fig. 4b**).

Extended Data Figure 3. HSPC localization in the mouse sternum. **c**, Map showing the distribution of the indicated HSPC in relation to the bone (detected using second harmonic generation). The histogram indicates that the HSPC do not map near bone surfaces ($n = 19$ LT-HSC, 50 ST-HSC, 9 MPP2/MyE, 27 MPP3/MPP4/CMP, 34 MkP, 35 Pre Meg-E in 4 sternum segments from 3 mice). **d**, In agreement, neither ESAM⁺ HSPC nor CFU-E localize near the transcortical blood vessels (TCV) that penetrate the bone. **e**, Intravital bone marrow confocal images 48hrs (top) and 51hrs (bottom) following the transplantation of a Tie2⁺ HSC co-labelled with Dil (ref) (Scale bar: 50 μ m). (Top) As a single cell was transplanted, the two cells visible at 48h are necessarily originating from cell division. (Bottom) One of the two cells visible at 48hrs was no longer visible in the whole calvaria at 51hrs.

1) Suppl. Fig. 4k/l: the amount of Ly6G label in the marrow appears to be extremely low. Neutrophil contents in murine marrow are several millions per long bone (see e.g. <https://www.ncbi.nlm.nih.gov/pmc/articles/PMC3732092/>) and hence one should see many Ly6G+ cells. Where does this incompatibility come from? Or is it a specificity of sternal bone marrow?

We believe that the appearance of low numbers of Ly6G+ cells in **old Fig. 4k/l (now Supplementary Figure 2d)** is due to the small image size and because we used dark purple (over black background) for that channel making difficult to appreciate the dimmer Ly6G+ cells. In our hands Ly6G consistently stains 20-30% of all hematopoietic cells (**Reviewer Fig. 2** and reference 1). Below we show a larger version with increased contrast and a zoomed out image of the sternum stained with the same antibodies.

Reviewer Figure 2.

12) The scheme in Fig. 2o (now new Fig. 2s, t) suggests that CFU-E and MDP are within blood vessels. Is this intended? From the data and logics, it appears they should be close to but not inside of sinusoids.

We apologize for the lack of clarity in the scheme. These progenitors are attached to the marrow-facing side of the vessel. A professional artist has now redrawn these schemes (new Fig. 2s, t) to make this point clearer.

Figure 2. Anatomy of steady-state hematopoiesis in young mice. **s**, Scheme highlighting different localization and clustering patterns during differentiation. **t**, Scheme showing the overall anatomy of hematopoiesis in 2-month-old mouse sternum

13) Fig. 3e: the studies on stress hematopoiesis are very interesting. It is, however, very unexpected, that neutrophil production does not seem to sense these changes. Hence, I am missing clinically relevant models, that mimic gross neutrophil changes in the bone marrow. Mobilization by G-CSF/AMD3100 can mimic cytokine/drug-induced neutrophil/stem cell mobilization and bone marrow reconstitution after lethal irradiation mimics clinical stem cell transplantation. These models are all very close to real-world clinical situations and should thus be investigated with the tools established in the study.

We thank the reviewer for these suggestions that have led us to uncover completely new biology. As requested, we stimulated mice with G-CSF to induce neutrophil expansion. We first imaged the bone marrow response to G-CSF in the sternum because our data demonstrated identical anatomical organization across the skeleton in the steady-state (tibia, humerus, sternum, lumbar vertebrae and skull, **Extended Data Figure 6**). G-CSF treatment led to dramatic reductions in the numbers and output of all types of production sites examined (**Fig. 3b-j** reproduced on the next page). This was extremely surprising because G-CSF leads to increases in granulopoiesis in long-bones (**Fig. 3a** and **Supplementary Fig. 5**). This led us to hypothesize that stress responses vary across the skeleton. To test this hypothesis, we imaged granulopoiesis in the sternum and tibia after G-CSF. While the tibia responded to G-CSF by increasing neutrophil output the sternum responded by shutting down neutrophil production leading to loss of neutrophil production sites (**Fig. 4a-c** on page 13). Both bones showed comparable anatomy in saline-treated mice (**Fig. 4a-b**). The tibia and sternum displayed similar suppression of erythropoiesis, monoopoiesis, and lymphopoiesis indicating that both bones are equally exposed to G-CSF (**Fig. 4d**). To determine whether this phenomenon extended to other insults we induced phlebotomy and examined the response across the skeleton via FACS and imaging. Phlebotomy caused a powerful expansion in erythroid production site numbers and output in the sternum, tibia, vertebrae, and humerus (**Fig. 4e-f** and **Extended Data Fig. 10a,b**). However, we did not detect changes in erythroid production site numbers or output in the skull even though phlebotomy-induced reductions in lymphopoiesis were similar in all bones indicating that the phlebotomy was sensed by hematopoietic cells in the skull (**Fig. 4f-g** and **Extended Data Fig. 10c**). These experiments demonstrate that the response of bone marrow

production sites to systemic insults is bone- and insult-specific and uncover unprecedented heterogeneity of stress responses across the skeleton.

Figure 3. Anatomy of stress responses. **a**, Heat map summarizing changes (normalized to steady-state or saline-treated) to the indicated populations (exemplars of peak erythropoiesis, lymphopoiesis, granulopoiesis, and myelopoiesis responses) two days after phlebotomy (except for lymphopoiesis which corresponds to day 8 as this is the nadir of the Lymphopoiesis response as shown Supplementary Figure 3), six days after *L. monocytogenes* infection, four days of G-CSF treatment, and 20 months of age (n=7, 6, 4 and 6 mice for each indicated population two days after phlebotomy, six days after phlebotomy, 4 days of GCS-F treatment and 20-month old aging mice,).

b, Maps showing HSPC location at the indicated time points and insults. Map dots are three times the average size of the relevant cell. **c-j**, the graphs quantify the number of production sites per sternum (left panels, n= 3 sternum segments from 3 mice) and the cellularity of each production site (right panels, n = 6 randomly selected production sites in three sternum segments from 3 mice) at the same time point and insult as in "a". The maps show the distribution of the indicated cells in a large region of a sternum segment. Statistical differences were calculated using two way ANOVA t-tests if the distributions were normal or Kruskal-Wallis test if not normal; P values are shown.

Figure 4. The hematopoietic response to stress varies across the skeleton. **a**, maps showing the distribution of granulocyte progenitors (GP, pink) and preneutrophils (yellow) in whole-mounted sternum and tibia treated with saline (S) or G-CSF (G, 250mg/kg/day) for 4 days. **b**, after G-CSF the sternum has fewer GP and the neutrophil sites disaggregate (n= 3 sternum or tibia from 3 mice per treatment). **c**, **d**, Number of indicated cells in sternum and tibia quantified by FACS (n=4 mice in four independent experiments). **e**, maps showing the distribution of CFU-E (red, terminal erythroid progenitors) in whole-mounted sternum, tibia, or lambdoid suture of the skull bone in the steady-state or 2 days after phlebotomy. **f**, Quantification of erythroid production site frequency (upper panel n=3 bones each from 3 mice) and number of CFU-E per site (n= 10 randomly selected production sites from 3 bones each from 3 mice) in control (ctl) or 2 days after phlebotomy (Phl). Statistical differences were calculated using two way Student's T-tests.

We agree that using our tools to image how stem and progenitors engraft after lethal irradiation would be of interest. However, myeloablation also destroys the vessels that organize hematopoiesis (doi:10.1016/j.stem.2009.01.006, doi:10.1038/nature12612, doi:10.1038/ncb3570, doi:10.1016/j.celrep.2013.07.048, doi:10.1038/nm.4448). In addition, the kinetics of recovery are complex and highly dependent on the amount (and types) of cells transplanted (doi:10.1038/nm.4448). Because of this, we believe that these specific analyses are too complex and outside of the scope of this manuscript.

14) I very much appreciate the graphical summary in Fig. 4I. I would suggest to also make some kind of animation of these main features and provide this as additional supplemental movie, since the subject is very complex and not easy to grasp for non-experts.

This is an excellent suggestion. The requested animation is now provided as Supplementary movie 7.

15) Discussion: I consider the statement, that the bone marrow "was the last major tissue, where the spatial relationships between progenitors and differentiating cells – the anatomy- remained uncertain" a bit far-

fetched. I am certain we do not have comparably detailed information on many other tissues. Hence, I suggest to tone down the wording here.

We have removed this statement. The text now reads: “The spatial relationships between progenitors and differentiating cells – the anatomy- in the bone marrow remained uncertain”.

16) Authors state in discussion, that "stem cells migrate through bone marrow". It has been shown by intravital imaging long ago (see e.g. 10.1182/blood-2008-12-195644) that they actually hardly move but instead are mainly sessile. Things change dramatically after stress mobilization which is actually the reason for my critique above regarding the investigations on stress hematopoiesis. Please adapt the wording.

We have removed this statement. The text in the discussion now reads: “Here we developed strategies to visualize stepwise hematopoiesis across the mouse skeleton. We discovered a sophisticated and elegant anatomy of hematopoiesis. It is characterized by long-distance spatial segregation –likely mediated by cell separation after cell division of multi- and oligopotent progenitors– that are enriched near megakaryocytes; recruitment of lineage-committed progenitors to distinct blood vessels; and defined microanatomical production sites responsible for producing mature cells for each major blood lineage”

Please note that (at the requests of reviewers 3 and 4) we provide new live imaging in situ data supporting that HSPC separate after cell division. As the reviewer points out, previous intravital imaging studies showed that single HSC in the marrow are largely sessile (new references ^{9,11,12}). Because most HSC are found as single cell in the marrow (^{29,30} and **Fig. 2a**) the mobility of HSC when adjacent to other HSC/MPP has not been examined. To explore this in detail we established a collaboration with the laboratories of Keisuke Ito and Charles Lin. We performed follow-up analysis of microscopy-guided transplantation of single Dil- labeled, Tie2⁺ HSC in the mouse calvarium as described ³². In one instance we observed that (48 hours after the initial transplant) the sole transplanted HSC had divided generating two Dil- labeled, Tie2⁺ cells that were in close proximity. Three hours later one of the daughter cells was no longer visible in the whole calvarium suggesting that it had moved away (**Extended Data Fig. 3e**). These results led us to hypothesize that HSC move away from each other when in close proximity. Follow-up analysis of single vs multiple cell transplants (5 recipients received one LT-HSC and 6 recipients received respectively 5, 5, 5, 17, 19, and 22 Tie2⁺ labeled LT-HSC, **Fig. 2c reproduced on next page**). We found that a single cell was visible in 80% of the single cell recipients (4 out of 5) and the cells were all found in near (<100µm) of the transplantation site. Only one out of 73 cells was found –as a single cell– in the mice that received multiple HSC (**Fig. 2d**). Importantly, all tracked recipients showed long-term HSC engraftment that correlated with the number of HSC transferred (Supplementary Table 3 and ³²) indicating that the absent HSC did not die or terminally differentiate. These experiments indicate that HSC move away from each other, when in close proximity, in vivo. The fact that multipotent HSPC are always found as single cells whereas lineage-committed progenitors form clusters with daughter cells (**Fig. 2a, e, m**) prompted us to investigate whether this was mediated by cell autonomous mechanisms. In live imaging analyses of cultured cells, we found that –after cell division- the offspring of HSC and MPP rapidly moved away from each other. In contrast most committed progenitors remained tightly attached after cell division (**Fig 2q, Extended Data Fig. 5, and Supplementary movies 2-6**). These results explain the differences in progenitor clustering found using our ex vivo mapping and support that cell autonomous mechanisms control differential progenitor clustering at different maturation stages.

Figure 2. c, Tie2⁺ HSC purified from actin-GFP mice were transplanted directly into the calvarial bone marrow of living mice using the approach described in reference 32 as either single cells (5 recipients received 1 cell) or multiple cells (6 recipients received respectively 5, 5, 5, 17, 19, and 22 cells). **d**, The graph reports the fraction of cells found using intravital microscopy in the whole calvarial bone marrow 24hrs following transplantation ($p^{***} = 1.96 \times 10^{-9}$, chi-square test to compare two proportions). A single cell was visible in 80% of the single cell recipients (4 out of 5) and the cells were all found in close proximity ($<100 \mu\text{m}$) of the transplantation site. Only one out of 73 cells was found [as a single cell] in multiple cell recipients.

Recipient	number of transplanted cells	Peripheral blood chimerism (week 4)
M1	1	not tracked
M2	1	1.14
M3	1	0.46
M4	1	0.23
M5	1	2.76
M6	5	2.59
M7	5	not tracked
M8	5	5.4
M9	17	not tracked
M10	19	69.4
M11	22	13.3

Supplementary Table 3. Peripheral blood engraftment in mice after microscopy-guided single or multiple HSC transplantation

Figure 2. q, Distance between daughter cells at the indicated time points after division from a cultured $n = 30$ HSC, MPP, CFU-E, or MDP and $n = 18$ GP or CLP. **r**, Percentage of daughter cells that have separated more than $50 \mu\text{m}$, 2 hours after division. ($n = 5$ wells for each indicated progenitors).

17) I cannot find references in the text to Refs. 30 and 31.

These references can be found in the methods section, lines 764-767 and 849-853 respectively. To facilitate the reviewer's work the specific sentences are indicated below.

The specific text for reference 30 (now 40) reads: Whole-mount sternum immunostaining has been described before⁴⁰. Briefly, the sterna were dissected and cleaned of soft and connective tissue, followed by sectioning along the sagittal or coronal plane to expose the bone marrow under a dissecting microscope (Nikon SMZ1500 Stereomicroscope).

The specific text for reference 31 (now 41) reads: "For generating random distributions of cells in experiments using confetti mice we first obtained the coordinates and confetti color for each type of cell in each section analyzed. Then we used Research Randomizer⁴¹ to randomize the confetti label while maintaining the spatial coordinates of each cell. We then measured the distances between these cells with randomized colors. Each random simulation was repeated 100-200 times"

Referee #2 (Remarks to the Author):

Here, Wu et al develop new 4-6 color labeling scheme for hematopoietic progenitors that can be used with confocal microscopy. They use this scheme to perform broad microscopic analysis of hematopoiesis in the bone marrow at steady state and under stress conditions of phlebotomy and infection. The application of innovative microscopic analysis to characterize the number, distribution, and proximity of a wide variety of hematopoietic progenitor cells and differentiated hematopoietic cells in situ is a significant advance and important to the field.

We are deeply grateful to the reviewer for the positive review and extremely helpful comments. We have addressed all the critiques in detail in the point-by-point response below. We would also like to highlight three new major pieces of data that further increase the rigor and novelty of our findings. These are:

1) Identical anatomy of normal hematopoiesis across the skeleton: in addition to the sternum we have now determined the anatomy of hematopoiesis across the skeleton by mapping four additional bone types: tibia, humerus, vertebrae, and the lamboid sutures of the skull. In all cases we found that the anatomy of hematopoiesis *–in the steady-state–* was essentially identical to that of the sternum (which we described in the initial version of the manuscript). *These new results are now shown in **Extended Data Fig. 6**.*

2) Differential movement and clustering of hematopoietic progenitors: combining in vivo HSC transfer with live imaging of the skull we demonstrate that HSC move away from each other when in close proximity in vivo (**Fig. 2c,d**). Using live imaging of cultured cells we demonstrate that the offspring of HSC and multipotent progenitors always relocate away from each other. In contrast, the offspring of committed progenitors remains in contact (**Fig.2 q,r** and **Extended Data Figure 5**). These results explain the differences in progenitor clustering found using our ex vivo mapping and support that cell autonomous mechanisms control differential progenitor clustering at different maturation stages. *These new results are now shown in **Figures 2c,d; Fig. 2q,r** and **Extended Data Figure 5**.*

3) Stress responses vary across the skeleton: at the request of reviewer 1, we mapped the anatomy of the stress response after G-CSF. At the request of reviewer 4 we also mapped the stress responses across the skeleton. These great suggestions led us to an unexpected discovery: the bone marrow response to a hematopoietic insult varies depending on the bone examined. For example, while tibia responds to G-CSF treatment by dramatically increasing neutrophil production the sternum responds to G-CSF by shrinking neutrophil production sites and neutrophil output. Similarly, hemorrhage triggers CFU-E amplification in the tibia and sternum but not in the skull. These experiments demonstrate that the bone marrow response to hematopoietic insults varies across the skeleton, and that this response is bone and insult specific. This is a new paradigm in the field that was possible due to our ability to interrogate smaller bones that are difficult to interrogate by FACS and thus have not been routinely examined in the literature. *These new results are now shown in **Figure 4** and **Extended Data Figure 10**.*

Overall the manuscript would benefit from better clarity and rigor in presentation and validation by comparison with existing data. For example, once the authors apply their scheme for identifying various cell types, they should quantify the total number of cells of each cell type in a large section of marrow and compare that to known frequencies from published work.

We apologize to the reviewer for the lack of clarity and appreciate the detailed feedback. We have extensively edited the manuscript as suggested to increase the rigor of our presentation and writing throughout our manuscript.

To rigorously test our imaging strategies, we validated that the cell frequencies detected by FACS for each population match the frequencies detected in large marrow sections (whole mounted sternum segments). This is the established method for validation of confocal analyses in the bone marrow [*Nature* 526, 126–130 (2015), *Nature* 590, 457–462 (2021), *Blood* 136 2296-2307(2020)]. These analyses are provided in **Extended Data Fig. 1h** (for multipotent HSPC), **Extended Data Fig. 1q** (Erythropoiesis), and **Extended Data Fig. 1t** (Lymphopoiesis) and are reproduced below, the validations of the stains for myelopoiesis were published previously (reference #1). Please note that -because we use new cell surface markers and gating strategies to identify the different progenitors (**Fig. 1e**)- direct comparisons of cell frequencies with those published by others are not possible (e.g., our ESAM⁺CD117⁺CD41⁻CD48⁺CD150⁺ population contains a mix of MPP3, MPP4 and CMP). Further, we are the first to report imaging of most of these populations (which is one of the major advances of the manuscript). The exceptions are LT-HSC and CLP which have been imaged by other groups. As shown in **Extended Data Fig. 1h** below our LT-HSC frequency is 0.01% percent. This is comparable to the HSC frequency described by other groups (~0.008%) when using α -catulin^{GFP} reporter to detect HSC (*Nature*. 2015; 526(7571): 126–130.) As shown in **Extended Data Fig. 1t** below our CLP frequency is 0.04% which is the same as previously described for CLP (0.03%, *Nature*. 2021;591(7850):438-444).

Extended Data Fig. 1. **h**, Comparison of cell frequencies for the indicated cell types detected by FACS (white) or confocal imaging (orange) in the sternum when using the HSPC strategy shown in Fig. 1e (n = 9 mice for each group). **q**, Comparison of cell frequencies for the indicated 8 populations in the sternum bone marrow by FACS (white) or image (orange) (n = 3 mice for each group) when detected using the erythroid imaging strategies shown in m.o. **t**, Frequencies for the indicated B cell populations in the sternum bone marrow by FACS (white) or image (orange) (n = 3 mice for each group).

Throughout the manuscript they refer to “production lines” but no definition of this item is given, and the figure (e.g. Figure 2E) where this is supposedly demonstrated resembles more of a snake or a blob than a line. Therefore, it is unclear how the number of production lines or their cellularity is calculated for Figure 4C.

We apologize to the reviewer for not making this information clearer. We used the term “cellular production line” akin to a production assembly line in a factory because these structures recruit progenitors that differentiate into mature cells followed by recruitment of additional progenitors. It was never our intention to suggest that these organize into physical lines. To avoid confusion, we have renamed these structures as “production sites”.

We originally described how we defined and identified each type of production site by comparison with random cells in the methods. To increase the rigor and clarity of this description we now provide a step-by-step

identification for erythroid (**Extended Data Fig. 4h**), lymphoid (**Extended Data Fig. 4m**), neutrophil (**Supplementary Fig. 2f**), and DC/Ly6C^{lo} monocyte (**Supplementary Fig. 2e**) production sites (reproduced below)

Extended Data Figure 4. h, Step by step identification of erythroid production sites. In contrast to random cells, CFU-E cluster together and attach to sinusoids. Because adjacent CFU-E and erythroblasts are clonally related (see panels f, g) we consider that the center of the erythroid production site is a sinusoid decorated with 3 or more adjacent CFU-E. The mean distance of random cells to the closest CFU-E was 46.18 μm for early erythroblasts, 48.33 μm for late erythroblasts, 48.98 μm for reticulocytes, and 49.98 μm for erythrocytes (Fig. 2i). We thus considered all erythroid cells within 50 μm of the CFU-E decorated sinusoid as belonging to that production site. These analyses are then repeated through the bone to identify all production sites in the section. In the rare cases when the edges of two production sites overlap the erythroid cells were assigned to the closest of the two production sites

Extended Data Figure 4. m, Identification of lymphoid production sites. In contrast to random cells CLP cluster together with Pre-pro B and Pro B cells. The mean distance for random cells to the closest CLP was 135.5 μm for Pre-pro B, 136.1 μm for Pro B, and 161.9 μm for Pre B (Fig. 2o). Based on this we defined each production site as a 150 μm region centered around each CLP. These analyses are then repeated through the bone to identify all production sites in the section. In the rare cases when the edges of two production sites overlap the B cells were assigned to the closest of the two production sites

e

f

Supplementary Figure 2. e, Identification of Mono/Dendritic production sites. In contrast to random cells MDP cluster together with Ly6C^{lo} Monocytes and dendritic cells. The mean distance for random cells to the closest MDP was 203 μm for Ly6C^{lo} monocytes, and 207 μm for cDC. Based on this we defined each production site as Ly6C^{lo} monocytes and cDC cluster tightly, cells closer than 100 μm (half of the median distance of random cells to the MDP) to be part of the production site; These analyses are then repeated through the bone to identify all production sites in the section. f, Identification of Granulopoiesis production sites. In contrast to random cells GP cluster together with PN cells. The mean distance for random cells to the closest GP was 113 μm for PN cells. Based on this we defined each production site as cells closer than 50 μm (half of the median distance of random cells to the GP) to be part of the production site; these analyses are then repeated through the bone to identify all production sites in the section. In the rare cases when the edges of two production sites overlap the cells were assigned to the closer of the two production sites.

The authors describe some highly unexpected data in which progeny from confetti-labeled cells cluster but do not map near their immediate precursors. The explanation for this is rather perfunctory and does not address the larger question of whether it is an experimental artifact (e.g. symmetric division or loss of labeling (from progenitor cells)).

We understand that this comment relates to point #6 that we have addressed in detail below.

Finally, the authors tend to make gross generalizations about their data rather than just reporting the findings. A simple reporting of the data and demonstration of its validity would make for a stronger paper.

This was a great suggestion and we have edited the manuscript to present the data and its validity and avoid generalizations.

Specific Comments:

1) The authors use ESAM as a marker to image HSPCs in mice under different conditions including infection and aging. ESAM is known to be widely expressed in HSCs and plays a role in hematopoiesis (Sudo et al., 2012, Sudo et al. 2016); it is not clear whether ESAM is functionally important in maintaining the anatomy of the bone marrow. What are the factors contributing to the distribution of HSCs and the migration of progenitors?

This is an important question. Since ESAM is indispensable to map the different types of multipotent HSPC (**Fig. 1a-f**) we are unable to interrogate the anatomy of hematopoiesis in the absence of ESAM (please note also that ESAM knockout mice display an HSC defect during development DOI: 10.1016/j.stemcr.2019.11.002).

To investigate HSPC distribution and migration in detail we analyzed live imaging experiments of HSC via a collaboration with the laboratories of Drs. Keisuke Ito and Charles Lin. We performed follow-up analysis of microscopy-guided transplantation of single Dil- labeled, Tie2⁺ HSC in the mouse calvarium as described³². In one instance we observed that (48 hours after the initial transplant) the sole transplanted HSC had divided generating two Dil- labeled, Tie2⁺ cells that were in close proximity. Three hours later one of the daughter cells was no longer visible in the whole calvarium suggesting that it had moved away (**Extended Data Fig. 3e**). These results led us to hypothesize that HSC move away from each other when in close proximity. Follow-up analysis of single vs multiple cell transplants (5 recipients received one LT-HSC and 6 recipients received respectively 5, 5, 5, 17, 19, and 22 Tie2⁺ labeled LT-HSC, **Fig. 2c reproduced on next page**). We found that a single cell was visible in 80% of the single cell recipients (4 out of 5) and the cells were all found in near (<100µm) of the transplantation site. Only one out of 73 cells was found –as a single cell– in the mice that received multiple HSC (**Fig. 2d**). Importantly, all tracked recipients showed long-term HSC engraftment that correlated with the number of HSC transferred (**Supplementary Table 3** and³²) indicating that the absent HSC did not die or terminally differentiate. These experiments indicate that HSC move away from each other, when in close proximity, in vivo. The fact that multipotent HSPC are always found as single cells whereas lineage-committed progenitors form clusters with daughter cells (**Fig. 2a, e, m**) prompted us to investigate whether this was mediated by cell autonomous mechanisms. In live imaging analyses of cultured cells, we found that –after cell division- the offspring of HSC and MPP rapidly moved away from each other. In contrast most committed progenitors remained tightly attached after cell division (**Fig 2q, Extended Data Fig. 5, and Supplementary movies 2-6**). These results explain the differences in progenitor clustering found using our ex vivo mapping and support that cell autonomous mechanisms control differential progenitor clustering at different maturation stages.

Figure 2. c, Tie2⁺ HSC purified from actin-GFP mice were transplanted directly into the calvarial bone marrow of living mice using the approach described in reference 32 as either single cells (5 recipients received 1 cell) or multiple cells (6 recipients received respectively 5, 5, 5, 17, 19, and 22 cells). **d**, The graph reports the fraction of cells found using intravital microscopy in the whole calvarial bone marrow 24hrs following transplantation ($p^{***} = 1.96 \times 10^{-9}$, chi-square test to compare two proportions). A single cell was visible in 80% of the single cell recipients (4 out of 5) and the cells were all found in close proximity ($<100 \mu\text{m}$) of the transplantation site. Only one out of 73 cells was found [as a single cell] in multiple cell recipients.

Recipient	number of transplanted cells	Peripheral blood chimerism (week 4)
M1	1	not tracked
M2	1	1.14
M3	1	0.46
M4	1	0.23
M5	1	2.76
M6	5	2.59
M7	5	not tracked
M8	5	5.4
M9	17	not tracked
M10	19	69.4
M11	22	13.3

Supplementary Table 3. Peripheral blood engraftment in mice after microscopy-guided single or multiple HSC transplantation

Figure 2. q, Distance between daughter cells at the indicated time points after division from a cultured $n = 30$ HSC, MPP, CFU-E, or MDP and $n = 18$ GP or CLP. **r**, Percentage of daughter cells that have separated more than $50 \mu\text{m}$, 2 hours after division. ($n = 5$ wells for each indicated progenitors).

How do the authors address the possibility that ESAM-negative HSCs, which are missed by this technique, are localized closer to progenitors?

Our data demonstrates that all HSC are ESAM+ (**Fig. 1c**, reproduced to the right) and thus were not missed by the technique. Note that this agrees with previous studies that showed that all LT-HSC are uniformly ESAM+^(20,21). To rigorously confirm that our ESAM+CD117+CD41-CD48-CD150+ LT-HSC were the same as classically isolated Lin-CD117+Sca1+Flt3-CD150+ (SLAM) LT-HSC we performed limiting dilution competitive transplants and found that both populations contain identical HSC frequencies (**Extended Data Fig. 1g**). For the reviewer's convenience these panels are reproduced below:

Figure 1c, ESAM expression in indicated HSPC.

Extended Data Figure 1g, Experimental scheme and plots of extreme limiting dilution analyses showing the estimated LT- and ST-HSC frequency (solid bars) and confidence intervals (dotted lines) in the bone marrow of recipients transplanted with the indicated numbers of SLAM or ESAM LT- or ST-HSC (n=15 recipient mice in independent experiments per group and dilution except for the single cell transplanted group were n=20). Cell frequencies were calculated using Extreme Limiting Dilution Analysis.

2) The “experimental pipeline” is depicted in Figure 1A, but the details of this pipeline are not provided in the methods or the supplement. How are the populations defined? CMP are not in this figure. Are we to glean the staining schemes from the incomplete references shown in the figure e.g. “Pietras et al?”

We thank the reviewer for raising this point. We now provide a detailed explanation of the experimental pipeline in the corresponding figure legend (reproduced below). We also indicate the markers used to identify each population in **Figure 1b** and provide detailed gating strategies with complete references in **Extended Data Fig. 1a,b**.

Because of the staining strategies used (**Extended Data Fig. 1a,b**) we did not immunophenotype CMP directly. Instead, we profiled the Pre-GM which is a mixed population containing both MDP and CMP (**Extended Data Fig. 1c**). We now indicate in **Figure 1b** (reproduced below) that Pre-GM are a mixed population. We subsequently identified bona-fide CMP within the Pre-GM gate as shown in **Extended Data Fig. 1e,f**.

Figure 1. Strategies to image stepwise hematopoiesis. **a**, Experimental pipeline. Bone marrow hematopoietic progenitors were immunophenotyped by FACS and differentially expressed markers identified. When marker expression was heterogeneous the positive and negative fractions were FACS-purified and their function examined in colony forming or transplantation assays. Guided by this information we developed new staining strategies allowing detection of all cells of interest by FACS followed by validation in whole-mount imaging experiments. Validated strategies were then used to define the anatomy of hematopoiesis. **b**, Histograms showing differential expression of 35 cell surface markers across 14 HSPC types

Was IL7Ra (CD127) included as a marker for CLP identification?

Indeed, CD127 was used to identify the CLP as described in Nature 591, 438–444 (2021) –reference 8- and shown in **Extended Data Fig. 1r,s** (old Extended Data Fig. 2l,m)

Extended Data Figure 1 r-s, FACS gating strategies (r) for isolation of -and representative images (s) - of B lymphopoiesis (the Lin panel contains CD2, CD3e, CD5, CD8, CD11b, Ter119, Ly6G, IgM, and IgD).

3) It is interesting that ESAM+ Pre Meg-E differentiate more towards monocyte and neutrophil compared to ESAM- cells (Figure E1C) This should be in the main figure.

This was an excellent suggestion. Reviewer 4 further requested that we validated these findings in transplantation experiments. These transplants confirmed our original observations demonstrating that all monocyte and neutrophil engraftment potential is restricted to the ESAM positive fraction of the Pre Meg-E (Fig. 1d, and see below). Since the transplantation experiments provide a more rigorous readout of progenitor function, we placed those in the main figure (Fig. 1d, reproduced below) and kept the colony assays in Extended Data Fig. 1d.

Figure 1. d, Percentage of donor-derived cells in the peripheral blood after transplantation of the indicated progenitors (n = 4 mice). All data represent mean \pm s.e.m. Statistical analysis was performed using Two-way ANOVA, followed by Multiple unpaired t tests, P values are shown.

4)

5) Throughout the manuscript the authors refer to “production lines” but no definition of this item is given, and the figure (e.g. Figure 2E) where this is supposedly demonstrated resembles more of a snake or a blob than a line. Therefore, it is unclear how the number of production lines or their cellularity is calculated for Figure 4C.

We apologize to the reviewer for not making this information clearer. We used the term “cellular production line” akin to a production assembly line in a factory because these structures recruit progenitors that differentiate into mature cells than then leave the structure. It was never our intention to suggest that these organize into physical lines. To avoid confusion, we have renamed these as “production sites”.

We now provide step by step identification of erythroid (Extended Data Fig. 4h), lymphoid (Extended Data Fig. 4m), neutrophil (Supplementary Fig. 2f), and DC/Ly6C^{lo} monocyte (Supplementary Fig. 2e) production sites (please see these panels in response to the overall critiques above).

6) Re:

“Unexpectedly, confetti-labeled MkP did not map near megakaryocytes labeled by the same color (Extended Data Fig. 3f, g). These results support a model where megakaryocyte progenitors are recruited towards active sites of megakaryopoiesis and terminally differentiate to generate 2-4 mature megakaryocytes.” This is an unexpected finding and the proposed explanatory model is not at all clear. How do the authors exclude other possibilities such as disappearance of the originating MkP due to symmetric differentiation or loss of labeling of the relevant MkP. The same criticism applies to the discussion of CLP /Pre-Pro B, Pro B, Pre B cells: Unexpectedly, clonal fate mapping demonstrated that confetti-labeled CLP do not map near Pre-pro B, Pro B,

or Pre B labeled with the same fluorescence protein (Extended Data Fig. 4h). These results indicate that the location of common lymphoid progenitors identifies oligoclonal B cell production lines near arterioles.

We are very grateful to the reviewer for raising this point. In the *Ubc-cre-ERT2: confetti model*, cre-mediated recombination leads to irreversible expression of one out of four possible fluorescent proteins [GFP, CFP, YFP, or RFP, reference 34 doi:10.1016/j.cell.2010.09.016 (2010)]. Because recombination efficiency is approximately 7% we can only conclusively determine if the clusters/production sites that we observe are monoclonal (when a cluster contains confetti-labeled cells and all the cells in the cluster are labeled by the same fluorescent tag) or oligoclonal (when a cluster contains confetti-labeled cells there are labeled with more than two fluorescent tags). Since we were unable to distinguish clear clonal relationships between MkP and megakaryocytes we have removed these panels –and corresponding text– from the manuscript to avoid confusion (**Old Extended Data Fig 3.f-g**).

In the case of CLP and B cell production sites the confetti experiments conclusively demonstrate that these structures are oligoclonal (**New Extended Data Fig. 4I**, reproduced below). Based on this we have edited the text as follows (Lines 208-217): “Most Pre-pro B, Pro B, and Pre B cells are selectively enriched near CLP, forming loose clusters (2 ± 1 Pre-pro B, 3 ± 2 Pro B, and 16 ± 8 Pre B within $150 \mu\text{m}$ of each CLP). The more mature cells located farther away from the CLP suggesting movement away from the cluster (**Fig. 2m-o**). *Ubc-creERT2:confetti* fate mapping showed that these clusters were oligoclonal. We found differentiating cells labeled in the same confetti color as the CLP but these did not map closer to the CLP than expected from random cells (**Extended Data Fig. 4I**). Taken together these experiments suggests that daughter cells moved away from the CLP after division but remain associated in loose clusters. This is in agreement with live imaging studies showing that Pre-B cells are highly motile³⁵. We propose that these clusters of CLP and differentiating B cells are oligoclonal B cell production sites (see **Extended Data Fig. 4m** for formal definition) near arterioles”.

Extended Data Fig. 4I. Representative image and distance analyses of confetti-labeled CLP or B cell precursors ($B220^+Lin^-$ cells, the Lin panel contains CD2, CD3e, CD5, CD8, CD11b, Ter119, Ly6G, IgM, and IgD, containing all Pre-pro B, Pro B, and Pre B cells) to the closest B cell precursors labeled in the same color ($n = 11$ confetti-labeled CLP, 23 confetti-labelled B cell precursors in 3 sternum segments from 3 tamoxifen-treated confetti mice).

7) In general the figures are not at all easy to follow and would benefit from fewer panels shown in a larger format. In one particularly important example, Figure 2E is supposed to show an erythroid “production line” The distribution of the cells hardly seems to resemble a line but rather shows a snaking pattern of clustered cells. How is this “production line” defined? What is the evidence that these cells originate from the same progenitor? How does one distinguish one production line from a neighboring one?

As suggested, we have strived to make most immunofluorescence panels larger. We have also simplified Figures 2, 3, and 4. Please see response to point #5 regarding how each production site is defined. The confetti experiments (**Extended Data Fig. 4f** for erythroid production sites and **Extended Data 4i** for lymphoid production sites) demonstrate that production sites are oligoclonal and thus are generated by more than one progenitor.

Distinguishing the production sites from each other is trivial as the production sites for B cells, neutrophils, and monocytes/dendritic cells are all centered around a lineage-committed progenitor (CLP, GP or MDP, **Fig. 2m,n** and **Supplementary Figure 2** and ¹⁾) and these are far enough from each other that they minimally overlap (**Extended Data Fig. 4n**). Erythroid production sites are also easily distinguished from each other as they are defined by a cluster of CFU-E attached to a sinusoid (**Fig. 2e,f**, **Extended Data Figure 4h** and **Supplementary Video 1**). A neighboring erythroid production site is simply attached to a different sinusoid. In the very rare cases when the edges of two production sites overlap the erythroid cells were assigned to the closest of the two production sites.

8) The confetti data in Figure 3f is an important point but is so small that the reader must take the authors' word for it. The representative yellow Mkp does have other yellow cells around it but those are not noted in the diagram.

Since we were unable to distinguish clear clonal relationships between Mkp and adjacent megakaryocytes we have removed these panels from the manuscript to avoid confusion (**Old Extended Data Fig 3.f-g**). Indeed, the yellow (YFP+) Mkp had other yellow cells around it but these were not stained with CD41, CD117, or ESAM indicating that they were not HSC, MPP, or Pre-MegE nor megakaryocytes (this panel is reproduced below as **reviewer Figure 3**). This indicates that they are not upstream or downstream of the yellow Mkp.

Reviewer Fig. 3 corresponding to OLD Figure 3f. Representative image showing confetti-labeled megakaryocyte progenitor (MkP) and megakaryocyte (MK) in tamoxifen-treated confetti mice.

9) In general the results are written inappropriately in the present tense and with broad overgeneralization. For example, the authors state that “Infection causes minimal increases in myeloid production lines.” The data are hardly sufficient to support such a broad statement. One might say, “In this model, minimal changes in myeloid production were seen after infection”

We have edited the manuscript in the style suggested.

As another example of over-generalization, the authors state that “all production lines for a given lineage are synchronized as they simultaneously expand or contract in response to insults.” This is an overstatement as the timing of expansion and contraction has not been measured.

We thank the reviewer for this suggestion. We have edited the manuscript to avoid over-generalization. To assess whether production sites in the same lineage expand in a synchronized manner we performed kinetics analyses for all insults. We found that the number of cells in the production sites changed following kinetics that were remarkably consistent for all the time points analyzed (within the same bone). For the reviewer convenience these panels (**Extended Data Fig. 8**) are reproduced on the next page:

10) In evaluating the effects of aging, the authors state that “number of production lines was maintained or slightly increased when compared to 2-month-old mice (Fig. 4c).”

How are “production lines” defined? How was cellularity per cluster defined?

Please see response to point #5 regarding how each production site is identified. Cellularity per cluster was quantified by counting the number of cells for each lineage at the indicated distances from the progenitor (e.g. 150µm around a CLP, Extended Data Fig. 4m) or vessel (e.g. 50µm around a sinusoid decorated by CFU-E, Extended Data Fig. 4h)

Phlebotomy: **increased** erythroid, **reduced** lymphoid and no changes in mono/DC or neutrophil production site output

L. monocytogenes infection: **Reduced** erythroid, **reduced** lymphoid, small changes in neutrophil, and **increased** dendritic cell production site output

G-CSF: **Reduced** erythroid, **reduced** lymphoid, **reduced** neutrophil, and **reduced** dendritic cell production site output

Extended Data Figure 8. See legend on next page

Extended Data Figure 8. Changes in the microarchitecture of the production sites in response to acute hematopoietic stress. **a-b**, After phlebotomy the erythroid production sites respond with increases in CFU-E, and early and late erythroblasts and reductions in reticulocytes and red blood cells. The images (a) show the changes in cell content in representative erythroid production sites at the indicated time points after phlebotomy. The dot plots show the number of indicated cells per production site. CFU-E (b, n = 65, 99, 58, and 40 production sites for days 0, 2, 8, and 20, 3 sternum segments from 3 mice per time point) and indicated erythroid cells (c-d, n = 5 randomly selected production sites) in erythroid production sites after phlebotomy. **b**, Representative images and number of cells per B cell site at the indicated time points after phlebotomy (n=10 randomly selected production sites in 3 sternum segments from 3 mice for each time point). **c**, After infection erythroid production sites show reduced output that is progressively restored. The image (g) shows a representative erythroid production site after infection. The dot plots (h) show the number of CFU-E (n = 65, 45, and 60 production sites for days 0, 6, 20, three sternum segments from 3 mice per time point) and indicated erythroid cells (n = 5 randomly selected production sites for each time point, in 3 sternum segments from 3 mice) from control or *L. monocytogenes*-infected mice. **d**, Representative image and number of cells per B cell production site from control or *L. monocytogenes*-infected mice after infection (n = 12 randomly selected production sites in 3 sternum segments from 3 mice for each time point). **e**, Representative images and number of cells showing increased dendritic cell and reduced monocyte production in DC/Ly6C^{lo} sites after *L. monocytogenes* infection mice (n = 49, 35, and 29 randomly selected production sites for days 0, 6, 20, in 3-5 sternum segments from 3 mice per time point). **f**, as (e) but for granulocyte production sites (n = 96, 88, and 50 randomly selected production sites in 3-5 sternum segments from 3-4 mice per time point). **g-i**, representative images and number of cells per production site demonstrating that G-CSF treatment leads to reductions in the output of erythroid (g, n= 6, 6, 6), lymphoid (h, n=6, 6, 6), mono/DC (i, n= 15, 9, 14), and neutrophil (j, n=15, 11, 15) production sites followed by a return to homeostasis at day 30. Statistical differences were calculated using one-way ANOVA, Dunnett's multiple comparisons test; P values are shown.

Referee #3 (Remarks to the Author):

Wu et al. used new combinations of antibodies to image hematopoietic stem cells, multipotent progenitors and various kinds of restricted hematopoietic progenitors. Along with the prior study published by this group (Zhang et al. Nature 2021), they are performing more complex imaging of the bone marrow than other groups, using more markers to image more cell populations. In this study, they mapped various aspects of blood cell production, including multipotent cells, megakaryocyte, erythroid and lymphoid progenitors. Some of what they reported would be expected from prior studies but the suggestion that multipotent daughter cells migrate far away from their mother cells while differentiating cells remain in clonal “production lines” is new and interesting. The authors conclude that erythropoiesis takes place in production lines adjacent to sinusoids and B lymphopoiesis takes place in production lines near arterioles. Prior studies have localized these progenitors to these blood vessels but Wu et al. add to the prior studies by showing that the differentiated progeny of these restricted progenitors undergo local clonal expansion, in contrast to multipotent cells. The use of the confetti reporter to distinguish different clones is a notable innovation that helps provide insight into what cells remain adjacent to sibling cells and what cells migrate away. Finally, they report that stresses don't seem to alter the overall anatomy of the bone marrow but increase or decrease the expansion of differentiating clones next to blood vessels.

We are deeply grateful to the reviewer for the positive review and extremely helpful comments. We have addressed all the critiques in detail in the point-by-point response below. We would also like to highlight three new major pieces of data that further increase the rigor and novelty of our findings. These are:

1) **Identical anatomy of normal hematopoiesis across the skeleton:** in addition to the sternum we have now determined the anatomy of hematopoiesis across the skeleton by mapping four additional bone types: tibia, humerus, vertebrae, and the lamboid sutures of the skull. In all cases we found that the anatomy of hematopoiesis *–in the steady-state–* was essentially identical to that of the sternum (which we described in the initial version of the manuscript). *These new results are now shown in **Extended Data Fig. 6**.*

2) **Differential movement and clustering of hematopoietic progenitors:** combining in vivo HSC transfer with live imaging of the skull we demonstrate that HSC move away from each other when in close proximity in vivo (**Fig. 2c,d**). Using live imaging of cultured cells we demonstrate that the offspring of HSC and multipotent progenitors always relocate away from each other. In contrast, the offspring of committed progenitors remains in contact (**Fig.2 q,r** and **Extended Data Figure 5**). These results explain the differences in progenitor clustering found using our ex vivo mapping and support that cell autonomous mechanisms control differential progenitor clustering at different maturation stages. *These new results are now shown in **Figures 2c,d; Fig. 2q,r** and **Extended Data Figure 5**.*

3) **Stress responses vary across the skeleton:** at the request of reviewer 1, we mapped the anatomy of the stress response after G-CSF. At the request of reviewer 4 we also mapped the stress responses across the skeleton. These great suggestions led us to an unexpected discovery: the bone marrow response to a hematopoietic insult varies depending on the bone examined. For example, while tibia responds to G-CSF treatment by dramatically increasing neutrophil production the sternum responds to G-CSF by shrinking neutrophil production sites and neutrophil output. Similarly, hemorrhage triggers CFU-E amplification in the tibia and sternum but not in the skull. These experiments demonstrate that the bone marrow response to hematopoietic insults varies across the skeleton, and that this response is bone and insult specific. This is a new paradigm in the field that was possible due to our ability to interrogate smaller bones that are difficult to

examine by FACS and thus have not been routinely examined in the literature. These new results are now shown in Figure 4 and Extended Data Figure 10.

1. On lines 90-91 the authors conclude that “the eLT-HSC and eST-HSC gates are highly enriched in LT-HSC or ST-HSC (Fig. 1e) and have identical activity in competitive transplants as SLAM HSC”. But the relevant figures show that the authors transplanted hundreds or thousands of cells in the reconstitution assays. Using such large numbers of cells, it would be impossible to know if there are modest differences in purity between the markers they use and more widely used marker combinations. I suspect the purity of the cell populations they characterized are pretty good, but formally, one would have to perform single cell (or near single cell) assays to know how pure these cell populations are. Many prior studies have done this, performing single cell reconstitution assays in irradiated mice and sorting single cells into culture to determine what percentage of cells have the expected activities. It would be helpful if the authors could provide more information on the purity of the cell populations they imaged.

This is a very important point. We now have rigorously compared the purity of LT-HSC and ST-HSC purified using either SLAM or ESAM-based gating strategies by using limiting dilution assays. In these experiments we found no differences between the ESAM- or SLAM-purified HSC (**Extended Data Fig. 1g**). For the reviewer’s convenience these panels are reproduced below:

2. It was hard to read the first section of the paper because the combinations of markers that were used to identify each cell population didn’t seem to be listed in a simple way. Readers seem to have to figure it out by scanning through lots of flow cytometry plots in Figure 1. It would be useful to add a table that lists each cell population that was imaged in vivo, whether it is thought to be pure or a mixed population of progenitors (i.e. what it contains) and the markers used to identify the cells during imaging.

We are very grateful for this excellent suggestion. This is now shown as a scheme in **Figure 1g-j** and as a table in **Supplementary Table 2**.

Figure 1. g-j Scheme summarizing expression of cell surface markers used to interrogate multipotent cells (g), erythropoiesis (h), B lymphopoiesis (i) and myelopoiesis (j).

3. The observation that stem cells are scattered throughout the bone marrow as single cells has been published in several prior studies. Based largely on this observation, Wu et al. assert that “multipotent progenitors move away from their parent and sister cells and migrate hundreds of micrometers as they differentiate”; however, it’s not clear how they can conclude that. It seems like there are alternative explanations for the observation that HSCs are present as single cells in the bone marrow. One possibility is that daughter cells are released into circulation (rather than migrating through the bone marrow). Another possibility is that at least one of the daughter cells dies, or changes its markers by maturing out of the HSC/multipotent progenitor pool. Do the authors have data to rule out these possibilities? It may not be possible to test whether daughter cells enter circulation but it’s a possibility that should be acknowledged.

We are grateful to the reviewer for raising this important point and apologize for the lack of clarity. Please note that our initial assertion was based on the observation that ST-HSC, MPP2, and MPP3 (which we image here for the first time) do not map near LT-HSC and instead are found hundreds of microns away.

Our previous results (**Fig. 2a,b**) demonstrated that all HSPC are found as single cells in the bone marrow. Since, after cell division, daughter cells are necessarily adjacent, the results also indicated that the offspring of HSPC were either released into the circulation, differentiated into more mature cells that are not detected in the HSPC stain, or moved away from each other. In the steady-state a minute fraction of total HSPC is released into the circulation^{27,28} indicating that this mechanism is not sufficient to explain lack of HSPC adjacency in the bone marrow. Similarly, *in vivo* barcoding showed that MPP can extensively self-renew *in vivo* for long periods of time indicating that the offspring of HSPC divisions does not always terminally differentiate²⁹. Intravital imaging studies support different degrees of HSC motility in the marrow⁹⁻¹². Because most HSC are found as single cell in the marrow (^{30,31} and **Fig. 2a**) the mobility of HSC when adjacent to other HSC/MPP has not been examined.

To investigate HSPC distribution and migration in detail we analyzed live imaging experiments of HSC via a collaboration with the laboratories of Drs. Keisuke Ito and Charles Lin. We performed follow-up analysis of the single cell microscopy-guided transplantation of single Dil- labeled, Tie2⁺ HSC in the mouse calvarium as described³². In one instance we observed that (48 hours after the initial transplant) the sole transplanted HSC had divided generating two Dil- labeled, Tie2⁺ cells that were in close proximity. Three hours later one of the daughter cells was no longer visible in the whole calvarium suggesting that it had moved away (**Extended**

Data Fig. 3e). These results led us to hypothesize that HSC move away from each other when in close proximity. Follow-up analysis of single vs multiple cell transplants (5 recipients received one LT-HSC and 6 recipients received respectively 5, 5, 5, 17, 19, and 22 Tie2+ labeled LT-HSC, **Fig. 2c reproduced on next page**). We found that a single cell was visible in 80% of the single cell recipients (4 out of 5) and the cells were all found in near (<100µm) of the transplantation site. Only one out of 73 cells was found –as a single cell– in the mice that received multiple HSC (**Fig. 2d**). Importantly, all tracked recipients showed long-term HSC engraftment that correlated with the number of HSC transferred (**Supplementary Table 3** and ³²) indicating that the absent HSC did not die or terminally differentiate. These experiments indicate that HSC move away from each other, when in close proximity, in vivo. The fact that multipotent HSPC are always found as single cells whereas lineage-committed progenitors form clusters with daughter cells (**Fig. 2a, e, m**) prompted us to investigate whether this was mediated by cell autonomous mechanisms. In live imaging analyses of cultured cells, we found that –after cell division- the offspring of HSC and MPP rapidly moved away from each other. In contrast most committed progenitors remained tightly attached after cell division (**Fig 2q, Extended Data Fig. 5, and Supplementary movies 2-6**). These results explain the differences in progenitor clustering found using our ex vivo mapping and support that cell autonomous mechanisms control differential progenitor clustering at different maturation stages.

4. The authors conclude “LT-HSC, ST-HSC, and other multipotent progenitors do not occupy the same niches”. Again, it’s not clear how the authors could know that, or perhaps, what exactly they mean. Even if we accept that HSC daughter cells migrate hundreds of microns from the site of division, perhaps they are migrating to from one specialized microenvironment that is designed to sustain multipotent cells to another specialized microenvironment that is designed to sustain multipotent cells. Based on prior studies from this and other groups, there seem to be different microenvironments around blood vessels that are specialized for the maintenance of different kinds of stem and progenitor cells. If so, maybe there is a single kind of environment that is specialized to maintain LT-HSCs, ST-HSC, and multipotent progenitors, distributed in different locations around the sinusoids throughout the bone marrow, and HSCs/multipotent daughter cells migrate among these specialized sites when they divide?

We are grateful for this comment and apologize for the lack of clarity. Our intention was to indicate that HSC and downstream cells are not directly regulated by the same adjacent cells. Our interpretation is the same as the reviewer’s: we believe that there is a common microenvironment - far from arterioles and the endosteum, and enriched in megakaryocytes (**Fig. 2a,b and Extended Data Fig. 3a-c**) through which the multi/oligopotent HSPC move after cell division.

We have corrected the text as follows: “These results indicate that LT-HSC and other multipotent progenitors don’t share a physical location and are thus not regulated by the same adjacent stromal cells”.

Figure 2. c, Tie2⁺ HSC purified from actin-GFP mice were transplanted directly into the calvarial bone marrow of living mice using the approach described in reference 32 as either single cells (5 recipients received 1 cell) or multiple cells (6 recipients received respectively 5, 5, 5, 17, 19, and 22 cells). **d**, The graph reports the fraction of cells found using intravital microscopy in the whole calvarial bone marrow 24hrs following transplantation ($p^{***} = 1.96 \times 10^{-9}$, chi-square test to compare two proportions). A single cell was visible in 80% of the single cell recipients (4 out of 5) and the cells were all found in close proximity ($<100 \mu\text{m}$) of the transplantation site. Only one out of 73 cells was found [as a single cell] in multiple cell recipients.

Recipient	number of transplanted cells	Peripheral blood chimerism (week 4)
M1	1	not tracked
M2	1	1.14
M3	1	0.46
M4	1	0.23
M5	1	2.76
M6	5	2.59
M7	5	not tracked
M8	5	5.4
M9	17	not tracked
M10	19	69.4
M11	22	13.3

Supplementary Table 3. Peripheral blood engraftment in mice after microscopy-guided single or multiple HSC transplantation

Figure 2. q, Distance between daughter cells at the indicated time points after division from a cultured $n = 30$ HSC, MPP, CFU-E, or MDP and $n = 18$ GP or CLP. **r**, Percentage of daughter cells that have separated more than $50 \mu\text{m}$, 2 hours after division. ($n = 5$ wells for each indicated progenitors).

5. A prior study concluded that at least some B lineage progenitors localize to sinusoids (Immunity 20:707, 2004) while Wu et al., in agreement with another prior study, conclude that lymphoid progenitors are around arterioles. These studies do not necessarily contradict each other as they may each be imaging different lymphoid progenitors. However, it would be helpful if Wu et al. could address the question of whether they think all lymphoid progenitors localize to arterioles or whether they see some evidence for a subset of lymphoid progenitors that is around sinusoids.

We carefully read the *Immunity* manuscript cited by the reviewer. We were unable to find any data in that manuscript examining B lineage progenitor localization to sinusoids. We believe that the reviewer might be referring to a study by Joao Pereira and Jason Cyster (*Nat Immunol.* 2009 Apr; 10(4): 403–411) showing that immature B cells form clusters inside bone marrow sinusoids. In this manuscript we have not imaged the localization of Immature B cells.

To address the reviewer question in detail we compared the distribution of CLP, Pre-pro B, Pro B and PreB cells to the closest sinusoid and arteriole. We found that all of these populations were farther than random cells from sinusoids and closer than random cells to arterioles (Reviewer Fig. 4 below). These results support that arterioles are the main site for B cell production.

Reviewer Fig. 4. Representative image and quantification demonstrating the Pre pro B, Pro B, and Pre B are selectively enriched near arterioles but not sinusoids. n = 38 CLP, 32 Pre proB, 16 Pro B and 53 Pre pro B randomly selected from 4 sternum segments from two mice.

6. The authors conclude that in aged mice “lymphoid- and myeloid-biased LT-HSC localize much closer to other LT-HSC than predicted... leading in some cases to the emergence of loose clusters of myeloid- and lymphoid-biased LT-HSC”. Can the authors be more precise about what they mean here? Are lymphoid biased HSCs clustered with other lymphoid biased HSCs (and myeloid with myeloid) or do HSCs cluster together irrespective of whether they are myeloid or lymphoid biased? It’s an important detail because lymphoid biased HSCs probably arise from myeloid biased HSCs only after they have undergone multiple rounds of division.

We found that aged mice have rare LT-HSC clusters and that these contain both lymphoid-biased and myeloid biased HSC (Now **Supplementary Figure 7**). To make this point clearer the text now reads: “note rare clusters of 2-4 MPP2/MyE after infection and rare clusters containing both CD41⁻ and CD41⁺ LT-HSC in old age”

Referee #4 (Remarks to the Author):

The manuscript “Resilient anatomy and local microplasticity of naïve and stress hematopoiesis” by Wu, Zhang et al. describes a new strategy to image long term hematopoietic stem cells (LT), multipotent progenitors, megakaryocyte progenitors, erythroid progenitors, and lymphoid progenitors in their bone marrow niche. The authors use the novel imaging strategy to describe the anatomy of murine hematopoiesis in steady state and after acute stress and upon aging. The data show distinct microanatomical areas of selective hematopoietic production lines. According to the data, the different production lines are resilient upon acute stress and aging, while the anatomy of the bone marrow is temporarily (in case of acute stress) or permanently (upon aging) remodeled in a cell- and lineage-specific manner.

The manuscript contains a vast amount of data and high-quality images that show with excellent resolution and very elegant graphical support the anatomy of the bone marrow in the murine sternum. The overall dataset is in its nature quite descriptive. The imaging technology is absolutely novel and qualitatively impressive. However, the major biological findings are largely anticipated by previous publications. Overall, the data constitutes mainly an in-depth high-quality confirmation of previous observations. There are a few points where the authors offer an interpretation of their dataset that could expand our knowledge on the biology of the hematopoietic system. However, these claims either require to be explained and phrased more clearly or deserve to be corroborated by additional mechanistic data.

We are deeply grateful to the reviewer for the positive review and extremely helpful comments that have led us to new biology. We have addressed all of them in detail in the point by point response below. However, we would like to first directly address the critique that “the major biological findings are anticipated by previous publications”. We respectfully –but strongly- disagree with this statement. We believe that the reviewer has missed some of the key innovations of this manuscript. Specifically:

1. We develop methods to image for the first time (and simultaneously) ST-HSC, MPP2, MPP3 and MPP4, Pre-MegE, and Pre-CFU-E. Further, we demonstrate for the first-time stepwise visualization of all steps of erythropoiesis and lymphopoiesis. Thus, we provide the field with the tools to visualize blood cell production in situ. This has been a major gap in the field for the last 40 years.

2. Additionally, we define the anatomy of normal hematopoiesis for the first time. We show that it is characterized by spatial segregation or multipotent HSPC hundreds of microns away from each other (but with selective localization near megakaryocytes). Lineage committed progenitors then localize near specific vessels where they are serially recruited to lineage-specific production sites where they form clusters with their differentiating progeny. These production sites have unique spatial and clonal architectures that we define here for the first time. This anatomy is completely unexpected as prior to our findings it was not known whether HSC colocalized with more differentiated cells (eg., doi:10.1016/j.stem.2018.11.022; doi: 10.1016/j.devcel.2021.05.018; doi: 10.1038/s41580-019-0103-9); whether erythroid or lymphoid cell production took place in specialized regions of the bone marrow; or whether production of the major blood lineages overlapped in time and space. **In new live imaging experiments** (see point “b” below) we demonstrate 2.1) that HSC move away from each other, when in close proximity, in vivo. 2.2) And that differential clustering of multipotent HSPC vs committed progenitors is mediated by a cell autonomous program.

3. We also define the anatomy of bone marrow responses to acute insults and old age for the first time. We show that the overall anatomy of the bone marrow is extremely resilient to insults as it was maintained in all stress conditions examined. More importantly, we show that changes in the number and output of the production sites mediates hematopoietic plasticity and allow the bone marrow to adjust blood cell production. This indicated that stress hematopoiesis uses the same structures as steady-state hematopoiesis for generating blood. This is completely unexpected because prior studies proposed specialized structures (e.g. GMP clusters doi: 10.1038/nature2169, and hemospheres DOI: 10.1038/emboj.2012.308) that mediated the bone marrow response to stress. **In new experiments using confetti mice** (see response to point #15) we demonstrate that the increased output of the production sites is mediated by increased progenitor recruitment to the site. Further, we show that that all production sites for a given lineage are synchronized as they simultaneously expand or contract in response to insults (**Extended Data Figure 8**).

We would also like to highlight three new major pieces of data that further increase the rigor and novelty of our findings. These are:

a) **Identical anatomy of normal hematopoiesis across the skeleton:** as requested, in addition to the sternum, we have now determined the anatomy of hematopoiesis across the skeleton by mapping four additional bone types: tibia, humerus, vertebrae, and the lamboid sutures of the skull. In all cases we found that the anatomy of hematopoiesis *–in the steady-state–* was essentially identical to that of the sternum (which we described in the initial version of the manuscript). *These new results are now shown in Extended Data Fig. 6.*

b) **Differential movement and clustering of hematopoietic progenitors:** combining in vivo HSC transfer with live imaging of the skull we demonstrate that HSC move away from each other when in close proximity in vivo (**Fig. 2c,d**). Using live imaging of cultured cells we demonstrate that the offspring of HSC and multipotent progenitors always relocate away from each other. In contrast, the offspring of committed progenitors remains in contact (**Fig.2 q,r** and **Extended Data Figure 5**). These results explain the differences in progenitor clustering found using our ex vivo mapping and support that cell autonomous mechanisms control differential progenitor clustering at different maturation stages. *These new results are now shown in Figures 2c,d; Fig. 2q,r and Extended Data Figure 5.*

c) **Stress responses vary across the skeleton:** at the request of reviewer 1, we mapped the anatomy of the stress response after G-CSF. At your request we also mapped the stress responses across the skeleton. These great suggestions led us to an unexpected discovery: the bone marrow response to a hematopoietic insult varies depending on the bone examined. For example, while tibia responds to G-CSF treatment by dramatically increasing neutrophil production the sternum responds to G-CSF by shrinking neutrophil production sites and neutrophil output. Similarly, hemorrhage triggers CFU-E amplification in the tibia and sternum but not in the skull. These experiments demonstrate that the bone marrow response to hematopoietic insults varies across the skeleton, and that this response is bone and insult specific. This is a new paradigm in the field that was made possible due to our ability to image smaller bones that are difficult to interrogate by FACS and thus have not been routinely examined in the literature. *These new results are now shown in Figure 4 and Extended Data Figure 10.*

Major general points:

1) Was only the sternum used throughout the analyses? Why? Would the long bones (femurs, tibiae) or the calvarium show any difference? Whole-mount histology has been described in several type of bones. The validation of the differentiation lines' localization in different bones would corroborate the findings. In line with this point, also vascular and differentiation lines alteration upon stress and aging should be verified in other bones.

We are deeply grateful to the reviewer for suggesting these experiments that have greatly improved our manuscript. We originally chose the sternum because it is the easiest bone to image in whole mount analyses. We have now determined the anatomy of hematopoiesis across the skeleton by mapping four additional bone types: tibia, humerus, vertebrae, and interparietal bone (skull). In all cases we found that the anatomy of hematopoiesis in the steady-state was essentially identical to that of the sternum (**Extended Data Fig. 6**, reproduced on next page).

As requested, we also examined the anatomy of stress responses in different bones. These experiments were specially challenging because performing 4 different stains (HSPC, erythropoiesis, lymphopoiesis, myelopoiesis) in 5 different bones (humerus, tibia, sternum, vertebrae, and skull) in response to four insults (phlebotomy, *L. monocytogenes* infection, G-CSF, and aging) across multiple mice would have required ~1,800 hours of microscopy imaging acquisition and ~5,000 hours of image analyses. To overcome this we decided to focus on the bone marrow response to G-CSF across the skeleton. This is because FACS analyses of long bones confirmed increased granulopoiesis (**Fig. 3a** and **Supplementary Fig. 5**) but imaging analyses of sternum showed dramatic reductions in neutrophil production sites and neutrophil production (**Fig. 3g,h**). This led us to hypothesize that stress responses vary across the skeleton. To test this hypothesis, we imaged granulopoiesis in the sternum and tibia after G-CSF. While the tibia responded to G-CSF by increasing neutrophil output the sternum responded by shutting down neutrophil production leading to loss of neutrophil production sites (**Fig. 4a-c**, please see figure on page 41 of this document). Both bones showed comparable anatomy in saline-treated mice (**Fig. 4a-b**). The tibia and sternum displayed similar suppression of erythropoiesis, monopoiesis, and lymphopoiesis indicating that both bones are equally exposed to G-CSF (**Fig. 4d**). To determine whether this phenomenon extended to other insults we induced phlebotomy and examined the response across the skeleton via FACS and imaging. Phlebotomy caused a powerful expansion in erythroid production site numbers and output in the sternum, tibia, vertebrae, and humerus (**Fig. 4e-f** and **Extended Data Fig. 10a,b**). However, we did not detect changes in erythroid production site numbers or output in the skull even though phlebotomy-induced reductions in lymphopoiesis were similar in all bones indicating that the phlebotomy was sensed by hematopoietic cells in the skull (**Fig. 4f-g** and **Extended Data Fig. 10c**). These experiments demonstrate that the response of bone marrow production sites to systemic insults is bone- and insult-specific and uncover unprecedented heterogeneity of stress responses across the skeleton.

Extended Data Figure 6. The anatomy of hematopoiesis in the steady-state is maintained across the skeleton. **a**, To determine whether the anatomy of hematopoiesis was comparable across the skeleton we imaged and compared hematopoiesis in the sternum, tibia, humerus, L5 vertebrae, and the lamboid sutures of the skull. In all bones we observed long distance separation of multipotent stem and progenitors, selective recruitment of committed progenitors to vessels, and terminal maturation in lineage specific production sites. Since bone sizes are dramatically different, the maps show the distribution of the indicated cells in areas of similar size for each bone. **b-f**, We noted that some cell types display large variations in frequencies between bones –e.g. LT-HSC were less abundant in the tibia– leading to changes in distances to closest cells. The graphs show the distances from each LT-HSC (**b**, n= 19, 21, 25, 23 and 26), CFU-E (**c**, n=221, 501, 495,139 and 548), CLP (**d**, n=28, 13, 50, 15 and 22), GP (**e**, n=79, 51, 51, 46 and 46) or MDP (**f**, n=29, 29, 30, 19 and 23) to the closest indicated cell in the five bones examined.

Figure 4. The hematopoietic response to stress varies across the skeleton. **a**, maps showing the distribution of granulocyte progenitors (GP, pink) and preneutrophils (yellow) in whole-mounted sternum and tibia treated with saline (S) or G-CSF (G, 250mg/kg/day) for 4 days. **b**, after G-CSF the sternum has fewer GP and the neutrophil sites disaggregate ($n = 3$ sternum or tibia from 3 mice per treatment). **c**, **d**, Number of indicated cells in sternum and tibia quantified by FACS ($n = 4$ mice in four independent experiments). **e**, maps showing the distribution of CFU-E (red, terminal erythroid progenitors) in whole-mounted sternum, tibia, or lambdoid suture of the skull bone in the steady-state or 2 days after phlebotomy. **f**, Quantification of erythroid production site frequency (upper panel $n = 3$ bones each from 3 mice) and number of CFU-E per site ($n = 10$ randomly selected production sites from 3 bones each from 3 mice) in control (ctl) or 2 days after phlebotomy (Phl). Statistical differences were calculated using two way Student's T-tests.

2) Line 77-78: “We thus defined ESAM+ Pre Meg-E as myeloerythroid (MyE) progenitors”. In Extended figure 1c, ESAM+ MegE are contributing much less to erythroid cells in comparison to the ESAM- counterpart. The data suggest that ESAM+ Meg-E give rise to myeloid cells while mainly ESAM- Meg-E differentiate to erythroid cells. This aspect should be clarified by demonstrating in Tx that ESAM+ Meg-E are sufficient to produce most erythroid cells, or the analysis and interpretation should be revised accordingly.

We performed these transplants as requested and confirmed that ESAM+ PreMegE give rise to myeloid and erythroid cells whereas ESAM- Pre-MegE generate red blood cells. These transplants are shown in new Figure 1d and reproduced below:

Figure 1. d. Percentage of donor-derived cells in the peripheral blood after transplantation of the indicated progenitors (n = 4 mice). All data represent mean \pm s.e.m. Statistical analysis was performed using Two-way ANOVA, followed by Multiple unpaired t tests, P values are shown.

3) Page 5-6, lines 117-123: “all stem cells and multipotent progenitors are found as single cells, evenly distributed through the bone marrow, and are no closer to each other than predicted from random simulations. This spatial segregation is not due to differential interaction with the microenvironment as most multipotent progenitors showed localization similar to that of LT-HSC: closer than random to megakaryocytes but farther than random from arterioles”. How is the random distribution calculated? Many conclusions throughout the manuscript are drawn related to the random distribution, and it would help the reader having this aspect clearly described in the main text.

We apologize for not making this point clearer. For random simulations we stained with CD45 and Ter119 antibodies (to detect all hematopoietic cells) and ESAM to detect sinusoids, arterioles and megakaryocytes. These images were processed to segment each cell and structure and to obtain the X, Y, and Z coordinates of all hematopoietic cells (**Extended Data Fig. 2a reproduced on the next page**). To compare observed (**Extended Data Fig. 2b**) with random distributions (**Extended Data Fig. 2c**) we used Research Randomizer software to randomly select coordinates and place dots representing each type of hematopoietic cell found in each stain in these coordinates. These “random” hematopoietic cells were placed at the exact same frequencies as those observed with the corresponding stain. We then measured the distance between these random cells (taking into account the average radius for each cell type) or with vessels. Each random simulation was repeated a hundred times. As requested, we made this approach clearer in the methods section and revamped **Extended Data Fig. 2** (reproduced below) to better explain how the random simulations are performed.

Extended Data Figure 2. Strategies for automatic segmentation and random simulations. **a**, Experimental workflow for automatic cell segmentation. Fluorescence in the individual channels is merged, converted to 12-bit data, and intensities normalized. The “3D darkspot detection” algorithm detects cells of different sizes. We then watershed each dark centroid to the bright membrane. This is repeated across all z sections until each cell is annotated as an individual object. The generated “inside cell” binary data was exported from Nikon Elements software to Bitplane Imaris software and used to place dots representing each hematopoietic cell. The upper histogram shows sensitivity (= number of correctly segmented cells divided by the number of cells identified manually) and specificity (= number of correctly segmented cells divided by the number of identified cells using the automatic algorithm) of cell segmentation (n = 6 areas in 2 sternum segments from 2 mice). The lower histograms compare the distribution of distances between randomly selected cells segmented manually or through the automatic algorithm. **b**, Experimental workflow to perform statistical analyses by comparing observed distributions with that of random cells. Stained bone marrows are imaged and all cells and structures of interest identified and their X, Y, Z coordinates recorded. The cells of interest are replaced with color coded spheres corresponding to each cell type to generate maps with the observed distributions found in vivo. The cell coordinates are then used to quantify distances between any cell and structure of interest. In parallel, different bones are stained with anti-CD45 and anti-Ter119 antibodies (to detect all hematopoietic cells in the tissue) and ESAM and Ly6C (to detect megakaryocytes, sinusoids, and arterioles). **c**, We then obtain the coordinates of all (CD45⁺ and/or Ter119⁺) hematopoietic cells as well as sinusoids, arterioles, and megakaryocytes (identified as shown in a). The coordinates of the hematopoietic cells are used to randomly place dots –representing each type of hematopoietic cells for which random simulations are desired– at the same frequencies found vivo to generate random distributions. The coordinates of the selected random dots are then used to measure the distances between these random cells or to sinusoids, megakaryocytes, or arterioles. Each random simulation is repeated a hundred times.

Does the random distribution take into consideration the frequency of a given cell type? Can cell frequency affect randomness?

Yes, a main advantage of this method is that it takes into consideration the frequencies of all cell and structures of interest. Indeed, more abundant cells are found closer to each other in the random simulations whereas rarer cell types are further away in these simulations. To better illustrate this point we show in **Reviewer Figure 5** below the changes in the distribution of random cell distances as the number of cells in the bone marrow increases.

Reviewer Figure 5. The left map shows the possible coordinates of all hematopoietic cells in a sternum slice. The right maps show an example of random distributions for LT-HSC, MkP, and CFU-E at the same frequencies found in vivo. The dot plots show the distances between each type of random cell to the closest random cell (left plot), sinusoid (central plot), or arteriole (right plot). As cell frequencies increase the distances between random cells diminish. Increasing cell frequencies minimally affect the distribution of distances between random cells and sinusoids or arterioles. Each random simulation was repeated a hundred times.

ST look closer to arterioles and further from sinusoids than LT. Also, ST look closer to the other cell types than to LT. so the conclusion about not being closer than predicted from random distribution is right but not complete in describing that some cell types are closer to others.

This is an important point. To better describe and summarize HSPC distribution in the bone marrow we now show a summary of the distances to all other HSPC or component of the microenvironment for all types of HSPC in **Fig. 2b** (reproduced below). We show detailed comparisons with the random cells in **Extended Data Fig. 3a,bc**.

Figure 2. b, dot plot summarizing the median distance from each HSPC to all other indicated cells and structures in a 35-µm optical slice of the mouse sternum (n = 35 LT-HSC, 52 ST-HSC, 22 MPP2/MyE, 38 MPP3/MPP4/CMP, 61 MkP, 93 MkP in 5 sternum segments from 4 mice). Multiple unpaired t tests, P values are shown

We now clearly describe this in the text of the manuscript as follows (lines 149-153): “For most HSPC we found no differences between the distances from each of the HSPC assayed to all other HSPC when compared to random simulations. The exception were the MPP2/MyE that located closer to each other than random cells (Fig. 2a, and Extended Data Fig. 3a).

In sharp contrast, most HSPC showed preferential localization within the microenvironment. At the population level, all the HSPC -except MPP2- were enriched near megakaryocytes when compared to random cells. All HSPC –except ST-HSC and Pre-MegE – were farther than random cells from arterioles. No HSPC showed specific interaction with the endosteum (including transcortical blood vessels) or sinusoids, although, due to the abundance of sinusoids most multipotent HSPC localized within 10 µm of these vessels (Fig. 2a and Extended Data Fig. 3b-d)”

4) The statistical approach used for the analysis in fig2 f-g, Extended figure 3c and in similar analyses throughout the manuscript should be revised. The data don’t look like having a normal distribution and the choice of a parametric test should be reconsidered, unless an analysis of normality distribution has been performed. In this case this point should be justified in M&M section.

We have revised the statistics as suggested. We now use T-tests or Anova if the data follows normal distribution and Mann-Whitney or Kruskal-Wallis tests if the data is not normal. This is now indicated in the "Statistics" section of the methods and the individual Figure legends.

The random distribution in each of the graphs of fig2 f-g and Extended figure 3c (now Extended Data Figure 3b) looks the same for all the different cell types and it changes based on the reference to sinusoids, arterioles or Megakaryocytes. The difference in frequency of each cell type doesn’t affect the random distribution calculation for that given cell type?

As indicated in the response to point 3 above, the abundance of a cell type affects the random distribution of distances as more abundant cells are closer to each other in random distributions than less abundant cells (**Reviewer Fig. 5 above**). However, sinusoids are so abundant that virtually every random cell is adjacent to a sinusoid. Increasing cell frequencies thus do not change the median distance of the random cells to the closest sinusoid. In contrast, arteries are so rare that almost all random cells will localize far away from an artery. Thus, increasing cell frequencies do not change the median distance of a random cell to an artery.

5) Page 6, lines 125-127: "These results indicate that multipotent progenitors move away from their parent and sister cells and migrate hundreds of micrometers as they differentiate". It is difficult to support that the data show that multipotent progenitors "move away" from their parent and sister cells. If there are no difference from random co-localization, it is difficult to believe that is not just a stochastic observation and that each cell is where it is because of randomness and in this sense that there is any direction to move away from anywhere. This conclusion needs to be better supported by data or by stronger mathematical modeling. LT and progenitors have different proliferation rate. Cells actively dividing and sister cells would be found closer after division. It is unexpected not to see differences at all between quiescent LT and more active progenitor cells. Do the authors observe any specific localization of proliferating cells? For example, can the authors stain for Ki67+ LT and progenitors and inform on whether there is any difference based on proliferation?

We are deeply grateful for this critique as it led us to novel findings above the differential movement and clustering of multipotent HSPC vs committed progenitors in the bone marrow. Please note that HSPC distribution in the marrow is not random as all HSPC (except ST-HSC) are farther than predicted from random to arterioles and closer than random cells to megakaryocytes (except MPP2). Please see **Fig. 2b** (reproduced in response to point #3 and **Extended Data Fig. 3a-c**). Unfortunately, the suggested Ki67 experiment will not be able to resolve the question of whether HSPC that just divided localize together. This is because Ki67 rapidly degrades after cell division and thus HSPC that just divided will be Ki67-, same as HSPC that did not divide.

Our previous results (**Fig. 2a,b**) demonstrated that all HSPC are found as single cells in the bone marrow. Since, after cell division, daughter cells are necessarily adjacent, the results also indicated that the offspring of HSPC were either released into the circulation, differentiated into more mature cells that are not detected in the HSPC stain, or moved away from each other. In the steady-state a minute fraction of total HSPC is released into the circulation^{27,28} indicating that this mechanism is not sufficient to explain lack of HSPC adjacency in the bone marrow. Similarly, in vivo barcoding showed that MPP can extensively self-renew in vivo for long periods of time indicating that the offspring of HSPC divisions does not always terminally differentiate²⁹. Intravital imaging studies support different degrees of HSC motility in the marrow⁹⁻¹². Because most HSC are found as single cell in the marrow (^{30,31} and **Fig. 2a**) the mobility of HSC when adjacent to other HSC/MPP has not been examined.

To investigate HSPC distribution and migration in detail we analyzed live imaging experiments of HSC via a collaboration with the laboratories of Drs. Keisuke Ito and Charles Lin. We performed follow-up analysis of microscopy-guided transplantation of single Dil- labeled, Tie2⁺ HSC in the mouse calvarium as described³². In one instance we observed that (48 hours after the initial transplant) the sole transplanted HSC had divided generating two Dil- labeled, Tie2⁺ cells that were in close proximity. Three hours later one of the daughter cells was no longer visible in the whole calvarium suggesting that it had moved away (**Extended Data Fig. 3e**). These results led us to hypothesize that HSC move away from each other when in close proximity. Follow-up

analysis of single vs multiple cell transplants (5 recipients received one LT-HSC and 6 recipients received respectively 5, 5, 5, 17, 19, and 22 Tie2+ labeled LT-HSC, **Fig. 2c reproduced on next page**). We found that a single cell was visible in 80% of the single cell recipients (4 out of 5) and the cells were all found in near (<100µm) of the transplantation site. Only one out of 73 cells was found –as a single cell– in the mice that received multiple HSC (**Fig. 2d**). Importantly, all tracked recipients showed long-term HSC engraftment that correlated with the number of HSC transferred (**Supplementary Table 3** and ³²) indicating that the absent HSC did not die or terminally differentiate. These experiments indicate that HSC move away from each other, when in close proximity, in vivo. The fact that multipotent HSPC are always found as single cells whereas lineage-committed progenitors form clusters with daughter cells (**Fig. 2a, e, m**) prompted us to investigate whether this was mediated by cell autonomous mechanisms. In live imaging analyses of cultured cells, we found that –after cell division- the offspring of HSC and MPP rapidly moved away from each other. In contrast most committed progenitors remained tightly attached after cell division (**Fig 2q, Extended Data Fig. 5, and Supplementary movies 2-6**). These results explain the differences in progenitor clustering found using our ex vivo mapping and support that cell autonomous mechanisms control differential progenitor clustering at different maturation stages.

6) Overall, the claim of randomness vs implying that LT-HSC, ST-HSC, and other multipotent progenitors do not occupy the same niches it's confusing. If, as assumed in the manuscript, LT and progenitors are evenly distributed through the bone marrow and there is no differential interaction with the microenvironment, it is not clear how it has been defined that each cell type has a niche. It looks more similar to a homogeneous suspension of different cells and the co-distribution of cells would be mainly affected by the frequency of a given cell type. Can the authors clarify this point?

We are grateful for this comment and apologize for the lack of clarity. Our intention was to indicate that HSC and downstream HSPC are not directly regulated by the same adjacent niche cells. We have corrected the text as follows: "These results indicate that LT-HSC and other multipotent progenitors don't share a physical location and are thus not regulated by the same adjacent stromal cells".

Please note that our results show that HSPC localization in the microenvironment is not random. At the population level, all the HSPC –except MPP2– were enriched near megakaryocytes when compared to random cells. All HSPC –except ST-HSC– were farther than random cells from arterioles. No HSPC showed specific interaction with the endosteum (including transcortical blood vessels) or sinusoids, although, due to the abundance of sinusoids most multipotent HSPC localized within 10 µm of these vessels (**Fig. 2a, b and Extended Data Fig. 3b-d**).

Figure 2. c, Tie2⁺ HSC purified from actin-GFP mice were transplanted directly into the calvarial bone marrow of living mice using the approach described in reference 32 as either single cells (5 recipients received 1 cell) or multiple cells (6 recipients received respectively 5, 5, 5, 17, 19, and 22 cells). **d**, The graph reports the fraction of cells found using intravital microscopy in the whole calvarial bone marrow 24hrs following transplantation ($p^{***} = 1.96 \times 10^{-9}$, chi-square test to compare two proportions). A single cell was visible in 80% of the single cell recipients (4 out of 5) and the cells were all found in close proximity (<100 μm) of the transplantation site. Only one out of 73 cells was found [as a single cell] in multiple cell recipients.

Recipient	number of transplanted cells	Peripheral blood chimerism (week 4)
M1	1	not tracked
M2	1	1.14
M3	1	0.46
M4	1	0.23
M5	1	2.76
M6	5	2.59
M7	5	not tracked
M8	5	5.4
M9	17	not tracked
M10	19	69.4
M11	22	13.3

Supplementary Table 3. Peripheral blood engraftment in mice after microscopy-guided single or multiple HSC transplantation

Figure 2. q, Distance between daughter cells at the indicated time points after division from a cultured $n = 30$ HSC, MPP, CFU-E, or MDP and $n = 18$ GP or CLP. **r**, Percentage of daughter cells that have separated more than $50 \mu\text{m}$, 2 hours after division. ($n = 5$ wells for each indicated progenitors).

7) Eventually the data show only that LT and progenitor cells are localized in different niches, **there is no experimental evidence of migration of these cells**. The term migration implies a directionality and it is very speculative from the data to suggest which cell moves and where. So, unless experimentally demonstrated, this statement should be rephrased.

This is a great suggestion. Since we cannot demonstrate directional migration, we have followed the reviewer's advice and replaced the term "migration" with "separated". Please see response to point #5 above providing experimental evidence that multipotent HSPC separate from each other when in close proximity in vivo (**Fig. 2c,d**) and in vitro (**Fig. 2q,r** and **Extended Data Figure 5**).

8) Line 140-142 (for the reviewer's convenience we reproduce the original text next):

"These results support a model where megakaryocyte progenitors are recruited towards active sites of megakaryopoiesis and terminally differentiate to generate 2-4 mature megakaryocytes". The authors claim that the absence of the progenitor in the same area where the differentiation cluster is localized indicates that the progenitor terminally differentiates. The idea is very nice and the in situ identification of this step has never been provided so far. However, this claim should be supported for example by an in vitro differentiation assay or a tracer specific-line, or by live imaging, as the histology per se can't exclude that the differentiated cells have been produced in another niche and then migrated in the new one, as suggested for LT and progenitors.

We are very grateful to the reviewer for raising this point. In the *Ubc-cre-ERT2: confetti model*, cre-mediated recombination leads to irreversible expression of one out of four possible fluorescent proteins (GFP, CFP, YFP, or RFP, reference 34 doi:10.1016/j.cell.2010.09.016 (2010). Because recombination efficiency is approximately 7% we can only conclusively determine if the clusters/production sites that we observe are monoclonal (when a cluster contains confetti-labeled cells all the cells in the cluster are labeled by the same fluorescent tag) or oligoclonal (when a cluster contains confetti-labeled cells and there are more than two fluorescent tags). Since we were unable to distinguish clear clonal relationships between MkP and megakaryocytes we have removed these panels from the manuscript to avoid confusion (**Old Extended Data Fig 3.f-g**).

9) Line 143-145 (for the reviewer's convenience we reproduce the original text next): *"Erythropoiesis takes place in discrete production lines in the sinusoids. Pre Meg-E and Pre CFU-E are found as single cells through the tissue. Pre CFU-E migrate away from Pre Meg-E towards sinusoids (60% in direct contact with sinusoids) but do not map near CFU-E."* Same point about the term migration as explained before.

Thank you for the suggestion. We have replaced "migrate" with "separate". The text (lines 185- 186) now reads: "Pre CFU-E separated from Pre Meg-E and localized to sinusoids (60% in direct contact with sinusoids) but do not map near CFU-E"

10) Line 150-153 (for the reviewer’s convenience we reproduce the original text next): “*Higher powered images revealed that terminal erythropoiesis starts when CFU-E detach from the sinusoids, downregulate CD117 and upregulate CD71, progressively giving rise to several small clusters of early erythroblasts that bud from the vessel*”. Experimental proof of the suggested differentiation steps are missing.

These sequential downregulation and upregulation of CD117, CD71, and Ter119 during terminal erythropoiesis has been described by numerous groups before (references ^{16,23,33}). To make this point clear the text now reads: ‘Terminal erythropoiesis takes place via sequential downregulation of CD117 and CD71, and upregulation of Ter119^{16,23,33}. Higher powered images revealed that when CFU-E detach from the sinusoids, they downregulate CD117 progressively giving rise to several small clusters of early erythroblasts that bud from the vessel. These progressively upregulate Ter119 to generate large, nearly homogenous, clusters of 19 to 96 (Mean = 40 ± 4) late erythroblasts that, in turn differentiate into reticulocytes and erythrocytes that remain in close vicinity to the CFU-E strings (Fig. 2h, Extended Data Fig. 4d and Supplementary Movie 1)’

Moreover, the authors should show that CFU-E CD177- CD71+ are far from sinusoids but not co-localizing with differentiate erythroid cells. It would be interesting to support the data showing the distance from sinusoids for CFU-E, CFU-E cd177- CD71+, differentiated erythroid cells together in the same plot.

We are confused by this question. We did not use CD177 in our experiments. If the reviewer is referring to CD117, then in our hands –and in the literature- all CFU-E are CD117+ as CD71+CD117- cells are either early (Ter119-) or late (Ter119+) erythroblasts (Extended Data Fig. 1o and references 16 doi:10.1016/j.stem.2007.07.005 (2007) and 23 doi:10.1073/pnas.0909296106 (2009)).

11) Line 161 (for the reviewer’s convenience we reproduce the corresponding text next): “*Arterioles are a niche for CLP. In agreement, we found that CLP are selectively enriched near arterioles (mean distance = 53 μm) and depleted near sinusoids (Fig. 2i, j and Extended Data Fig. 4g)*” Extended figure 4 g (now Extended Data Fig. 4j) is confusing. Arterial staining is missing or at least not shown despite the distance analysis is present (fig 2j). Until this point in the manuscript the ESAM staining has been exclusively associated with stem cells, megakaryocytes and sinusoid identification, how is this explained? In general, all these supporting cartoons are slightly confusing because when compared to the microscope image next to it they don’t look like depicting the same. Either it should be explained better what do these cartoons depict or they should be shown next to the exact same image (maybe simplified) that was used to generate them

We apologize for the confusion. ESAM is a pan endothelial and megakaryocyte marker and stains all bone marrow sinusoids, and megakaryocytes and most arterioles. This is now clearly shown Extended Data Fig. 1i below. Ly6C labels arterioles (reference ¹) and we included Ly6C in every panel requiring arteriole imaging. In the case of old Extended figure 4 g we included Ly6C in the lineage mix to visualize arterioles. We agree with

the reviewer that the image was not clear enough and have replaced it with a new one (please see below) to better illustrate that CLP localize near arterioles (as shown by reference 8 and other publications)

Extended Figure 1i Representative images and quantification demonstrating that ESAM stains all CD31CD144⁺ bone marrow sinusoids, Ly6C⁺ arterioles, and CD41⁺ megakaryocytes. n=3 sternum segments from 3 mice.

Extended Figure 4j Representative image showing that CLP map near Ly6C⁺ arterioles

12) Line 164-169 (for the reviewer’s convenience we reproduce the original text next): “*Most Pre-pro B, Pro B, and Pre B cells are selectively enriched near CLP, forming loose clusters (2 ± 1 Pre-pro B, 3 ± 2 Pro B, and 16 ± 8 Pre B within $150 \mu\text{m}$ of each CLP). The more mature cells located farther away from the CLP indicating migration away from the cluster (Fig. 2k-n). Unexpectedly, clonal fate mapping demonstrated that confetti-labeled CLP do not map near Pre-pro B, Pro B, or Pre B labeled with the same fluorescence protein (Extended Data Fig. 4h)*”. Analysis in confetti mice (line 168-169) does not recapitulate the previous observation (line 164-165). Can these data be explained more clearly?

We apologize for the confusion. As shown in **Fig. 2 m-p** (old Fig. 2k-n) Pre-pro B, Pro B, and Pre B cells cluster around CLP. In the *Ubc-cre-ERT2: confetti model*, cre-mediated recombination leads to irreversible expression of one out of four possible fluorescent proteins (GFP, CFP, YFP, or RFP, reference ³⁴). Because recombination efficiency is approximately 7% we can only conclusively determine if the clusters/production sites that we observe are monoclonal (when a cluster contains confetti-labeled cells all the cells in the cluster are labeled by the same fluorescent tag) or oligoclonal (when a cluster contains confetti-labeled cells and there are more than two fluorescent tags). In the case of CLP and B cell production sites the confetti experiments (now **Extended Data Fig. 2i**) conclusively demonstrate that these structures are oligoclonal. The data also shows that the Pre-pro, Pro-, and Pre-B cells labeled with the same confetti color as the CLP are not closer than not-labeled Pre-pro, Pro-, and Pre-B cells suggesting that these cells moved away from the CLP. To indicate this more clearly, we have edited the text as follows (Lines 208-217): “Most Pre-pro B, Pro B, and Pre B cells were selectively enriched near CLP, forming loose clusters (2 ± 1 Pre-pro B, 3 ± 2 Pro B, and 16 ± 8 Pre B within $150 \mu\text{m}$ of each CLP). The more mature cells located farther away from the CLP suggesting

movement away from the cluster (Fig. 2m-o). *Ubc-creERT2:confetti* fate mapping showed that these clusters were oligoclonal. We found differentiating cells labeled in the same confetti color as the CLP but these did not map closer to the CLP than expected from random cells (Extended Data Fig. 4l). Taken together these experiments suggests that daughter cells moved away from the CLP after division but remain associated in loose clusters. This is in agreement with live imaging studies showing that Pre-B cells are highly motile³⁵. These results indicated that clusters of CLP and differentiating B cells are oligoclonal B cell production sites (see Extended Data Fig. 4m for formal definition) near arterioles”.

13) Line 211 (for the reviewer’s convenience we reproduce the original text next): *“Hemorrhage inhibits lymphopoiesis without reducing the number of B cell production lines (Fig. 3e and Extended Data Fig. 5a, 6g).”* Data in figure 3e and Extended figure 5a are in contrast with the text: in the text the authors indicate that number of cells in the B cell line does not decrease after hemorrhage, however data in figure 5a indicated a reduction of these cells. Figure 3e only shows that global lymphoid cell number is not affected, there is no proof that B cells are not altered here. Can the author clarify this point?

In **Old figure 3e** we showed that the number of B cell production sites (defined as clusters of Pre-pro B, Pro B, and Pre B cells identified as shown in **Extended Data Fig. 4m**) is not reduced. However, the number of cells in each of these production sites is dramatically reduced and this was shown in **Old Figure 3l**. In other words, the number of B cell “factories” does not change but their output is severely reduced.

This critique (and those on points #14-17) made us realize that these results were not presented clearly enough. We have completely revamped **Fig. 3** to show, side by side, the changes in number of production sites and in their cellular content at the peak of each type of perturbation. More detailed analyses of production site composition and kinetics are shown in Extended Data Fig. 8 and 9. Please see Figure 3 in the next page. Changes in B cell production sites are shown in **Fig. 3e, f**.

14) Line 212-213 (for the reviewer’s convenience we reproduce the original text next): *“Instead, there is a selective accumulation of Pre-pro B cells around the CLP but reductions in Pro B and Pre B cells indicating delayed differentiation at the Pre-pro B stage (Fig. 3k, l)”*: the data is only qualitative. A quantification is necessary to support the statement.

We believe that the reviewer missed the quantification. **Old Fig. 3k** showed a representative image of how the B cell production sites change after phlebotomy. **Old Fig. 3l** quantified the number of cells per B cell production site for 10 production sites in 3 sternums.

This critique (and those on points #13-17) made us realize that these results were not presented clearly enough. We have completely revamped **Fig. 3** as shown in response to point #13 to show, side by side, the changes in number of production sites and in their cellular content in response to each type of perturbation. Changes in B cell production sites are shown in **Fig. 3e, f**.

Figure 3. Anatomy of stress responses. **a**, Heat map summarizing changes (normalized to steady-state or saline-treated) to the indicated populations (exemplars of peak erythropoiesis, lymphopoiesis, granulopoiesis, and myelopoiesis responses) two days after phlebotomy (except for lymphopoiesis which corresponds to day 8 as this is the nadir of the Lymphopoiesis response as shown Supplementary Figure 3), six days after *L. monocytogenes* infection, four days of G-CSF treatment, and 20 months of age (n=7, 6, 4 and 6 mice for each indicated population two days after phlebotomy, six days after phlebotomy, 4 days of GCS-F treatment and 20-month old aging mice,).

b, Maps showing HSPC location at the indicated time points and insults. Map dots are three times the average size of the relevant cell. **c-j**, the graphs quantify the number of production sites per sternum (left panels, n= 3 sternum segments from 3 mice) and the cellularity of each production site (right panels, n = 6 randomly selected production sites in three sternum segments from 3 mice) at the same time point and insult as in "a". The maps show the distribution of the indicated cells in a large region of a sternum segment. Statistical differences were calculated using two way ANOVA t-tests if the distributions were normal or Kruskal-Wallis test if not normal; P values are shown.

15) Line 207-217(for the reviewer's convenience we reproduce the original text next): "*The data also showed dynamic adaptations of the production lines in response to stress. Two days after phlebotomy there are 50% more erythroid production lines, and these contain 75% more CFU-E but 1.5- fold fewer reticulocytes and 12- fold fewer erythrocytes as these are released into the circulation (Fig. 3e-j). Hemorrhage inhibits lymphopoiesis without reducing the number of B cell production lines (Fig. 3e and Extended Data Fig. 5a, 6g). Instead, there is a selective accumulation of Pre-pro B cells around the CLP but reductions in Pro B and Pre B cells indicating delayed differentiation at the Pre-pro B stage (Fig. 3k, l). These microanatomical changes are progressively restored until, on day 20 after phlebotomy, the anatomy of erythroid and lymphoid production lines is indistinguishable from steady-state mice (Fig. 3g-j, l). Phlebotomy did not affect the number and anatomy of neutrophil and monocyte/dendritic cell production sites at any time point examined (Fig. 3e and Extended Data Fig. 5a).*" This analysis is very interesting, but the outcome is somehow expected. As the differentiation lines adapt/respond differently to the stress it would be interesting to check how the different cells in each line are affected (i.e., clonal contribution). It would be nice to perform the experiment in confetti mice or in reporter cell lines to better describe the plasticity of the lines.

We respectfully disagree with the reviewer statement that the outcome is somehow expected. The anatomy of hematopoiesis after stress responses has only been superficially examined before due to lack of tools to visualize differentiation. Importantly, we show that changes in the number and output of the production sites mediates hematopoietic plasticity and allow the bone marrow to adjust blood cell production. This indicated that stress hematopoiesis uses the same structures as steady-state hematopoiesis for generating blood. This is completely unexpected because prior studies proposed specialized structures (e.g. emergence of GMP clusters doi: 10.1038/nature2169, and hemospheres DOI: 10.1038/emboj.2012.308) that preferentially responded to stress.

As requested we performed clonal analyses of the production sites that expand under stress using confetti mice. The increased output of the production sites in response to stress can be mediated by either increased self-renewal of the cells in the site or increased recruitment of progenitors to the site. To distinguish between these two possibilities, we used confetti mice. Because expression of the fluorescent proteins in this model is irreversible³⁴, increased self-renewal will necessarily lead to the accumulation of cells labeled by the same fluorescent protein. These analyses failed to reveal increases in the confetti cells indicating that production sites expand in respond to stress by recruiting additional upstream progenitors (**Extended Data Fig. 9e** reproduced on the next page)

16) Line 220(for the reviewer's convenience we reproduce the original text next): "*Reductions in lymphopoiesis take place via reduced output of B cell production lines, starting at the Pro B stage (Extended Data Fig. 7a, b), while the number of B cell lines is unaffected (Extended Data Fig. 6g, o)*". the B cell line per se has not been tested, but only CLPs and lymphoid cells in general. Quantification of B cell number and localization should be given to support the data.

We apologize for not presenting these data clearly enough. **Old Extended Data Fig. 7a** showed a representative image of how the B cell production sites change after *L. monocytogenes* infection. **Old Fig. 7b** quantified the number of cells per B cell production site for 10 production sites in 3 sternums after *L. monocytogenes* infection. The quantifications in **Old Extended Data 6g,o** showed that *L. monocytogenes* infection did not change the number of B cell production sites. Similar to phlebotomy, *L. monocytogenes*

infection does not change the number of B cell “factories” but their output is severely reduced. As discussed above these results are now shown side by side in Fig. 3 to avoid confusion. Changes in B cell production sites are shown in Fig. 3e, f.

Extended Data Figure 9. e, To examine how productions site grow in response to stress we used confetti mice. If the site grows via increased self-renewal, then the number of cells labeled by the same fluorescent protein should increase. Alternatively, if the site grows via increased cell recruitment to the site then the number of confetti-labeled cells in the cluster will remain constant. The top images and dot plots show the changes in total and confetti-labeled cells in each erythroid production site in the steady-state or 2 days after phlebotomy-driven expansion. The bottom images and dot plots show the changes in total and confetti-labeled dendritic cells in each mono/DC production site 6 days after *L. monocytogenes* infection or 20 months of age. n=10 for erythroid production sites, n=10 for mono/DC sites in response to infection, and n=10 for mono/DC sites in 20-month-old mice. Statistical differences were calculated using two-tailed unpaired Student’s t-tests; P values are shown.

17) Line 225 (for the reviewer’s convenience we reproduce the original text next): “*However, it dramatically increases dendritic cell - and reduces Ly6Clo monocyte- output in dendritic cell/monocyte production lines (Extended Data Fig. 7f-h)*” Missing quantification in Extended figure 7f-h.

This critique is similar to the points above. In **Old Extended Data Fig. 6 j** we showed that *L. monocytogenes* caused an increase in the overall number of monocyte/dendritic cell factories while increasing their production of dendritic cells and reducing their output of monocytes (quantified in **Old Extended Data Fig. 7h**). As discussed above these results are now shown side by side in Fig. 3 to avoid confusion. Changes in dendritic cell/monocyte production sites are shown in **Fig. 3i, j**.

18) Page 11 lines 259-265 (for the reviewer’s convenience we reproduce the original text next): *Strikingly, aging caused microanatomical perturbations that were cell- and lineage-specific: in young mice LT-HSC are*

found as single cells evenly distributed through the bone marrow (Fig. 2a-c). In contrast, in aged mice, lymphoid- and myeloid-biased LT-HSC localize much closer to other LT-HSC than predicted from random simulations – leading in some cases to the emergence of loose clusters of myeloid- and lymphoid-biased LT-HSC (Fig. 4a, b and Extended Data Fig. 9n). This correlates with selective lymphoid LT-HSC attachment to the remodeled sinusoids (Fig. 4d), which have been implicated in protecting HSC from aging”: It would be clearer to show a direct comparison between young and aged samples. The data in panels (Fig. 2a-c) are difficult to compare with (Fig. 4d).

We have modified these graphs (now **Extended Data Fig. 7e**) to directly compare young and aged samples.

19) Page 11 lines 265-270 (for the reviewer’s convenience we reproduce the original text next): *“The production lines for each lineage also showed lineage-specific microanatomical adaptations to age. The number of erythroid production lines increases slightly but each individual line shows reductions in almost all steps of erythropoiesis indicating reduced output (Fig. 4e-f). In other cases, we observe delayed differentiation: B cell lines display selective accumulation of Pre-pro B cells and severe reductions in Pro B and Pre B cells indicating stage-specific delayed differentiation at the Pre-pro B stage (Fig. 4g, h and Extended Data Fig. 10a) “* The reduction in erythropoiesis and lymphopoiesis upon aging have been extensively reported. It is unclear why the authors describe them as “microanatomical adaptations”? What do the authors mean with “microanatomical adaptations”? It looks like the changes upon aging are due to an adaptation to the changes in the microenvironment but these microenvironmental changes are not clearly described nor the mechanism that should justify the “adaptation”. Can the authors provide data that the reduction in erythropoiesis and lymphopoiesis are due to a microanatomical adaptations? Otherwise, it looks unclear what are the authors referring to.

We are very grateful for this critique that have improved the clarity of the manuscript. In our first submission we used the term “microanatomical adaptations” to refer to changes (in cellularity or cell identity) per production site. We used the term “macroanatomical adaptations” to refer to changes in the number of production sites. Based on the reviewer’s critique we have edited the text as follows:

“In aged mice, the number of production sites for all lineages was maintained when compared to young mice but erythroid, lymphoid, and granulocyte production sites displayed reduced output (**Fig. 3b-h**). Monocyte and dendritic cell production sites displayed reduced monocyte – but increased dendritic cell- output (**Fig. 3i-j**). These results indicate that changes at the macro (numbers of production sites) and micro-anatomical level (cell content and output) of the production sites orchestrate hematopoietic plasticity to stress (**Fig. 3k**)”

Minor points:

- Line 70-72 (for the reviewer’s convenience we reproduce the original text next): ESAM as a marker of HSPCs has already been described/suggested (i.e. Yokota T. Blood 2009 and Tomohiko I. Blood 2015). References should be included.

These references (20,22) have been included. The text now reads: In agreement with previous studies the immunophenotyping showed that ESAM is uniformly expressed in all HSC²⁰⁻²² and subsets of multi/oligopotent HSPC but absent in lineage-committed progenitors (Fig. 1b, c)

- A table to recap all the surface markers associated to each cell type for each production line would be extremely helpful.

We are very grateful for this excellent suggestion. This is now shown as a scheme in **Figure 1g-j** and as a brief table in **Supplementary Table 2**.

Figure 1. g-h, Scheme summarizing expression of cell surface markers used to interrogate multipotent cells (g), erythropoiesis (h), B lymphopoiesis (i) and myelopoiesis (j).

- A table to recap the surface markers Abs and the secondary Abs and relative conjugated fluorophores used in the histological analysis would be helpful to guarantee data reproducibility.

This information is now provided as **Supplementary Table 2**.

Reviewer Reports on the First Revision:

Referees' comments:

Referee #1 (Remarks to the Author):

I would like to thank the authors for extensively revising their paper and responding very well to almost all of my comments. I think the paper has gained enormously as a result. I am also very happy about the new biology regarding G-CSF and phlebology. This makes the paper even more convincing. Future work should then show, how this is controlled. I am looking forward to that. I have some minor remaining issues (in chronological order):

- 1) List of used antibodies: please provide complete information, i.e. containing data on the antibody clone, color, supplier, catalog nr. etc. This is going to be a very important resource for the field. Also, information on which antibodies don't work or were negative in the author's hands is important.
- 2) New fig 1d: graph much clearer now. Also having the same numbers of cells everywhere now is much stronger. I would make things extra clear in the legend by writing "...after transplantation of the indicated progenitors (n = 4 mice/group)". Currently the inattentive reader might still think that just 4 mice in total had been analyzed.
- 3) You might call me hair-splitting, but the last two figs in Fig. 1f still do not align with the first 4 (at least on my screen). I am sure this can be solved.
- 4) In general, I find the fig. 1 extremely busy. Maybe it would be wise to think about splitting it in two.
- 5) As far as I can see the fact that HSPC do not map near bone surfaces and hence also not close to TCVs is currently hidden deeply within the supplemental data. Please make a notion of this fact also in the main text.
- 6) New fig. 3a: the colored dots for "increased" are red, while in the legend the term has an orange color. Please align.
- 7) If I got it right, the data in Fig. 3 are only from the sternum (and then partly related to the appearance in long bones). Hence, I suggest to change the title of the fig. into "Anatomy of stress responses in sternal bone marrow" (or comparable) and also make this comparison again very clear in the legend.
- 8) Interpretation of the results in Extended Data Fig. 3e: authors claim, that the daughter cell has moved away. This can only be concluded when it has been observed by constant imaging. If there has been no imaging in these 3 hours, the daughter cell can also just have died and disappeared because of that (as e.g. visible in some of the cultured cells). Please tone down the wording here.

Referee #2 (Remarks to the Author):

The authors have revised the manuscript and added several additional pieces of data which strengthen the paper. The presentation is clarified and the language is toned down from the prior version. It is particularly notable that confetti mouse experiments show increased recruitment of progenitors in response to stress, rather than an increase in cellularity at each production site. This is a significant advance in our understanding of stress hematopoiesis. The data suggesting that responses to G-CSF differ by bone are intriguing but preliminary. How do the two tibia in a single mouse compare to each other? How have the authors accounted for possible effects of cell mobilization / flux in the G-CSF treated mouse that would affect the number of cells localized at the production sites? Overall, the paper represents an important technical advance in the field and provides innovative insights, but the question of flux posed above should be answered prior to publication.

Specific comments:

1) At line 56-59, the authors state that "Lineage-committed progenitors are serially recruited to blood vessels where they contribute to lineage-specific microanatomical structures, composed of progenitors and immature cells, which function as the production sites for each major blood lineage."

What is meant by "serially recruited?" The data presented are insufficient to conclude this is a serial process, but should just state that the progenitors are located along blood vessels where they function as production sites.

2) At line 278-9, the authors state that "They also indicated that aging irreversibly perturbed the production sites (Fig. 3k and Extended Data Fig. 9a-d)."

The authors have not demonstrated that this process was irreversible; rather they showed that aging perturbed the production sites.

3) Figure 4: After G-CSF treatment the number of neutrophil production sites in the tibia did not increase, the density of GP barely increased and the total number did not change, while these numbers went down in the sternum. This contradicts the earlier conclusion that "production sites expand in response to stress by recruiting additional upstream progenitors" since no additional progenitors are seen here in the tibia.

4) Furthermore, the number of mature neutrophils increased in the sternum but hardly at all in the tibia. These results do not match the description provided:

"While the tibia responded to G-CSF by increasing neutrophil output the sternum responded by shutting down neutrophil production leading to loss of neutrophil production sites"

If the sternum shut down neutrophil production, why are the numbers twice as high? Meanwhile the neutrophil numbers in the tibia are maybe 15% higher. What are the peripheral neutrophil numbers in the saline versus G-CSF treated mice? Can the increased peripheral neutrophil number be accounted for by a 15% increase in cells in tibia?

These contradictory points raise the issue of flux as a major possible confounder of these studies. As the authors acknowledge, cell numbers are not tracked over time in this system, and therefore if a production site is highly productive but the cells are rapidly released from that site, an accumulation of cells would not be evident. There is a possibility that the vascularity or flux out of the tibia is higher than out of the sternum in this particular instance, which would make it appear that the tibia is not producing much. For Figures 4a-c it seems that the conclusions are based on imaging of one sternum and one tibia in three mice. What is the reproducibility of these findings between two tibia in the same mouse?

5) The authors state that "The tibia and sternum displayed similar suppression of erythropoiesis, monoopoiesis, and lymphopoiesis indicating that both bones are equally exposed to G-CSF (Fig. 4d).

This should state "suggesting that" otherwise the authors should spin down the bone marrow and measure the cytokine concentrations in different sites, or do qPCR to show the transcriptional response to G-CSF in different sites.

6) Little is said about the microanatomical location of HSCs during stress. Figure 4a shows the relative abundance of these cells after phlebotomy infection, etc, but the conclusion is limited to a general comment that the anatomy of the marrow is unchanged after stress. Does the HSC proximity to megakaryocytes change upon various types of stress? What about MPPs? In Figure 4B it looks like there is a proliferation of ST-HSCs and/or MPP2's after infection but this is not commented on.

Referee #3 (Remarks to the Author):

Wu et al. have strengthened the manuscript by adding significant new data and by clarifying some of their conclusions. I have only minor comments for them to consider, all of which could be addressed by alterations to the text.

1. The authors argue that pairs of HSCs are rarely observed adjacent to each other not because of the mobilization of one or both daughter cells from an HSC division into circulation because "in the steady-state a minute fraction of total HSPC is released into circulation". But only a minute fraction of HSCs is dividing at any one time. So the number of HSCs in circulation may be completely consistent with the number of HSCs that are dividing at any one time. My sense is that it's still not clear why pairs of HSCs are rarely observed together, and whether this reflects the migration of HSCs away from each other within the bone marrow, as favored by the authors, or the intravasation of one or both daughter cells into circulation, or the death or differentiation of one of the daughter cells.
2. The authors note that "in vivo barcoding showed that MPP can extensively self-renew in vivo for long periods of time". My impression is that these data show that MPPs can persist for long periods of time in vivo while contributing to hematopoiesis but I'm not aware of evidence that they extensively self-renew while doing so. Aren't the data consistent with the possibility that MPPs persist for long periods of time, while remaining mainly quiescent and dividing intermittently? The MPPs might divide only a handful of times despite persisting for long periods of time.
3. The authors concluded "These results indicate that LT-HSC and other multipotent progenitors don't share a physical location and are thus not regulated by the same adjacent stromal cells". I realize the authors have already tried to clarify this statement but its meaning is still unclear. Is the point that clonally-related HSCs and their immediate progeny don't remain in the same location, such that they don't remain under the influence of the same stromal cell? Or is this a more general statement about cells that are not necessarily clonally related? Is this statement intended to imply that HSCs and MPPs do not obtain growth factors from the same class of stromal cells? My impression is that isn't what the authors meant, but it could be read that way. The published data show that all multipotent HSPCs depend upon SCF synthesized by LepR+ stromal cells.
4. The authors stated that "No HSPC showed specific interaction with the... sinusoids, although, due to the abundance of sinusoids most multipotent HSPCs localized within 10 um of these vessels". What does this mean? The authors seem to imply that while HSCs and MPPs all localized adjacent to sinusoids, that this doesn't reflect a "specific interaction" with the sinusoids but rather it occurs by chance as a result of the abundance of sinusoids throughout the bone marrow. But I don't think there's any evidence to support that way of interpreting the observation. The key question is whether there's something special about the environment immediately around the sinusoids that helps to promote the maintenance of HSCs, MPPs, and some early restricted progenitors. It's quite possible that there is. For example, a specific subset of LepR+ cells that is fated to form adipocytes and that produces high levels of HSC niche factors seems to localize to sinusoids and not to arterioles. Megakaryocytes also localize to the sinusoids for a reason – they are able to insert processes through the fenestrated walls of sinusoids to deposit platelets into circulation. I don't think the data favor the idea that the cells that are very closely associated with sinusoids are there by chance alone – there are likely functionally important attributes of that environment that are caused by the proximity to sinusoids.
5. The authors might note in the erythroid progenitor section that Comazzetto et al. also showed that early erythroid progenitors localize to sinusoids.

Referee #4 (Remarks to the Author):

The authors have addressed all my comments and have dramatically improved the clarity of the manuscript and the explanation of the results for the reader. It is now clear how they describe the localization of HSPC within the BM with respect to other hematopoietic and non-hematopoietic cells. The experiments tracking single HSC divisions are very important and interesting. The final description of the anatomy of hematopoiesis has dramatically improved by including imaging of different bones, which show very reproducible patterns, distinguishing HSPCs that localize to distinct niches (MK) compared to progenitors that localize at specific vessels (arterioles

for CLPs and sinusoids for neutrophils and erythroid lineages) without overlapping. Moreover, it is very interesting the observation that committed progenitors form clusters with daughter cells, while HSPCs separate from each other, which is an intrinsic characteristic of these different cell types, since reproducible also in vitro.

The data of the macroanatomy and microanatomy after all the different stressors are also extremely improved and overall, the manuscript includes now an impressive amount of high quality data.

A few focused points to be addressed:

1) The results in Fig. 2c-d should be clarified: lines 178-180: I understand that after 1 hour single transplanted HSC and 5 transplanted HSCs are visible at the transplantation site. I don't understand why they claim that only 1 out of 73 cells was found as single cell in the multiple (HSC) recipients... After how long they look at the transplantation site? 48hrs? what happened in most of the cases? No cells detected? Or did they detect more than 1 cell? Can the author explain better the results? The authors could include this info in the supplementary table 3 with one column indicating the number of HSC detected after 1 hour and another column indicating the number of HSC detected after 48 hrs.

2) The heatmap of fig 3a is based on which dataset? Please clarify.

3) In extended data figure 7 the number of sinusoids, arterioles and Mk is quantified per fragment. The same applies to the number of production sites quantified in Fig. 3c, e, g and i. Fragment size might be different (for example in aged bones). Can the authors normalize the measurements based on imaged BM area or BM volume like in Figure 4b?

4) Lines 258-259: the authors report on some changes in MPP2/MyE after infection and of HSCs upon aging. These observations are very interesting, while underscoring specific changes in HSPC upon infection that are different from aging. These observations deserve a more careful quantification. Can the authors analyze the statistic of extended data figure 7c and highlight the data that support the claim? Also, the authors define these clusters "rare"? can it be quantified better?

5) The "production site" as it is defined and investigated in this paper is novel and quite provocative. Each production site identifies discrete BM regions (microanatomy). The size of each production site varies according to the median distance of random cells from each specific progenitor. To improve the clarity for the reader, I would suggest including a table to recapitulate the general definition of "production site", "microanatomy", "macroanatomy" and the specific definition of each (for example as per Extended data Figure 4: B cell production site includes pre-proB, pro B and pre B cells closer than 150µm (median distance of random cells) to the CLP).

Author Rebuttals to First Revision:

Summary of changes to the manuscript

We would like to thank all the referees for the positive review and detailed critiques. We have addressed all of these in detail in the response for each referee below.

We would like to highlight three major changes to the manuscript:

1. In response to a critique from referee 1 regarding the complexity of the figures we have simplified Figure 1 and split Figure 2 into two figures. New figure 2 now examines HSPC distribution and Figure 3 erythropoiesis and lymphopoiesis in the steady state.
2. In response to a critique from referee 2 we have performed extensive new analyses of the response to G-CSF in the humerus and show that it behaves like the tibia (but not the sternum). This is now shown in new **Fig. 5a-c**. Additionally, we have used confetti mice to quantify the efflux of preneutrophils from the neutrophil production sites after G-CSF in tibia and sternum. These new results are now shown in new **Extended Data Fig. 10c**.
3. In response to questions from referees 2 and 3 we have now quantified -and performed detailed statistics- of HSPC localization with different components of the microenvironment (sinusoids, arterioles, megakaryocytes, endosteum) after stress. These results are now shown as a heat map in new **Extended Data Fig. 7e** and as dot plots in **Supplementary Figure 8a,b**.
4. We have deposited all the raw images corresponding to each panel of each figure and these are available through a public repository in the “Data Reporting” section of the manuscript.

Detailed responses to all the referee’s comments can be found on the following pages:

Referee 1	Page 2
Referee 2	Page 5
Referee 3	Page 14
Referee 4	Page 17

Referees' comments:

Referee #1 (Remarks to the Author):

I would like to thank the authors for extensively revising their paper and responding very well to almost all of my comments. I think the paper has gained enormously as a result. I am also very happy about the new biology regarding G-CSF and phlebology. This makes the paper even more convincing. Future work should then show, how this is controlled. I am looking forward to that. I have some minor remaining issues (in chronological order):

We are extremely grateful to the referee for the positive review and all the great suggestions that improved the manuscript. We have addressed all the minor issues in detail below.

1) List of used antibodies: please provide complete information, i.e. containing data on the antibody clone, color, supplier, catalog nr. etc. This is going to be a very important resource for the field. Also, information on which antibodies don't work or were negative in the author's hands is important.

This information is now provided in Supplementary Table 2 (an example for the HSPC stain is reproduced below). The list of antibodies that did not yield specific stains in our hands is now shown as Supplementary Table 4.

		Detection of multipotent HSPC						
Antibody information	Target	CD117	ESAM PE	CD41	CD48	CD150	Ly6C	Streptavidin
	Clone	2B8	1G8	MWReg30	HM48-1	TC15-12F12.2	HK1.4	-
	Fluorochrome	Brilliant Violet 480	PE	Alexa Fluor 488	Alexa Fluor 647	Brilliant Violet 421	Biotin	Alexa Fluor 750
	Catalog	566074	136204	133908	103416	115926	128004	S21384
	Supplier	BD Horizon	Biolegend	Biolegend	Biolegend	Biolegend	Biolegend	Thermo Fisher Scientific
								
Cells and structures identified	LT-HSC	+	+	-	-	+	-	
	ST-HSC	+	+	-	-	-	-	
	MPP2/MyE	+	+	-	+	+	-	
	MPP3/MPP4/CMP	+	+	-	+	-	-	
	MKP	+	+	+	+	+	-	
	Pre-MegE	+	-	-	+	+	-	
	Megakaryocytes	-	+	+	+	Some	-	
	Sinusoids	-	+	-	-	-	-	
	Arterioles	-	+	-	-	-	+	

2) New fig 1d: graph much clearer now. Also having the same numbers of cells everywhere now is much stronger. I would make things extra clear in the legend by writing "...after transplantation of the indicated

progenitors (n = 4 mice/group)". Currently the inattentive reader might still think that just 4 mice in total had been analyzed.

Great suggestion and edited as requested, please note that -in response to critique #4 below- we have moved this panel to Extended Data Fig. 1c.

3) You might call me hair-splitting, but the last two figs in Fig. 1f still do not align with the first 4 (at least on my screen). I am sure this can be solved.

We apologize for the omission. We have corrected this.

4) In general, I find the fig. 1 extremely busy. Maybe it would be wise to think about splitting it in two.

We agree with the referee that -with all the extra panels added during the revisions- Figures 1 and 2 are very busy. We have addressed this critique in two different ways. First, we moved the panel showing the ESAM+ or ESAM- HSPC transplants to Extended Data Fig. 1c. Additionally, we have divided former figure 2 (anatomy of HSPC, erythropoiesis, and lymphopoiesis) in two. Figure 2 now shows the HSPC distribution and the life imaging adoptive transfer and Figure 3 the data on erythropoiesis and lymphopoiesis. We hope that these changes address the critique, and we are grateful for any additional feedback.

5) As far as I can see the fact that HSPC do not map near bone surfaces and hence also not close to TCVs is currently hidden deeply within the supplemental data. Please make a notion of this fact also in the main text.

We believe that the reviewer missed this information. The fact that HSPC do not map near bone surfaces was summarized in the heat map shown in Fig. 2a and shown in detail in Extended Data Fig. 3c. The text (lines 158-161) reads "no HSPC subset preferentially localized to sinusoids or the endosteum (including transcortical blood vessels) when compared with random cells. This is likely due to the abundance of sinusoids as most random cells and multipotent HSPC localized within 10 μm of these vessels (Fig. 2a,b and Extended Data Fig. 4b-d)".

Figure 2. Anatomy of steady-state hematopoiesis in the sternum of young mice. b, dot plot summarizing the median distance from each HSPC to all other indicated cells and structures (b) in a 35-μm optical slice of the mouse sternum (n= 35 LT-HSC, 52 ST-HSC, 22 MPP2/MyE, 38 MPP3/MPP4/CMP, 61 MkP, 93 Pre Meg-E/Pre CFU-E in 5 sternum segments from 4 mice).

6) Fig. 3a (now Fig. 4a): the colored dots for "increased" are red, while in the legend the term has an orange color. Please align.

Good catch and edited as requested.

7) If I got it right, the data in Fig. 3 (now Fig. 4) are only from the sternum (and then partly related to the appearance in long bones). Hence, I suggest to change the title of the fig. into "Anatomy of stress responses in sternal bone marrow" (or comparable) and also make this comparison again very clear in the legend.

Great suggestion. We have changed the title as suggested.

8) Interpretation of the results in Extended Data Fig. 3e: authors claim, that the daughter cell has moved away. This can only be concluded when it has been observed by constant imaging. If there has been no imaging in these 3 hours, the daughter cell can also just have died and disappeared because of that (as e.g. visible in some of the cultured cells). Please tone down the wording here.

We agree with this comment and have edited the manuscript as follows (lines 169-172, changes in bold): "In one instance we observed that (48 hours after the initial transplant) the sole transplanted HSC had divided generating two Dil- labeled, Tie2⁺ cells that were in close proximity. Three hours later one of the daughter cells was no longer visible in the whole calvarium suggesting that it **had moved away or died** (Extended Data Fig. 4e)".

Referee #2 (Remarks to the Author):

The authors have revised the manuscript and added several additional pieces of data which strengthen the paper. The presentation is clarified and the language is toned down from the prior version. It is particularly notable that confetti mouse experiments show increased recruitment of progenitors in response to stress, rather than an increase in cellularity at each production site. This is a significant advance in our understanding of stress hematopoiesis. The data suggesting that responses to G-CSF differ by bone are intriguing but preliminary. How do the two tibia in a single mouse compare to each other? How have the authors accounted for possible effects of cell mobilization / flux in the G-CSF treated mouse that would affect the number of cells localized at the production sites? Overall, the paper represents an important technical advance in the field and provides innovative insights, but the question of flux posed above should be answered prior to publication.

We are extremely grateful to the referee for the positive review and all the great suggestions that improved the manuscript. We have addressed all the critiques in detail below. To address the question on whether G-CSF might induce faster mobilization/flux from the production sites we performed clonal fate mapping of the production sites using confetti mice. Our new results (discussed in detail in the response to point #4) demonstrate that G-CSF induces preneutrophils to move away from the GP and leave the production site. However, the data conclusively shows that this movement is identical between sternum and tibia (**New Extended Data Fig. 10c-g**).

Specific comments:

1) At line 56-59, the authors state that “Lineage-committed progenitors are serially recruited to blood vessels where they contribute to lineage-specific microanatomical structures, composed of progenitors and immature cells, which function as the production sites for each major blood lineage.”

What is meant by “serially recruited?” The data presented are insufficient to conclude this is a serial process, but should just state that the progenitors are located along blood vessels where they function as production sites.

We agree with this comment and have removed the word “serially”. The sentence now reads: “Lineage-committed progenitors are recruited to blood vessels where they contribute to lineage-specific microanatomical structures, composed of progenitors and immature cells, which function as the production sites for each major blood lineage”.

2) At line 278-9, the authors state that “They also indicated that aging irreversibly perturbed the production sites (Fig. 3k (now 4k) and Extended Data Fig. 9a-d).” The authors have not demonstrated that this process was irreversible; rather they showed that aging perturbed the production sites.

Great suggestion. We have removed “irreversibly”. That sentence now reads: “They also indicated that aging perturbed the production sites (Fig. 3k and Extended Data Fig. 9a-d)”.

3) Figure 4: After G-CSF treatment the number of neutrophil production sites in the tibia did not increase, the density of GP barely increased and the total number did not change, while these numbers went down in the sternum. This contradicts the earlier conclusion that “production sites expand in respond to stress by recruiting additional upstream progenitors” since no additional progenitors are seen here in the tibia.

We thank the reviewer for highlighting this. We have clarified our statement on lines 292-293 as follows: “**erythroid and mono/DC** production sites expand in response to stress by recruiting additional upstream progenitors”

4) Furthermore, the number of mature neutrophils increased in the sternum but hardly at all in the tibia. These results do not match the description provided: “While the tibia responded to G-CSF by increasing neutrophil output the sternum responded by shutting down neutrophil production leading to loss of neutrophil production sites” If the sternum shut down neutrophil production, why are the numbers twice as high? Meanwhile the neutrophil numbers in the tibia are maybe 15% higher.

This was due to an error on our part when assembling the figure. We inadvertently flipped the panel quantifying mature neutrophils in sternum and tibia in “old figure 4c”. As a result, we incorrectly reported that neutrophils numbers double in the sternum but minimally increase in the tibia. In fact, the opposite is true. The correct panel is provided below (now also including humerus data) and shown in Figure 5c.

We deeply apologize for this error.

Please note that **-to increase the rigor of our data -in addition to the experiments requested by the referee- we have now quantified neutrophil production after G-CSF in another long bone, the humerus,** and the results are identical to the tibiae (please see the correct panel on the right). These results confirm loss of neutrophil production in the sternum -and increased production in all other bones (please see also **Reviewer Fig. 2** in the response to the next question)- after G-CSF. The full data on G-CSF effect in the sternum is shown in Fig. 5a-c and reproduced on the next page.

Old (incorrect Fig. 4b)

New (correct) Fig. 5c

Reviewer Figure 1.

Figure 5. The hematopoietic response to stress varies across the skeleton. a, a, maps showing the distribution of granuloocyte progenitors (GP, red) and preneutrophils (green) in whole-mounted sternum, tibia and humerus treated with saline (S) or G-CSF (G, 250mg/kg/day) for 4 days. **b**, after G-CSF the sternum has fewer GP and the neutrophil sites disaggregate (n= 6 sternum, 6 tibia or 3 humerus from 6 mice per treatment). **c**, Number of indicated cells in sternum, tibia, and humerus quantified by FACS (n=4 mice in four independent experiments).

What are the peripheral neutrophil numbers in the saline versus G-CSF treated mice? Can the increased peripheral neutrophil number be accounted for by a 15% increase in cells in tibia?

G-CSF increases neutrophil numbers in peripheral blood from 0.8 to 10.6 million neutrophils per ml (**Reviewer Fig. 2a**). A 30-gram C57BL/6 male mouse has approximately 2.4ml of blood (J Clin Pathol. 1979 Jan; 32(1): 96). Thus, G-CSF increases peripheral neutrophil numbers from 1.9 to 25.4 million.

We performed flow cytometric analyses of mature neutrophil numbers across the skeleton. As shown in **Reviewer Fig. 2b**, all the bones examined – minus the sternum- have a 2-fold increase in neutrophil numbers after G-CSF. These results indicate that the increased neutrophils in peripheral blood are due to increased neutrophil production in all bones except the sternum. Femur and tibia (which show an increase in neutrophils after G-CSF of 8 and 5 million) are the biggest contributors to the overall neutrophil pool.

These contradictory points raise the issue of flux as a major possible confounder of these studies. As the authors acknowledge, cell numbers are not tracked over time in this system, and therefore if a production site is highly productive but the cells are rapidly released from that site, an accumulation of cells would not be evident. There is a possibility that the vascularity or flux out of the tibia is higher than out of the sternum in this particular instance, which would make it appear that the tibia is not producing much.

We have addressed this question in detail using confetti mice (see next paragraph). However, we would like to point out that the purpose of the experiments shown in Figure 5 was not to define the cellular mechanisms through which the sternum and tibia differentially respond to G-CSF (or how the skull fails to respond to hemorrhage). Instead, the purpose of these experiments is to highlight how interrogation of hematopoiesis in situ, using the imaging strategies and knowledge that we generated in Figures 1-4, can be used to discover new biology. In this case, heterogeneous hematopoietic responses to stress in different bones. To the best of our knowledge this is the first time that this has been reported.

To answer the referee's question in detail, we first performed detailed analyses of the neutrophil production sites in sternum, tibia, and humerus. Granulocyte production sites consist of a cluster of preneutrophils within 50µm of 1 or 2 GP (DOI: [10.1038/s41586-021-03201-2](https://doi.org/10.1038/s41586-021-03201-2) and **Supplemental Fig. 2f**). The frequency, spatial organization, and composition of these sites, in the saline-treated group, was identical for the three bones examined (**Extended Data Fig. 10a-c**).

Extended Data Figure 10. Heterogeneous stress responses across the skeleton. **a**, Number of neutrophil production sites per mm² (n = 4 sternum, 4 tibia from 4 mice and 3 humerus from 3 mice treated with saline (S) or treated with G-CSF for 4 days (G)). **b**, Number of CFU-E, MOP and Mature B cells in the indicated bones as quantified by FACS. Each dot represents one mouse in four different experiments. **c**, Representative images showing neutrophil production sites were lost in sternum and were disaggregated in the tibia and humerus after G-CSF treatment for 4 days. **d**, Number of PN per site (n = 78, 54, 28 neutrophil production sites from 3 sternum, 3 tibia and 3 humerus from 3 mice treated with saline (S) or n = 3, 14, 20 neutrophil production sites from 3 sternum, 3 tibia and 3 humerus from 3 mice treated with G-CSF for 4 days (G)). **e**, Distance from GP to the closest PN (n = 98, 88 and 34 GP cells from 6 sternum and 6 tibia from 6 mice or 3 humerus from 3 mice treated with saline (S) or n = 33, 27 and 33 GP from 3 sternum, 3 tibia and 3 humerus from 3 mice treated with G-CSF for 4 days). **f**, Representative images and distance analyses of confetti-labeled GP to the closest PN labeled in the same color (n = 15 confetti-labeled GP in 3 sternum and 3 tibia segments from 3 tamoxifen-treated confetti mice treated with saline (S) or G-CSF (G) for 4 days). **g**, Colony-forming assay for GP sorted from bone marrow of sternum or tibia treated with saline (S) or G-CSF (G) (n = 6 plate from 3 independent experiments).

After G-CSF, all neutrophil production sites were lost in the sternum but not in the long bones (**Extended Data Fig. 10a**). These reductions were due to the dramatic (2-fold) reductions in GP (which define the center of each production site) and preneutrophil numbers in the sternum (but not long bones, **Fig. 5a-b**). In all the bones G-CSF increased the distance from each preneutrophil to the GP (**Extended Data Fig. 10c-e**). These results suggested -as proposed by the referee- that G-CSF induces faster preneutrophil movement away from the GP.

To study this in detail we analyzed *Ubc-creERT2:confetti* mice. In this model transient Cre activation leads to irreversible GFP, YFP, RFP, or CFP expression in 7.3% of total cells. This allows examination of clonal relationships in short-lived cells^{1,32}. In agreement with our previous results¹, we found that – in the saline treated group, in all bones– GP fluorescently labeled by one of the four confetti alleles closely cluster (median distance 2.06µm) with 2-6 (median 3.5) preneutrophils labeled in the same color. After G-CSF, the number of preneutrophils labeled in the same color as the GP decreases 3-fold in tibia and sternum (**Extended Data Fig. 10f**) suggesting a loss of GP output at this time point of G-CSF treatment. We confirmed this loss of activity in colony forming assays (**Extended Data Fig. 10g**). Additionally, the preneutrophils with the same confetti label as the GP located much farther away from the GP than in the saline controls (**Extended Data Fig. 10f**). However, these distances were no different between tibia and sternum production sites (**Fig. 10f**). These results indicate that G-CSF induces preneutrophil migration away from the GP but that these cells are not more rapidly released from production sites in the sternum. We believe that these results fully address the referee's question.

Taken together these results indicate that the sternum produces less neutrophils than the tibia and humerus due to the dramatic loss (2-fold) in granulocyte progenitors (**Fig. 5a-c**) and not due to faster preneutrophil efflux from the production site.

For Figures 4a-c it seems that the conclusions are based on imaging of one sternum and one tibia in three mice. What is the reproducibility of these findings between two tibia in the same mouse?

This is an important control. To address this in detail we have compared the right and left tibia (and humerus for additional rigor), and the sternum in the same mice. As shown in **Reviewer Figure 3** on the next page, we found identical results when comparing the left and right bones of the same mouse. Importantly, in these same mice we find that G-CSF consistently induces dramatic reductions of granulopoiesis in the sternum but not the right or left tibia.

5) The authors state that “The tibia and sternum displayed similar suppression of erythropoiesis, monoopoiesis, and lymphopoiesis indicating that both bones are equally exposed to G-CSF (Fig. 4d).

This should state “suggesting that” otherwise the authors should spin down the bone marrow and measure the cytokine concentrations in different sites, or do qPCR to show the transcriptional response to G-CSF in different sites.

Great suggestion. We have edited this sentence as suggested, it now reads: “The tibia and sternum displayed similar suppression of erythropoiesis, monoopoiesis, and lymphopoiesis **suggesting that both bones are equally exposed to G-CSF (Extended Data Fig. 5b)”**

Reviewer Figure 3. G-CSF consistently induces dramatic reductions of granulopoiesis in the sternum but not the right or left tibia and humerus. **a**, Images and maps showing the distribution of granulocyte progenitors (GP, red) and preneutrophils (green) in whole-mounted sternum, right and left tibia or humerus from mice treated with saline or G-CSF (250mg/kg/day) for 4 days. (Images and maps arranged in one row are from the same mouse). **b**, Quantification of GP and PN cell numbers or Neutrophil production site numbers per 1mm² of sternum, right (R) and left (L) tibia or humerus (n= 3 bones each from the same three mice treated with saline (S) or G-CSF (G, 250mg/kg/day) for 4 days.

6) Little is said about the microanatomical location of HSCs during stress. Figure 4a shows the relative abundance of these cells after phlebotomy infection, etc, but the conclusion is limited to a general comment that the anatomy of the marrow is unchanged after stress. Does the HSC proximity to megakaryocytes change upon various types of stress? What about MPPs? In Figure 4B it looks like there is a proliferation of ST-HSCs and/or MPP2's after infection but this is not commented on.

We are grateful to the reviewer for highlighting this point. We have now performed extensive new analyses to quantify interaction of all types of HSPC with the microenvironment after stress. We now provide this information in detail (as heat maps) in **Extended Data Fig. 7e** (reproduced below) and as dot plots showing the distribution of distances in **supplemental Figure 8a, b** (reproduced on next page). Under all conditions the HPSC distribution is very similar to that in the steady state (single cells enriched near megakaryocytes and far from arterioles). Of all insults only *L. monocytogenes* infection causes a significant relocalization of all types of HSPC away from megakaryocytes when compared with steady-state mice. Note that this is not due to loss of megakaryocytes in infection (**Extended data Fig. 7d**). Because aging increases LT-HSC numbers by more than 5-fold

(**Extended Data Fig. 7e** and **Supplemental Fig. 8a, b**).

In the context of infection, we did find -as the reviewer highlights- increased MPP2/MyE numbers. Further we also found that -on occasion- 2 or 3 of these MPP2/MyE clustered together and this is shown in **Extended Data Fig 7f**). We did not detect ST-HSC clusters in the infection model.

In aged mice we also found rare (2-3 per mm²) clusters of LT-HSC and ST-HSC that are not observed in the young mice. These are imaged in **Supplemental Fig. 8c-e** and quantified in **Extended Data Fig. 7f**.

Extended Data Figure 7. Effect of the different insults in the organization of the microenvironment and distribution of multipotent HSPC. **e**, Dot plot summarizing the median distance from each HSPC to all other indicated cells and structures at the indicated time points after insult. (n = 41 LT-HSC, 66 ST-HSC, 30 MPP2/MyE, 41 MPP3/MPP4/CMP, 61 Mkp, and 82 Pre Meg-E in 4 sternum segments from 4 mice in steady-state for control; n = 21 LT-HSC, 35 ST-HSC, 16 MPP2/MyE, 30 MPP3/MPP4/CMP, 38 Mkp, and 73 Pre Meg-E in 3 sternum segments from 3 mice two days after phlebotomy; n = 15 LT-HSC, 19 ST-HSC, 56 MPP2/MyE, 39 MPP3/MPP4/CMP, 17 Mkp, and 57 Pre Meg-E in 3 sternum segments from 3 mice six days after infection). n = 10 LT-HSC, 16 ST-HSC, 11 MPP2/MyE, 23 MPP3/MPP4/CMP, 51 Mkp, and 16 Pre Meg-E in 3 sternum segments from 3 mice four days of G-CSF treatment. n = 300 LT-HSC, 236 ST-HSC, 39 MPP2/MyE, 32 MPP3/MPP4/CMP, 133 Mkp, and 57 Pre Meg-E in 3 sternum segments from 3 20-month old mice. LT-HSC in aged mice contain CD41+ and CD41- LT-HSC identified as shown in the next panels. **f**, Quantification of MPP2/MyE clusters six days after infection (n=3 sternum segments from 3 mice in steady-state for control, n= 3 sternum segments from 3 mice six days after infection) and LT-HSC and ST-HSC clusters in aging (n=7 sternum segments from 4 mice in steady-state for control, n=7 sternum segments from 4 20-month old mice).

Supplementary Figure 8. Myeloid and Lymphoid biased LT-HSC distribution in aged mice. **a,b**, Distance from each HSPC to all other indicated cells (a) and structures (b) at the indicated time points after insult. (n = 41 LT-HSC, 52 ST-HSC, 25 MPP2/MyE, 41 MPP3/MPP4/CMP, 61 MkP, and 82 Pre Meg-E in 4 sternum segments from four mice in steady-state; n = 21 LT-HSC, 35 ST-HSC, 16 MPP2/MyE, 30 MPP3/MPP4/CMP, 38 MkP, and 73 Pre Meg-E in 3 sternum segments from three mice two days after phlebotomy; n = 15 LT-HSC, 19 ST-HSC, 56 MPP2/MyE, 39 MPP3/MPP4/CMP, 17 MkP, and 57 Pre Meg-E in 3 sternum segments from three mice six days after infection ; n = 10 LT-HSC, 16 ST-HSC, 11 MPP2/MyE, 23 MPP3/MPP4/CMP, 52 MkP, and 16 Pre Meg-E in 3 sternum segments from three mice four days of G-CSF treatment n = 300 LT-HSC, 236 ST-HSC, 39 MPP2/MyE, 72 MPP3/MPP4/CMP, 133 MkP, and 57 Pre Meg-E in 3 sternum segments from three 20-month old mice. LT-HSC in aged mice contain CD41+ and CD41-LT-HSC identified as shown in the next panels. Statistical differences were calculated using two-way ANOVA t-tests if the distributions were normal or Kruskal-Wallis test if not normal; P values are shown.

Referee #3 (Remarks to the Author):

Wu et al. have strengthened the manuscript by adding significant new data and by clarifying some of their conclusions. I have only minor comments for them to consider, all of which could be addressed by alterations to the text.

We are grateful to the referee for all the great suggestions. The requested changes have been addressed as indicated in the point-by-point response below.

1. The authors argue that pairs of HSCs are rarely observed adjacent to each other not because of the mobilization of one or both daughter cells from an HSC division into circulation because "in the steady-state a minute fraction of total HSPC is released into circulation". But only a minute fraction of HSCs is dividing at any one time. So the number of HSCs in circulation may be completely consistent with the number of HSCs that are dividing at any one time. My sense is that it's still not clear why pairs of HSCs are rarely observed together, and whether this reflects the migration of HSCs away from each other within the bone marrow, as favored by the authors, or the intravasation of one or both daughter cells into circulation, or the death or differentiation of one of the daughter cells.

The referee is making an important point as the experiments do not allow us to distinguish between these possibilities. We have thus removed the statements highlighted by the referee in points #1-33. The text now reads (changes in bold) "**These results indicate that LT-HSC and other multipotent progenitors are not adjacent to each other.** Since, after cell division, daughter cells are necessarily adjacent, the results also indicated that the offspring of HSPC were either released into the circulation, differentiated into more mature cells that are not detected in the HSPC stain, or moved away from each other. Intravital imaging studies support different degrees of HSC motility in the marrow⁹⁻¹². Because most HSC are found as single cell in the marrow (^{28,29} and Fig. 2a) the mobility of HSC when adjacent to other HSC/MPP has not been examined. To explore this in detail we performed follow-up analysis of microscopy-guided transplantation of single Dil-labeled, Tie2⁺ HSC in the mouse calvarium as described ³⁰. In one instance we observed that (48 hours after the initial transplant) the sole transplanted HSC had divided generating two Dil-labeled, Tie2⁺ cells that were in close proximity. Three hours later one of the daughter cells was no longer visible in the whole calvarium suggesting that it had moved away or died (Extended Data Fig. 4e). These results led us to hypothesize that HSC move away from each other when in close proximity. Follow-up analysis of single vs multiple cell transplants (5 recipients received one LT-HSC and 6 recipients received respectively 5, 5, 5, 17, 19, and 22 Tie2⁺ labeled LT-HSC, Fig. 2c). We found that a single cell was visible in 80% of the single cell recipients (4 out of 5) and the cells were all found in near (<100µm) of the transplantation site. Only one out of 73 cells was found –as a single cell– in the multiple recipients (Fig. 2d). Importantly, all tracked recipients showed long-term HSC engraftment that correlated with the number of HSC transferred (Supplementary Table 3 and ³⁰) indicating that the absent HSC did not die or terminally differentiate. These experiments indicate that HSC move away from each other, when in close proximity, in vivo".

2. The authors note that "in vivo barcoding showed that MPP can extensively self-renew in vivo for long periods of time". My impression is that these data show that MPPs can persist for long periods of time in vivo while contributing to hematopoiesis but I'm not aware of evidence that they extensively self-renew while doing so.

Aren't the data consistent with the possibility that MPPs persist for long periods of time, while remaining mainly quiescent and dividing intermittently? The MPPs might divide only a handful of times despite persisting for long periods of time.

As indicated in the response to point #1 we have removed this statement and edited the text to avoid confusion.

3. The authors concluded "These results indicate that LT-HSC and other multipotent progenitors don't share a physical location and are thus not regulated by the same adjacent stromal cells". I realize the authors have already tried to clarify this statement but its meaning is still unclear. Is the point that clonally-related HSCs and their immediate progeny don't remain in the same location, such that they don't remain under the influence of the same stromal cell? Or is this a more general statement about cells that are not necessarily clonally related? Is this statement intended to imply that HSCs and MPPs do not obtain growth factors from the same class of stromal cells? My impression is that isn't what the authors meant, but it could be read that way. The published data show that all multipotent HSPCs depend upon SCF synthesized by LepR+ stromal cells.

This comment made us realize that our statement was still confusing. We just wanted to indicate that the multipotent stem and progenitors were not adjacent to each other. We have replaced the statement highlighted by the referee as follows: **"These results indicate that LT-HSC and other multipotent progenitors are not adjacent to each other"**.

4. The authors stated that "No HSPC showed specific interaction with the... sinusoids, although, due to the abundance of sinusoids most multipotent HSPCs localized within 10 um of these vessels". What does this mean? The authors seem to imply that while HSCs and MPPs all localized adjacent to sinusoids, that this doesn't reflect a "specific interaction" with the sinusoids but rather it occurs by chance as a result of the abundance of sinusoids throughout the bone marrow. But I don't think there's any evidence to support that way of interpreting the observation. The key question is whether there's something special about the environment immediately around the sinusoids that helps to promote the maintenance of HSCs, MPPs, and some early restricted progenitors. It's quite possible that there is. For example, a specific subset of LepR+ cells that is fated to form adipocytes and that produces high levels of HSC niche factors seems to localize to sinusoids and not to arterioles. Megakaryocytes also localize to the sinusoids for a reason – they are able to insert processes through the fenestrated walls of sinusoids to deposit platelets into circulation. I don't think the data favor the idea that the cells that are very closely associated with sinusoids are there by chance alone – there are likely functionally important attributes of that environment that are caused by the proximity to sinusoids.

This is an important point. It was not our intention to suggest that sinusoids are not a key structure that maintains HSPC. We only indicated that -in our hands- the HSPC distance to the closest sinusoid was not different from that of random cells. We also speculated that this was likely because there are so many sinusoids that all cells are proximal to these structures. To address this critique the text now reads (changes in bold): **"Even though sinusoids are a key regulatory niche for LT-HSC²⁷**, no HSPC subset preferentially localized to sinusoids or the endosteum (including transcortical blood vessels) when compared with random

cells. This is likely due to the abundance of sinusoids as most random cells and multipotent HSPC localized within 10 μm of these vessels (Fig. 2a and Extended Data Fig. 4b-d)”

Reference 27 corresponds to: “Acar, M. *et al.* Deep imaging of bone marrow shows non-dividing stem cells are mainly perisinusoidal. *Nature* **526**, 126-130, doi:10.1038/nature15250 (2015)”

5. The authors might note in the erythroid progenitor section that Comazzetto et al. also showed that early erythroid progenitors localize to sinusoids.

We apologize for this oversight. This has been corrected as follows: “Pre CFU-E separated from Pre Meg-E and localized in the sinusoids (60% in direct contact) but did not map near CFU-E. **A previous study showed that CFU-E localized to sinusoids**²⁵. In agreement, we found that CFU-E were found in large strings of 3 to 23 (Mean = 8 ± 4) CFU-E decorating the surface of a single sinusoid and away from arterioles and the endosteum (Fig. 2e-g and Extended Data Fig. 4a, b).”

Reference 25 corresponds to: “Comazzetto, S., Shen, B. & Morrison, S. J. Niches that regulate stem cells and hematopoiesis in adult bone marrow. *Dev Cell* 56, 1848-1860, doi:10.1016/j.devcel.2021.05.018 (2021).”

Referee #4 (Remarks to the Author):

The authors have addressed all my comments and have dramatically improved the clarity of the manuscript and the explanation of the results for the reader. It is now clear how they describe the localization of HSPC within the BM with respect to other hematopoietic and non-hematopoietic cells. The experiments tracking single HSC divisions are very important and interesting.

The final description of the anatomy of hematopoiesis has dramatically improved by including imaging of different bones, which show very reproducible patterns, distinguishing HSPCs that localize to distinct niches (MK) compared to progenitors that localize at specific vessels (arterioles for CLPs and sinusoids for neutrophils and erythroid lineages) without overlapping. Moreover, it is very interesting the observation that committed progenitors form clusters with daughter cells, while HSPCs separate from each other, which is an intrinsic characteristic of these different cell types, since reproducible also in vitro.

The data of the macroanatomy and microanatomy after all the different stressors are also extremely improved and overall, the manuscript includes now an impressive amount of high quality data.

We are very grateful for all the critiques and suggestions that helped in improving the manuscript. The requested changes have been addressed as indicated in the point-by-point response below.

A few focused points to be addressed:

1) The results in Fig. 2c-d should be clarified: lines 178-180: I understand that after 1 hour single transplanted HSC and 5 transplanted HSCs are visible at the transplantation site. I don't understand why they claim that only 1 out of 73 cells was found as single cell in the multiple (HSC) recipients... After how long they look at the transplantation site? 48hrs? what happened in most of the cases? No cells detected? Or did they detect more than 1 cell? Can the author explain better the results? The authors could include this info in the supplementary table 3 with one column indicating the number of HSC detected after 1 hour and another column indicating the number of HSC detected after 48 hrs.

Great suggestion. This information is now included in Supplementary Table 3 (see next page) as requested. To make this point clear the text (lines 174-183) now reads: "Follow-up analyses of single vs multiple cell transplants (5 recipients received one LT-HSC and 6 recipients received respectively 5, 5, 5, 17, 19, and 22 Tie2⁺ labeled LT-HSC; the transplanted cells were visualized 15 minutes after the transfer to confirm correct delivery; the same region was imaged 24 hours later. Fig. 2c) showed that the single transplanted cell was detected in four out of the five single cell recipients. In contrast, a single donor cell was visualized in one out of six recipients transplanted with multiple HSC while no donor cells were detected in the rest of the multiple HSC recipients (Fig. 2d and Supplementary Table 3). Importantly, all tracked recipients showed long-term HSC engraftment that correlated with the number of HSC transferred (Supplementary Table 3 and ³⁰) indicating that the absent HSC did not die or terminally differentiate. These experiments indicate that HSC move away from each other, when in close proximity, in vivo".

Recipient	number of transplanted cells	number of transplanted cells found 24 hrs after transplantation	fraction of transplanted cells found 24 hrs after transplantation	Peripheral blood chimerism (week 4)
M1	1	1	1	not tracked
M2	1	1	1	1.14
M3	1	1	1	0.46
M4	1	0	0	0.23
M5	1	1	1	2.76
M6	5	0	0	2.59
M7	5	1	0.2	not tracked
M8	5	0	0	5.4
M9	17	0	0	not tracked
M10	19	0	0	69.4
M11	22	0	0	13.3

Supplementary Table 3. Quantification of imaging analyses and peripheral blood engraftment after microscopy-guided HSC transplantation

2) The heatmap of fig 3a (Now Fig 4a) is based on which dataset? Please clarify.

This is based on the flow cytometric datasets shown in Supplementary Figures 3-6. To clarify this the legend to Fig 4a now reads (changes in bold): “a, Heat map summarizing changes (normalized to steady-state or saline-treated) to the indicated populations (exemplars of peak erythropoiesis, lymphopoiesis, granulopoiesis, and myelopoiesis responses **as shown in Supplementary Figures 4-7**) two days after phlebotomy (except for lymphopoiesis which corresponds to day 8 as this is the nadir of the lymphopoiesis response as shown in Supplementary Figure 3), six days after *L. monocytogenes* infection, four days of G-CSF treatment, and 20 months of age (n=7, 6, 4 and 6 mice for each indicated population two days after phlebotomy, six days after phlebotomy, 4 days of G-CSF treatment and 20-month old aging mice).

3) In extended data figure 7 the number of sinusoids, arterioles and Mk is quantified per fragment. The same applies to the number of production sites quantified in Fig. 3c, e, g and i. Fragment size might be different (for example in aged bones). Can the authors normalize the measurements based on imaged BM area or BM volume like in Figure 4b?

Great suggestion. The data (now Fig. 4c,e,g,i) has been formatted as requested.

4) Lines 258-259: the authors report on some changes in MPP2/MyE after infection and of HSCs upon aging. These observations are very interesting, while underscoring specific changes in HSPC upon infection that are different from aging. These observations deserve a more careful quantification. Can the authors analyze the statistic of extended data figure 7c and highlight the data that support the claim? Also, the authors define these clusters “rare”? can it be quantified better?

We are grateful for this comment. As referee #2 also requested us to highlight these interactions (HSPC with each other and the microenvironment) we have replaced the data on **Extended Data Fig. 7e** with heat maps showing the changes -and statistical analyses- in localization for each HSPC subset with all other cells and structures (reproduced on the next page). The **original dot plots** for interactions with other HSPC (old Extended Data Fig. 7e are now available in **Supplementary Figure 8a**). The dot plots for interactions of HSPC with the microenvironment are now available in **Supplementary Figure 8b**.

In the context of infection, we did find that -on occasion- 2 or 3 of these MPP2/MyE clustered together and this is shown and quantified (1-2 clusters per mm²) in **Extended Data Fig 7f**). In aged mice we also found rare (2-3

per square mm²) clusters of LT-HSC and ST-HSC that are not observed in the young mice. These are described in **Supplementary Fig. 8c-e** and quantified in **Extended Data Fig. 7f**.

Extended Data Figure 7. Effect of the different insults in the organization of the microenvironment and distribution of multipotent HSPC.

e, Dot plot summarizing the median distance from each HSPC to all other indicated cells and structures at the indicated time points after insult. (n = 41 LT-HSC, 52 ST-HSC, 25 MPP2/MyE, 41 MPP3/MPP4/CMP, 61 MkP, and 82 Pre Meg-E in 4 sternum segments from four mice in steady-state; n = 21 LT-HSC, 35 ST-HSC, 16 MPP2/MyE, 30 MPP3/MPP4/CMP, 38 MkP, and 73 Pre Meg-E in 3 sternum segments from three mice two days after phlebotomy; n = 15 LT-HSC, 19 ST-HSC, 56 MPP2/MyE, 39 MPP3/MPP4/CMP, 17 MkP, and 57 Pre Meg-E in 3 sternum segments from three mice six days after infection n = 10 LT-HSC, 16 ST-HSC, 11 MPP2/MyE, 23 MPP3/MPP4/CMP, 52 MkP, and 16 Pre Meg-E in 3 sternum segments from three mice four days of G-CSF treatment n = 300 LT-HSC, 236 ST-HSC, 39 MPP2/MyE, 72 MPP3/MPP4/CMP, 133 MkP, and 57 Pre Meg-E in 3 sternum segments from three 20-month old mice. LT-HSC in aged mice contain CD41+ and CD41- LT-HSC identified as shown in the next panels. **f**, Quantification of MPP2/Mye clusters six days after infection (n=3 sternum segments from 3 mice in steady-state for control, n= 3 sternum segments from 3 mice six days after infection) and LT-HSC and ST-HSC clusters in aging (n=7 sternum segments from 4 mice in steady-state for control, n=7 sternum segments from 4 20- μ m).

5) The “production site” as it is defined and investigated in this paper is novel and quite provocative. Each production site identifies discrete BM regions (microanatomy). The size of each production site varies according to the median distance of random cells from each specific progenitor. To improve the clarity for the reader, I would suggest including a table to recapitulate the general definition of “production site”, “microanatomy”, “macroanatomy” and the specific definition of each (for example as per Extended data Figure 4: B cell production site includes pre-proB, pro B and pre B cells closer than 150 μ m (median distance of random cells) to the CLP).

Thank you and very good suggestion. This table is now included as Supplementary Table 4 (see below).

Production site	Definition	Notes
Erythroid	All erythroid lineage cells within 50µm around a sinusoid decorated with 3 or more adjacent CFU-E.	See Extended Data Fig. 5h for step by step identification
Lymphoid	All Pre-pro B, Pro B, and Pre B cells within 150µm of a CLP	See Extended Data Fig. 5n for step by step identification
Mono/DC	All Ly6Clo monocytes and MHCIIretic dendritic cells within 100µm of a MDP	See Supplementary Fig. 3e for step by step identification
Neutrophil	All Preneutrophils within 50µm of a GP. After stress (infection, G-CSF, or aging) we found numerous GP that were close to each other (<20µm). In this case we considered them to be part of the same production site.	See Supplementary Fig. 3f for step by step identification

Supplementary Table 4. Definition of the production sites examined in this manuscript.

Reviewer Reports on the Second Revision:

Referees' comments:

Referee #2 (Remarks to the Author):

The authors have done a very thorough and extensive job responding to the prior critiques. Whereas I could continue the conversation, none of the questions I have should prevent the publication of the paper, which is an important contribution to the field, changes the paradigm about systemic effects of G-CSF and heterogeneity of responses between bones, and will generate many questions for future research.

Referee #4 (Remarks to the Author):

The authors have addressed all my last requests carefully and I am very happy with the final result. The manuscript is now sounding, clear and containing an impressive novel set of data for the scientific community. I have no further questions and I congratulate the authors.

Author Rebuttals to Second Revision:

Point-by-point response to referees.

Referee #2 (Remarks to the Author):

The authors have done a very thorough and extensive job responding to the prior critiques. Whereas I could continue the conversation, none of the questions I have should prevent the publication of the paper, which is an important contribution to the field, changes the paradigm about systemic effects of G-CSF and heterogeneity of responses between bones, and will generate many questions for future research.

We thank the referee for the positive review and all the suggestions that have improved the manuscript.

Referee #4 (Remarks to the Author):

The authors have addressed all my last requests carefully and I am very happy with the final result. the manuscript is now sounding, clear and containing an impressive novel set of data for the scientific community. I have no further questions and I congratulate the authors.

We thank the referee for the positive review and all the suggestions that have improved the manuscript.